# Single-cell nascent RNA sequencing unveils coordinated global transcription

Dig B. Mahat[1], Nathaniel D. Tippens[1], Jorge D. Martin-Rufino[2], Sean K. Waterton[1,3], Jiayu Fu[1,4], Sarah E. Blatt[1,5] & Phillip A. Sharp[1✉]

Transcription is the primary regulatory step in gene expression. Divergent transcription initiation from promoters and enhancers produces stable RNAs from genes and unstable RNAs from enhancers[1,2]. Nascent RNA capture and sequencing assays simultaneously measure gene and enhancer activity in cell populations[3]. However, fundamental questions about the temporal regulation of transcription and enhancer–gene coordination remain unanswered, primarily because of the absence of a single-cell perspective on active transcription. In this study, we present scGRO–seq—a new single-cell nascent RNA sequencing assay that uses click chemistry—and unveil coordinated transcription throughout the genome. We demonstrate the episodic nature of transcription and the co-transcription of functionally related genes. scGRO–seq can estimate burst size and frequency by directly quantifying transcribing RNA polymerases in individual cells and can leverage replication-dependent non-polyadenylated histone gene transcription to elucidate cell cycle dynamics. The single-nucleotide spatial and temporal resolution of scGRO–seq enables the identification of networks of enhancers and genes. Our results suggest that the bursting of transcription at super-enhancers precedes bursting from associated genes. By imparting insights into the dynamic nature of global transcription and the origin and propagation of transcription signals, we demonstrate the ability of scGRO–seq to investigate the mechanisms of transcription regulation and the role of enhancers in gene expression.

Transcription is a discontinuous process characterized by short bursts and long inter-burst silent periods[4,5]. Decoding the origin and circuits of burst signals is crucial for understanding the mechanisms of transcription regulation during the cell cycle, development and disease. Core promoter elements, transcription factors and enhancers are implicated in the regulation of burst kinetics, but their precise role in determining overall transcription output remains unsettled[6,7]. Whether the widely accepted view of stochastic transcription of individual genes conceals co-transcription of functionally related genes and coordination between enhancer–gene pairs holds broad significance in understanding gene regulation. From a clinical perspective, assessing the contribution of enhancers in the regulation of protein-coding genes can unlock a largely unexplored genomic landscape for therapeutics.

Active enhancers are occupied by transcription factors and RNA polymerase, similar to the gene promoters they regulate, which results in the synthesis of non-coding, non-polyadenylated and unstable RNA[3,8]. Enhancers are highly specific to cell types and states[9], and exert *cis*-regulatory effects over long genomic distances[10]. Genome-wide association studies further underscore the role of enhancers in gene regulation, showing that more than 90% of genomic loci associated with traits and diseases are found in non-coding regions with many overlapping enhancers[11]. However, linking enhancers that harbour causal variants to genes remains challenging. Although low throughput, genome-editing tools can potentially map enhancer–gene pairs, but the pleiotropic nature[6] and weak effect of individual enhancers hinder their utility.

Existing genomic tools that probe the coding and non-coding genome without perturbation by assessing chromatin conformation, histone modifications and chromatin accessibility have shed light on the molecular events that lead up to enhancer-mediated gene activation. However, these tools do not fully confirm the actual activation event[12]. Despite having similar chromatin features, the distinguishing feature of an active enhancer from its inactive counterpart is its transcription[13]. Nascent RNA sequencing assays, such as global run-on and sequencing (GRO–seq)[2] and precision run-on and sequencing (PRO–seq)[14], enable the simultaneous quantification of transcription in genes and enhancers. However, these bulk cell assays average the discontinuous transcription from individual cells, which makes it challenging to decipher transcription dynamics and to assign enhancer–gene relationships.

Here we present a new single-cell nascent RNA sequencing method, which we term scGRO–seq, that uses copper(I)-catalysed azide-alkyne cycloaddition (CuAAC or click chemistry)[15] to assess genome-wide nascent transcription in individual cells in a quantitative manner. Our

[1]Koch Institute for Integrative Cancer Research and Department of Biology, Massachusetts Institute of Technology, Cambridge, MA, USA. [2]Broad Institute of MIT and Harvard, Cambridge, MA, USA. [3]Present address: Department of Biology, Stanford University, Stanford, CA, USA. [4]Present address: Interdisciplinary Biological Sciences Graduate Program, Northwestern University, Evanston, IL, USA. [5]Present address: Exact Sciences, Madison, WI, USA. ✉e-mail: sharppa@mit.edu

analyses of genes and enhancers across 2,635 individual mouse embryonic stem (ES) cells provide a comprehensive view of the dynamic nature of transcription. We leverage elongating RNA polymerases as built-in clocks and measure the distance travelled from the transcription start site (TSS) to estimate transcriptional burst kinetics. Using a class of cell-cycle-phase-specific genes undetected by most single-cell methods, we quantify the dynamics of transcription during the cell cycle. We use the single-nucleotide temporal resolution of genome-wide transcription in individual cells to reveal the co-transcribed gene–gene and enhancer–gene networks that are turned on within a few minutes of each other. Using a set of validated enhancer–gene pairs, our data suggest that transcription initiates at enhancers before the activation of transcription at the associated genes. Overall, scGRO–seq bridges a gap in the study of temporal control of transcription and the functional association of enhancers and genes. These insights will shed light on gene regulatory mechanisms in essential cellular processes and disease.

## Development of scGRO–seq

The primary challenge in capturing and sequencing nascent RNA from individual cells is attaching unique single-cell tags onto nascent RNA. Existing nascent RNA sequencing methods selectively capture tagged nascent RNA from a cell population, which makes single-cell deconvolution impossible. By contrast, single-cell RNA sequencing (scRNA-seq) methods capture mRNA by annealing with the poly(A) tail and attaching single-cell barcode sequences by reverse transcription (RT). Nascent RNA lacks a terminal poly(A) tract or any other consensus sequence and must be selectively labelled and enriched from abundant total cellular RNA.

We designed a new strategy to selectively label nascent RNA through a nuclear run-on reaction in the presence of modified nucleotide triphosphates (NTPs) compatible with CuAAC conjugation. CuAAC is highly efficient and selective, robust under diverse reaction conditions, enzyme-free and compatible with automation. First, we developed, optimized and systematically characterized an assay for genome-wide transcriptome using click chemistry (AGTuC): a cell-population-based nascent RNA sequencing method that uses 3′-(O-propargyl)-NTPs in mouse ES cells (Extended Data Figs. 1a–f and 2a–d). It takes about 8 h to prepare an AGTuC library. However, the high concentration of ionic detergent in AGTuC disrupts nuclear membranes during the run-on reaction, which makes RNA from individual cells indistinguishable for single-cell barcoding. We therefore developed an iteration of AGTuC whereby nascent RNAs in individual nuclei are labelled with alkyne through run-on with 3′-(O-propargyl)-NTPs but without disrupting the nuclear membrane (termed intact-nuclei AGTuC (inAGTuC)) (Extended Data Figs. 3a–j and 4a–h). We prepared inAGTuC libraries in 96-well plates with 12 cells per well (c.p.w.), 120 c.p.w. and 1,200 c.p.w. (which is roughly equivalent to 1,000, 10,000 and 100,000 nuclei, respectively). We tested for correlation between this method and with PRO–seq (Extended Data Fig. 5a–d), and the results demonstrated the feasibility of profiling nascent RNA with small sample sizes. Based on the correlation slope, the inAGTuC library with as low as about 1,000 nuclei showed similar efficiency as PRO–seq in detecting nascent transcriptomes. The higher efficiency, lower cost, shorter library preparation time and lower sample input make AGTuC and inAGTuC viable alternatives to existing methods such as PRO–seq. By enabling the compartmentalization of intact nuclei that contain click-compatible nascent RNA and 5′-azide single-cell-barcoded (5′-AzScBc) DNA molecules using fewer nuclei, inAGTuC laid the ground for single-cell nascent RNA sequencing.

Building on this foundation, we applied our newly developed chemistry to single cells (Fig. 1a). For congruence with the original nascent RNA sequencing method of GRO–seq, we named this single-cell version scGRO–seq. Intact nuclei containing nascent RNA labelled with propargyl, following a nuclear run-on reaction with 3′-(O-propargyl)-NTPs,

were sorted individually into 96-well plates. Each well contained a small volume of 8 M urea, which lyses the nuclear membrane and denatures RNA polymerase and releases propargyl-labelled nascent RNA. The addition of CuAAC reagents led to the covalent linkage of propargyl-labelled nascent RNA to a unique 5′-AzScBc DNA molecule in each well. After CuAAC, single-cell-barcoded nascent RNAs from 96 wells were pooled, reverse transcribed in the presence of a template switching oligonucleotide (TSO), PCR amplified and sequenced (Extended Data Fig. 6). Despite a span of more than 3 years between the generation of various scGRO–seq library replicates, the different batches showed strong correlation at the level of the 96-well plate (Extended Data Fig. 7a).

The scGRO–seq results recapitulated the inAGTuC and PRO–seq profiles at both genes and enhancers (Fig. 1b) and provided a comprehensive map of nascent transcription in individual cells. We performed 17 batches of scGRO–seq experiments with 39 96-well plates and 3,744 cells, of which 36 plates and 2,635 cells passed the threshold (Methods). We captured an average of 3,665 reads and 1,503 features (genes and enhancers) per cell (Fig. 1c and Extended Data Fig. 7b). Moreover, pseudo-bulk scGRO–seq counts from collapsed single cells in genes and enhancers correlated well with bulk counts from inAGTuC (Fig. 1d). An analysis of the sequencing depth indicated the possibility that more reads and features per cell could be discovered with further development of the technology and deeper sequencing (Extended Data Fig. 7c). However, scGRO–seq is less efficient in capturing nascent RNA from promoter–proximal pause sites. We attribute this limitation to the reduced run-on efficiency of paused RNA polymerase II (PolII) in the absence of a high concentration of strong detergent[16]. This difference in promoter–proximal run-on efficiency was reflected in the reduced correlation between scGRO–seq and PRO–seq libraries (Extended Data Fig. 7d), as well as in the metagene profiles around the TSS of genes and enhancers (Extended Data Fig. 7e,f).

After confirming that scGRO–seq recapitulates results from bulk nascent RNA sequencing methods, we benchmarked scGRO–seq against other RNA-based single-cell assays. The closest single-cell method that probes nascent transcription is intron seqFISH, which is a multiplexed single-molecule in situ nascent RNA hybridization and imaging method[17]. We confirmed that the correlation between scGRO–seq and intron seqFISH is similar to the correlation reported between intron seqFISH and GRO–seq (Fig. 1e). By contrast, scGRO–seq poorly correlated with scRNA-seq (Extended Data Fig. 1g), which is probably due to differences in mRNA stability and capture methods. Nevertheless, as expected, scGRO–seq reads were more likely to be intronic or intergenic than scRNA-seq reads (Fig. 1f). Overall, the suite of genomic assays presented here utilizes a new biochemical approach to provide a snapshot of genome-wide transcription at various cell resolutions, including individual cells.

## Direct measurement of burst kinetics

Estimates of transcriptional kinetics primarily come from low-throughput live-cell imaging or fluorescent in situ hybridization in fixed cells[18,19]. The intron seqFISH method is limited to predefined gene targets, requires specialized probes and assumes that all intronic RNAs have the same kinetic fate. Approaches based on next-generation sequencing (NGS) are comprehensive and technically more accessible. However, the current methods measure polyadenylated mRNA from single cells[20] and fit a simple two-state mathematical model to infer transcriptional kinetics[7]. Bridging this gap, scGRO–seq combines high-throughput measurement of transcription with NGS, thereby enabling the detection of transcribing RNA polymerases genome-wide at single-nucleotide resolution (Fig. 2a).

With this new approach, we examined the evidence of bursting de novo without previous assumptions by quantifying the incidence of transcribing RNA polymerases. If transcription occurs in bursts, we

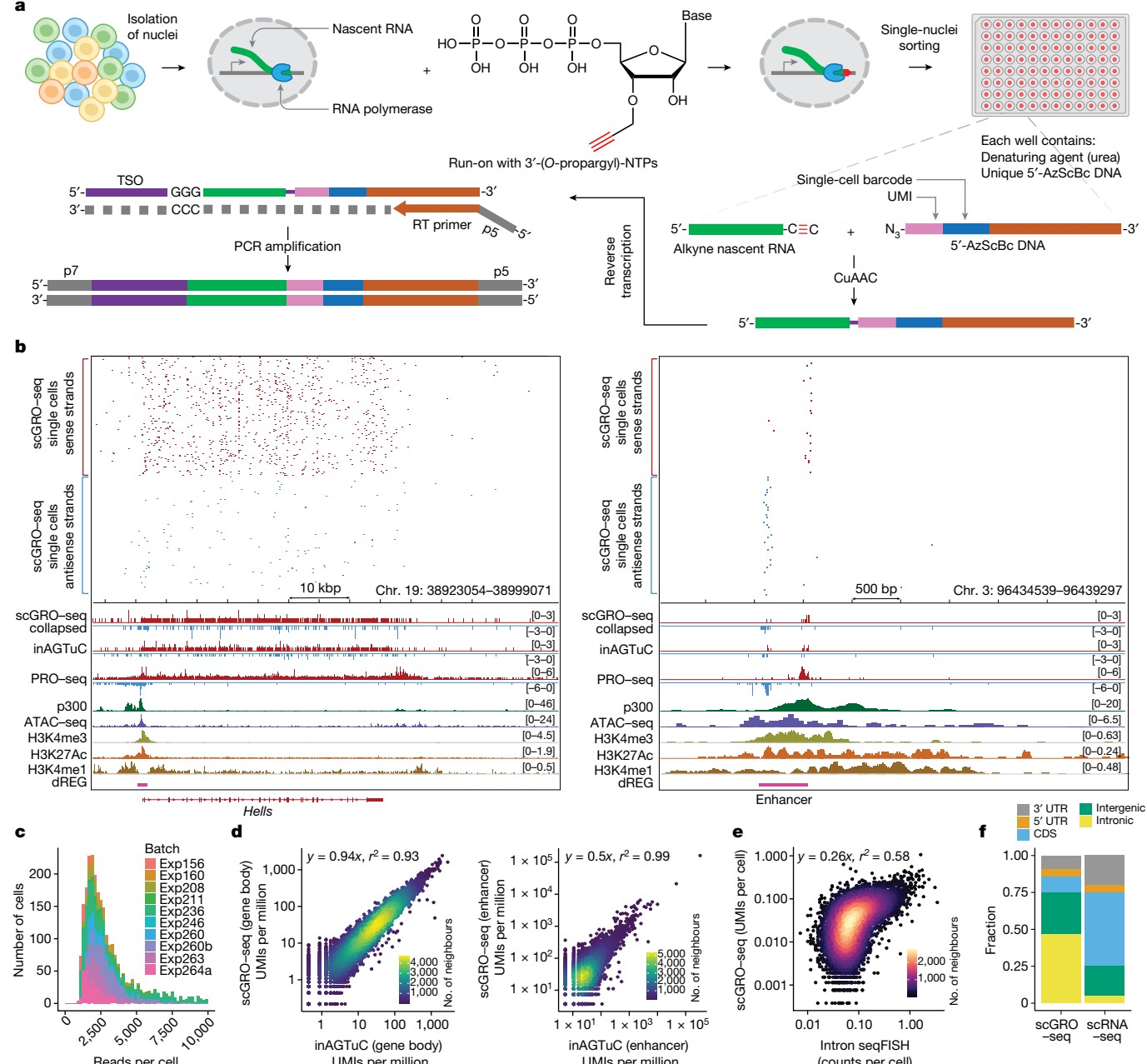

**Fig. 1 | Schematics and benchmarking of single-cell nascent RNA sequencing. a**, A summary of the scGRO–seq workflow. **b**, Representative genome browser screenshots showing scGRO–seq UMIs at a single-cell resolution, the aggregate scGRO–seq profile, the inAGTuC profile, the PRO–seq profile and chromatin marks around a gene (left) and an enhancer (right). **c**, Distribution of scGRO–seq UMIs per cell. **d**, Correlation between aggregate scGRO–seq and inAGTuC UMIs per million sequences in the body of genes (left, *n* = 19,961) and enhancers (right, *n* = 12,542). UMIs from the 500 bp regions from each end of the genes and 250 bp regions from each end of the enhancers analysed were removed to only include nascent RNA from elongating RNA polymerases. Data are plotted on a log–log scale to show the range of data distribution. **e**, Correlation between scGRO–seq UMIs per cell from up to the first 20 kb of genes and intron seqFISH counts per cell in the body of genes used in the intron seqFISH study (*n* = 9,666). **f**, Distribution of scGRO–seq and scRNA–seq UMIs in various genomic regions.

would anticipate a higher occurrence of more than one RNA polymerase per burst (multiplets) than would be expected by chance. Based on the approximately 10% capture efficiency of scGRO–seq estimated from comparison with intron seqFISH (Methods), the probability of detecting two consecutive RNA polymerases on a gene is 1%. To account for differences in unique molecular identifiers (UMIs) per cell, we devised a null model using permutation. We permuted reads among cells while keeping UMIs per cell and polymerase position unchanged (Methods; *n* = 200 permutations). We then compared the real data to the permuted control data and observed fewer singlets (*n* = 1,052, false discovery rate

(FDR) = 0.05) and a greater number of multiplets (*n* = 828, FDR = 0) in the real data, which provided evidence for the bursting nature of transcription (Fig. 2b). This result represents a significant 2.4% excess of multiplets in real data compared with permuted data. Transcriptional bursting would also result in more closely spaced RNA polymerases than what would be observed by random chance. When examining the distance between multiplets, we observed enrichment of closely spaced RNA polymerases (*P* < 0.05, two-sample Kolmogorov–Smirnov (KS) test) (Fig. 2c and Extended Data Fig. 8a), which further strengthened the evidence of bursting.

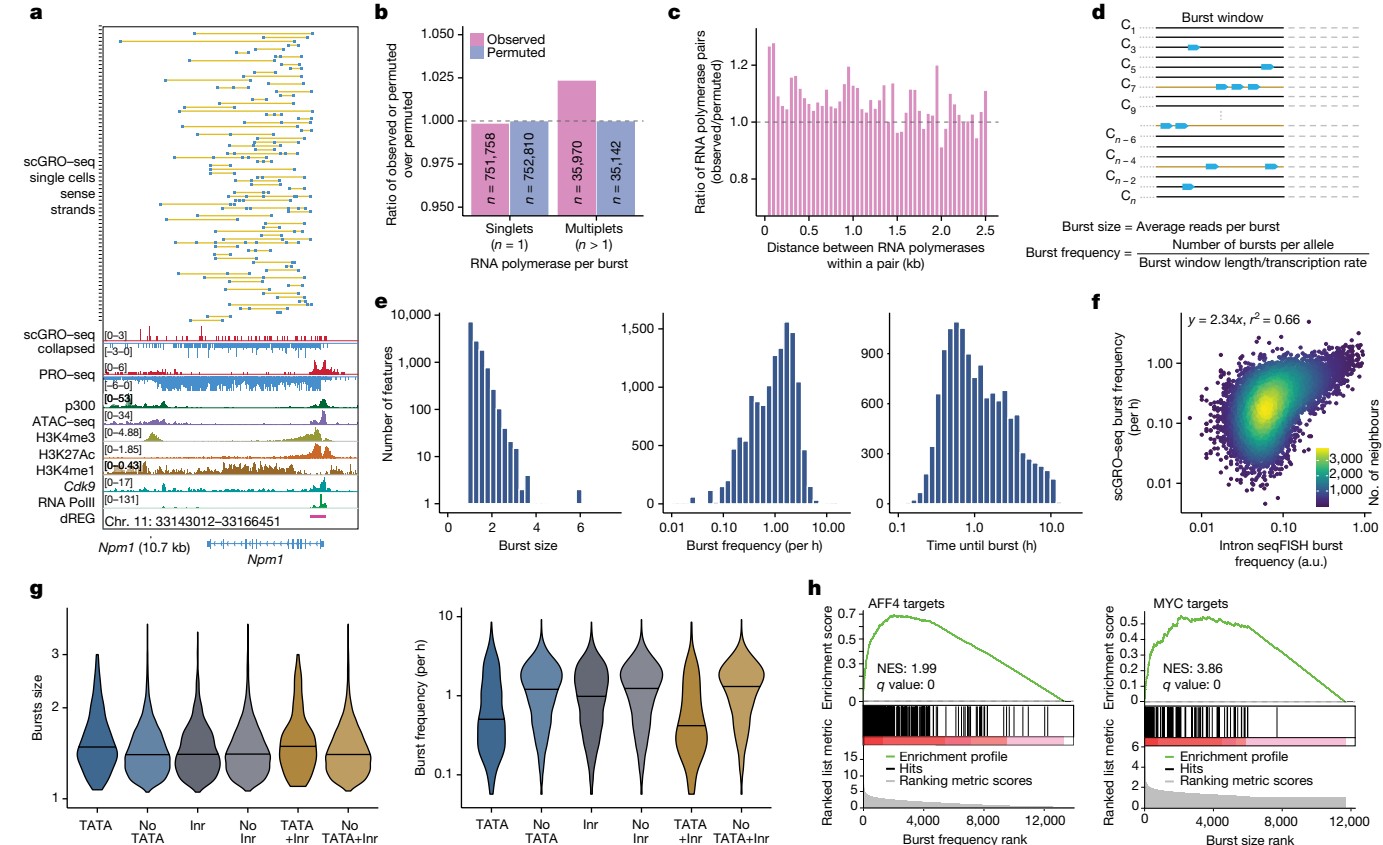

**Fig. 2 | Inference of transcription kinetics using scGRO–seq. a**, Single-cell view of multiplet RNA polymerase (blue dots) in *Npm1*. A yellow line connects RNA polymerases within the same cell. Randomly sampled 75 single cells containing more than one RNA polymerase are shown on the top, followed by the aggregate scGRO–seq, PRO–seq, chromatin marks and transcription-associated factors profiles. **b**, Ratio of observed or permuted burst sizes compared against the average burst sizes from 200 permutations. **c**, Ratio of the observed distance between consecutive RNA polymerases in the first 10 kb of gene bodies in individual cells against the permuted data. Distances up to 2.5 kb are shown in 50 bp bins. **d**, Illustration of the model used for direct inference of burst kinetics from scGRO–seq data. **e**, Histogram of burst size (left), burst frequency (middle) and duration until the next burst (1/burst frequency) (right) for genes that are at least 10 kb long (*n* = 13,142). **f**, Correlation of burst frequency of genes between scGRO–seq and intron seqFISH data. **g**, Effect of promoter elements in burst size greater than 1 (left) and burst frequency (right). Inr, the initiator motif. The centre line indicates the median of the distribution. **h**, Gene set enrichment analyses showing the role of transcription factors in determining burst frequency and burst size.

With confidence in the ability of scGRO–seq to discern bursting, we directly measured burst kinetics using scGRO–seq counts and their genomic positions. We estimated burst size as the average number of RNA polymerases per burst, whereas burst frequency was calculated as the number of bursts per allele per unit of time required for RNA polymerase to traverse through the burst window (Fig. 2d), corrected for capture efficiency (Methods). We considered genes longer than 11 kb (*n* = 13,564) and excluded 500 bp regions at either end that are known to harbour paused polymerases[21], thereby using the remaining 10 kb as the burst window. We assigned reads to a single allele based on previous evidence showing that alleles in mouse ES cells burst independently to generate monoallelic RNA[22]. With an average RNA PolII elongation rate of 2.5 kb min$^{-1}$ (ref. 23), using a 10 kb region limited the burst detection window to 4 min. This short burst window was consistent with bursts from one allele and aligned with previous reports[24]. We simulated kinetic measurements using synthetic data to validate the accuracy of the model and observed robust performance (Extended Data Fig. 8b). We then estimated the kinetic parameters of transcriptional bursts for expressed genes (Fig. 2e and Supplementary Table 1). Burst sizes ranged primarily between 1 and 4 RNA polymerases per burst, with a mean burst size of 1.23. The mean duration of approximately 2 h until the next burst obtained using scGRO–seq data matched the 2 h of the global nascent transcription oscillation cycle reported using intron seqFISH. Using the burst parameters estimated from scGRO–seq data, we again

tested our model by simulation and observed robust performance (Extended Data Fig. 8c). Burst frequency results from scGRO–seq data correlated well with intron seqFISH data (Fig. 2f), and the correlation was even stronger for genes with a higher burst frequency (Extended Data Fig. 8d). However, we observed a poor correlation between burst frequencies from scGRO–seq and scRNA–seq data, as well as between intron seqFISH and scRNA–seq data (Extended Data Fig. 8e). This finding highlights potential limitations in kinetic estimates derived from mature transcripts. In contrast to a previous report[18], we did not find an impact of gene length on kinetic estimates (Extended Data Fig. 8f). We further confirmed that the burst frequencies calculated from 10 kb and 5 kb burst windows showed strong agreement (Extended Data Fig. 8g), which indicated the reliability of burst kinetic calculations from scGRO–seq data.

Core promoter elements can modulate burst parameters[7,25]. We observed a significant variation in core promoter elements with burst kinetics (Fig. 2g). Specifically, genes with the TATA element exhibited a larger burst size than genes lacking it ($P = 4.6 \times 10^{-9}$), and the presence of the initiator sequence further increased the burst size ($P = 2.5 \times 10^{-13}$). The higher burst size but lower burst frequency of genes with TATA elements agreed with previous findings[26].

Transcription factors are also thought to regulate burst kinetics. Using a curated transcription factor binding database[27,28], we examined the effect of transcription factors on burst parameters. Gene set

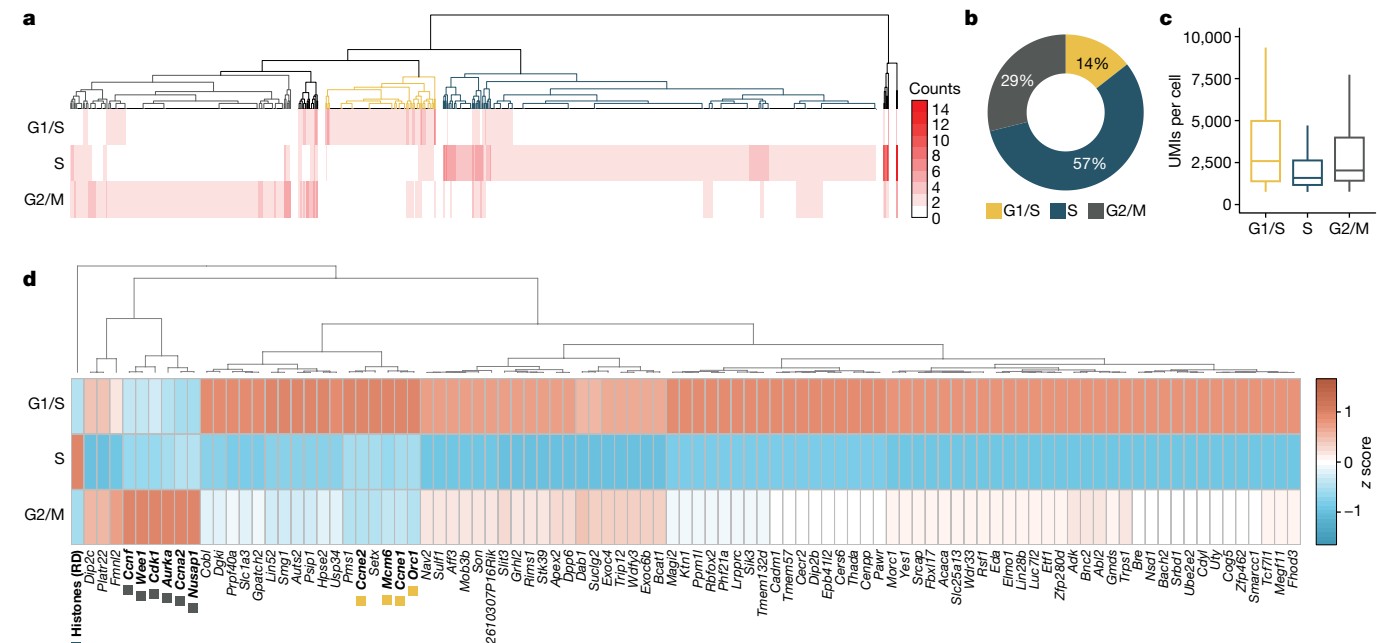

**Fig. 3 | Cell cycle inference by non-polyadenylated replication-dependent histone gene expression. a**, Heatmap of hierarchical clustering of single cells representing transcription of G1/S-specific, S-specific and G2/M-specific genes. The dendrogram colours represent cell clusters with cell-cycle-phase-specific gene transcription. **b**, Fraction of cells in the three primary clusters distinguished by transcription of G1/S-specific, S-specific and G2/M-specific genes. **c**, Distribution of scGRO–seq UMIs per cell in the three clusters of cells ($n$ = 122, 479 and 244 cells, respectively, from 10 independent batches) defined by cell-cycle-phase-specific gene transcription. The centre line indicates the median, the box represents the data between the first and third quantiles, the whiskers indicate the 1.5 interquartile range, and points outside the whiskers indicate outliers. **d**, Differentially expressed genes among the three clusters of cells defined by transcription of G1/S-specific, S-specific and G2/M-specific genes. The genes used to classify cells are denoted in bold and coloured boxes. Histones (RD) represent aggregate reads from replication-dependent histone genes.

enrichment analysis indicated that some transcription factors regulate burst size, whereas others regulate burst frequency (Supplementary Table 2). MYC and AFF4 are examples of each category. Genes bound by MYC had larger burst sizes, whereas AFF4 target genes were enriched for higher burst frequencies (Fig. 2h). Our observation supports a previous report whereby MYC increased the burst size by increasing the burst duration[29], and the association of the AFF4 transcription factor correlated with burst frequency[30]. Overall, we show that direct and comprehensive observation of transcription using scGRO–seq facilitates the study of transcription kinetics at the single-cell level.

## Cell cycle inference from histone genes

Investigating gene programs during cell cycle stages is essential for understanding biology and disease[31]. Polyadenylated RNA-dependent scRNA-seq methods rely on mature transcripts of cell cycle marker genes to determine the cell cycle state. However, the time required for mRNA processing, export and accumulation introduces a time lag. Except for a few total RNA-based single-cell methods[32,33], scRNA-seq fails to detect replication-dependent histone genes—the best characterized cell-cycle-phase-specific genes exclusively transcribed during the S phase[34]—owing to the lack of polyadenylation[35]. scGRO–seq enabled the detection of active transcription of replication-dependent histone genes in the histone locus body (Extended Data Fig. 9a) that could be used to classify cells in S phase. For G1/S and G2/M phase-specific genes, we used a set of transcriptionally characterized genes from a RNA velocity and deep-learning study of mouse ES cells[36]. Hierarchical clustering based on the expression of these three sets of cell-cycle-phase-specific genes revealed three significant clusters of individual cells (Fig. 3a).

Mouse ES cells have a short G1 phase and an extended S phase[37]. De novo classification of mouse ES cells based on the nascent transcription of these newly integrated marker genes recapitulated the lengths of cell cycle phases (Fig. 3b). Notably, cells in G1/S and G2/M phases exhibited higher transcription levels compared with cells in the S phase (Wilcoxon rank-sum test, $P = 6.3 \times 10^{-07}$ and $P = 1.2 \times 10^{-06}$, respectively) (Fig. 3c). We observed an approximately 40% decrease in total transcription when cells transition from the G1/S phase to the S phase, with a subsequent 20% increase after exiting the S phase to the G2/M phase. This observation indicates that transcription continues during DNA replication, albeit at a reduced level. The transition from the G2/M phase to the G1 phase is marked by an increase in transcription[38], which restores the transcription level observed during the G1 phase, thereby completing the cycle. An analysis of differentially expressed genes in cell cycle phases also revealed that certain genes restore transcription levels to those observed in the G1/S phase as they transitioned from the S phase to the G2/M phase, whereas others regained partial transcription (Fig. 3d and Supplementary Table 3). At the same time, some did not recover their transcription until exiting from G2/M to G1/S. By quantifying the active transcription of non-polyadenylated histone genes and a small subset of marker genes, scGRO–seq reveals a dynamic transcription program throughout the cell cycle.

## Co-transcription of interdependent genes

Co-expression of functionally related genes, as measured by accumulated mRNA, is widely reported[39]. However, assessing whether these genes are transcriptionally coordinated in steady-state has been challenging. By utilizing nascent transcription within the first 10 kb of the gene body, thereby limiting the co-transcription detection window to 4 min, we calculated pairwise Pearson correlation values between expressed genes (Fig. 4a). Gene pairs with a correlation coefficient greater than 0.1 and a $q$ value of less than 0.05, and an empirical FDR of less than 5% from 1,000 permutations, were considered co-transcribed (Supplementary Table 4). These stringent criteria controlled for

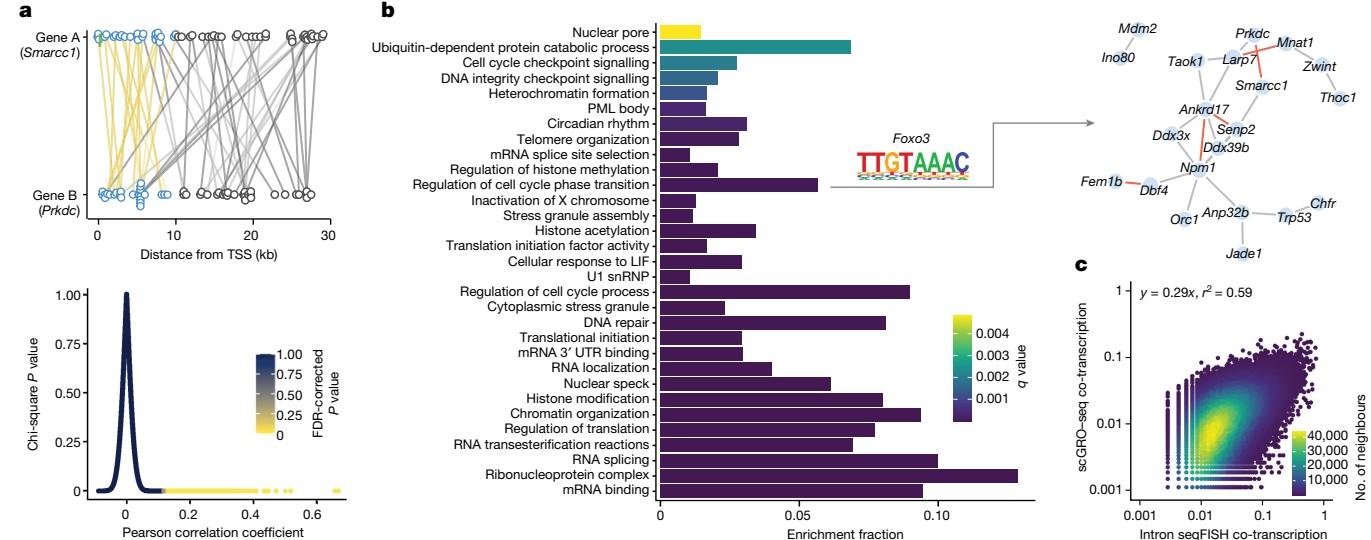

**Fig. 4 | Coordinated transcription of functionally related genes. a**, Top, a pair of co-transcribed genes. Reads within the first 10 kb of the gene pair (blue circle) expressed in the same cells are connected by a yellow line. Reads beyond the first 10 kb (grey circles and lines) were not used in the gene–gene correlation. Bottom, pair-wise Pearson correlation was calculated from a binarized genes by cells matrix. The relationship among the Pearson correlation coefficient, uncorrected chi-square $P$ value and the FDR-corrected $P$ value using the Benjamini–Hochberg correction method for pairwise gene–gene correlation. **b**, Gene ontology (GO) terms enriched in co-transcribed gene modules.

The transcription factor motif enriched in the promoters of genes associated with the GO term and the co-transcribed genes that contributed to the enrichment of the GO term is shown as an example on the right (red line indicating $\rho > 0.15$). A complete list of GO terms and the co-transcribed genes contributing to the enrichment of the GO terms is provided in Supplementary Table 5. **c**, Correlation of co-transcription of significantly co-transcribed gene pairs ($n = 164,380$) between scGRO–seq and intron seqFISH data. Axes represent the fraction of cells in which a gene pair is co-transcribed.

sampling biases and other confounding effects. We identified about 0.7% of the 112,807,710 gene pairs tested ($n = 800,888$) as significantly co-transcribed. We generated a graphical network from these significant pairs and identified 59 modules (genes per module > 10) of co-transcribed genes. This gene–gene transcriptional correlation probably reflects common temporal gene activation by a transcription factor or could reflect mechanistic coupling of transcription activation by clusters of genes separated across regions of chromosomes.

Conducting gene ontology analysis on these co-transcribed modules compared with all transcribed genes, we found enrichment of several related molecular functions, including cell cycle regulation, RNA splicing, translational control, DNA repair and circadian rhythm (Fig. 4b, Extended Data Fig. 9b and Supplementary Table 5). By scanning the promoters of co-transcribed genes, we discovered an enrichment of known transcription factor motifs, such as FOXO3 enriched in the promoters of co-transcribed genes associated with the 'regulation of cell-cycle phase transition' gene ontology term. A previous study[40] showed that FOXO3, in coordination with the DNA replication factor CDT1, is crucial in regulating cell cycle progression. We compared the co-transcription patterns of gene pairs obtained from scGRO–seq with those from intron seqFISH, and the results revealed concordant co-transcription profiles (Fig. 4c). This high-throughput and capability of scGRO–seq to directly examine transcriptional coordination between any gene pair or network of genes provides valuable insights into the functional organization of the genome.

## Enhancer–gene temporal coordination

Regulation of gene expression by distal regulatory elements is an area of broad interest. scGRO–seq captures transcripts from both genes and active enhancers, thereby enabling the measurement of co-activation in single cells. We analysed scGRO–seq reads within the first 10 kb of genes and at least 3 kb on each strand transcribing outwards around enhancers (Methods). We excluded 500 bp regions around the TSS of

genes and enhancers to avoid paused polymerase. We also included clusters of enhancers known as super-enhancers (SEs) that do not overlap with gene regions[41].

We used stringent criteria in permutation and correlation tests to identify enhancer–gene pairs that exhibit co-transcription (Methods). Out of 6,985,904 test pairs, 0.6% ($n = 44,361$) passed the threshold of the pairwise correlation coefficient, multiple hypothesis corrected chi-square $P$ value and empirical FDR from 1,000 permutations (Supplementary Table 6). We observed a significant enrichment (two-sample KS test, $P = 5.5 \times 10^{-09}$) of enhancer–gene co-transcription primarily within 200 kb of each other compared with uncorrelated pairs (Fig. 5a). SE–gene pairs were similarly enriched (two-sample KS test, $P = 1.3 \times 10^{-09}$) within 400 kb of each other (Extended Data Fig. 10a). When examining functionally related genes clustered together on the same chromosome[42], we found multiple enhancers correlated with each gene (Extended Data Fig. 10b), probably a further manifestation of cell cycle regulation.

We investigated a set of validated enhancers known to regulate pluripotency transcription factors[43–46]. We observed significant correlations between the transcription of *Sox2* and *Nanog* and their distal enhancers (Extended Data Fig. 10c). If enhancers and their target genes are temporally coupled and co-transcribed, we speculated that co-transcription of the pair could be even more prominent at finer temporal resolution. To test this idea, we divided enhancers and genes into 5 kb bins (representing a 2-min transcription window) and found that at least 1 enhancer bin correlated significantly with its target gene for all 4 genes (Fig. 5b). Notably, the correlated enhancer bin generally appeared further from its TSS than the gene bin, which implied that enhancer transcription may initiate before promoter transcription.

To test the enhancer–gene timing hypothesis, we examined a set of seven non-intronic mouse ES cell SEs validated by CRISPR perturbation[47]. CRISPR-mediated knockout of *Sall1* SE reduced *Sall1* expression by 92%, and we found a correlation between multiple enhancer bins and this gene (Fig. 5c). Overall, four out of seven SE–gene pairs

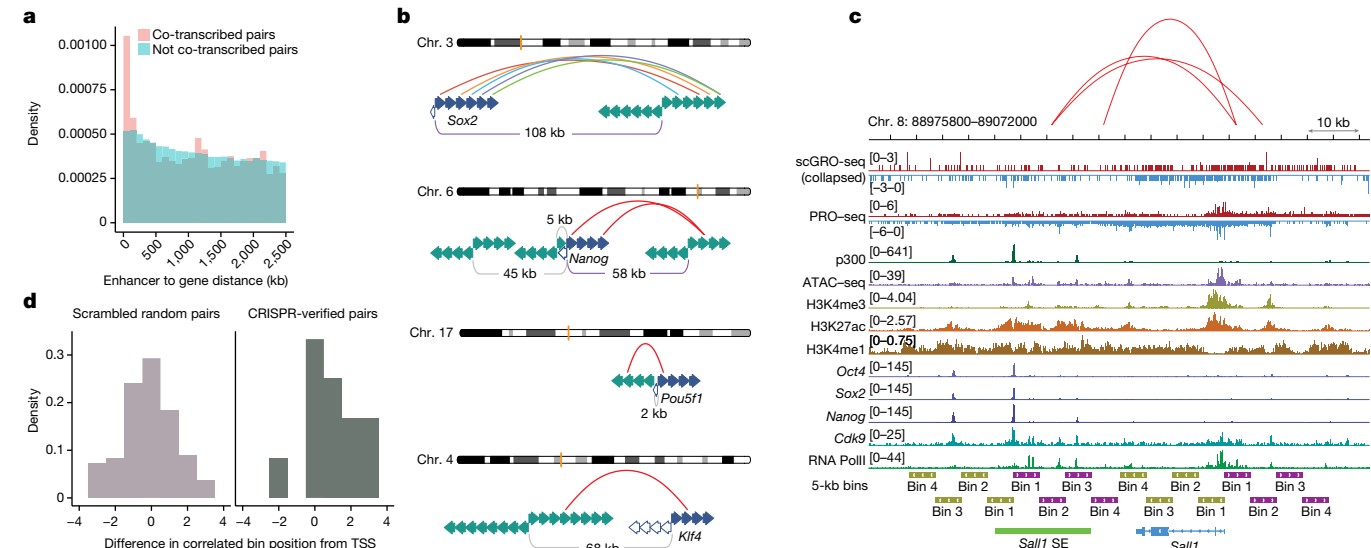

**Fig. 5 | Spatial and temporal coordination between genes and enhancers.**
**a**, Distance between correlated and non-correlated enhancer–gene pairs within 2.5 Mb of each other. **b**, Co-transcription between pluripotency genes (filled blue arrows indicate sense gene bins, open blue arrows indicate antisense gene bins) and their enhancers (represented by green arrows, and the arrow directions indicate sense and antisense directions). Correlated full-length enhancer–gene pairs (*Sox2* and *Nanog*) are shown with purple distance bars. For finer time resolution correlation, features are extended up to the end of the transcription signal and divided into 5 kb bins. Correlated bins are represented by a red arch, except for *Sox2* and its distal enhancer bins, which are shown in different colours for visual aid. **c**, Co-transcription between *Sall1* and its CRISPR-verified SE. Correlated SE–gene bins are denoted by arches. **d**, Summary of correlated bin positions in CRISPR-verified SE–gene pairs. Scrambled random pairs served as a control.

showed correlations of at least one bin. Notably, we observed that in most cases, enhancer transcription began earlier or around the same time as the transcription of their target genes (Fig. 5d). This temporal pattern could have mechanistic implications for enhancer–gene regulation. However, any conclusions will require a much deeper dataset. Nevertheless, our findings offer a glimpse into the temporal order in enhancer–gene transcription.

## Discussion

We developed scGRO–seq to enable the assessment of co-transcription and prediction of enhancer–gene regulatory networks in their native context. By reporting the activity of genes and distal regulatory elements—and therefore the functional consequences of transcriptional signals and networks—scGRO–seq is inherently multimodal for understanding transcription regulation in high detail. We illustrated these advantages by determining burst size and frequency for expressed genes, transcription dynamics during cell cycle phases and genome-wide gene–gene and enhancer–gene co-transcription detection. We restricted this study to mouse ES cells for comparison with large available datasets for validation.

The current scGRO–seq methodology has its limitations. The preservation of nuclear integrity, achieved through a low sarkosyl concentration, failed to promote the run-on of RNA polymerases in the pause complex, thereby limiting the detection of promoter–proximal paused polymerases. The read depth and cell numbers limited our analyses of burst kinetics and co-transcription of gene–gene and enhancer–gene pairs. Improved efficiency in future iterations will facilitate more precise evaluation of these phenomena.

scGRO–seq is also limited by the abundance of nascent RNA per cell at any given time, which is considerably lower than that of mature mRNA. Nascent RNA detection requires technology that does not depend on a polyadenylated terminus, which initially raised doubts about the feasibility of nascent RNA sequencing in single cells[48]. However, implementing highly efficient CuAAC has overcome this limitation, enabling the capture of approximately 10% of nascent RNA with the

current single-cell protocol. To streamline the process and to ensure compatibility with future automation, we optimized the biochemical steps by replacing multiple rounds of nascent RNA purification and nucleic acid ligation with click chemistry. Further adaptations, including high-throughput droplet encapsulation and enhanced capture efficiency, will extend the applicability of our scGRO–seq method in both research and clinical settings.

For clinical specimens, particularly for challenging tissues such as the brain and pancreas, which contain high levels of RNase, isolation of nuclei is preferred over intact cells. Single-cell methods such as sNuc-seq[49] profile polyadenylated RNA inside the nucleus of such tissues, but paint an incomplete view of single-cell gene expression. By contrast, the entire scGRO–seq substrate is present inside the nucleus. Furthermore, the compatibility of CuAAC-based nascent RNA sequencing methods with bulk low-input samples and single cells makes them desirable methods for clinical investigations. The adaptability and efficiency of scGRO–seq introduce new avenues for investigating transcriptional dynamics and regulatory mechanisms across diverse biological contexts, enriching our understanding of gene expression regulation and its ramifications in physiological and pathological conditions.

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

## Methods

### scGRO–seq conceptualization

Capturing nascent RNA with sufficient efficiency from single cells for meaningful analysis was deemed challenging. However, recognizing the potential insights into transcription mechanisms that single-cell nascent RNA sequencing could offer, we set out to develop a single-cell version of the GRO–seq method a decade after its use in cell populations. Our efforts were met with two significant challenges: selectively capturing a small fraction of nascent RNA among various RNA species within a cell and accurately distinguishing nascent RNAs from individual cells.

The primary limitation we encountered was capture efficiency. The quantity of nascent RNA from transcribing RNA polymerases in an individual cell, mainly due to the intermittent nature of transcription with short bursts and long latency periods, is significantly lower than the mRNA copies that accumulate over time. Traditional nascent RNA capture methods yield only a meagre number of nascent RNAs from single cells. Miniaturizing GRO–seq using strategies derived from scRNA-seq was not feasible because nascent RNA lacks the consensus polyadenylation sequence used in RNA-seq. Instead, GRO–seq and related methods selectively label nascent RNA in bulk cells using modified nucleotides and use single-stranded RNA–RNA ligation with PCR handles on both ends. This ligation process proved unsuitable for scGRO–seq owing to its low efficiency and the need for nascent RNA purification before ligation, which risks depleting the already scarce nascent RNA from single cells.

To overcome these challenges, we devised a strategy that involved labelling nascent RNA in cells and attaching single-cell barcodes to the labelled nascent RNA without requiring purification from other cellular RNA. After exploring several approaches without success, we turned to click chemistry, specifically CuAAC. We speculated that by sourcing or synthesizing CuAAC-compatible chain-terminating nucleotide triphosphate analogues and performing nuclear run-on with the modified nucleotides to selectively label nascent RNA, we could label nascent RNA from individual cells with 5′-AzScBc DNA with a PCR handle. Then, we could pool the barcoded nascent RNA from multiple cells for selective RT in the presence of a TSO and subsequent PCR amplification for sequencing.

To successfully implement this strategy, we identified three important biochemical hurdles to address. First, we needed to demonstrate the ability of native RNA polymerase to incorporate 3′-(O-propargyl)-NTPs during nuclear run-on reactions. Second, preserving the intactness of nuclei during the run-on reaction was essential to enable the separation of individual nuclei for single-cell barcoding. Finally, we had to confirm the ability of reverse transcriptase to traverse the triazole ring junction formed during CuAAC. Successful resolution of the first and third hurdles would pave the way for CuAAC-based nascent RNA sequencing in cell populations, whereas overcoming the second hurdle would establish the foundation for scGRO–seq.

### Development of AGTuC

To develop a nascent RNA tagging method suitable for capturing a small fraction of RNA from single cells, we initiated our approach by focusing on a cell-population-based strategy. We aimed to develop an enhanced nascent RNA tagging method that optimally integrates selective labelling and single-cell barcode tagging, bypassing the need for RNA purification. Among the tested methods, we identified click chemistry as the most suitable option because of its high selectivity, efficiency, robustness in diverse experimental conditions, cost-effectiveness and speed. Our goal was to selectively label nascent RNA through a nuclear run-on reaction, conjugate a single-stranded DNA PCR handle (that can accommodate a single-cell barcode for future use in single-cell analysis), reverse transcribe the RNA–DNA conjugate and prepare a NGS library.

To achieve single-nucleotide resolution of transcribing polymerases and efficient RT, we identified two click-chemistry-compatible, chain-terminating nucleotides with a relatively small functional group: 3′-(O-propargyl)-ATP and 3′-azido-3′-dATP (Extended Data Fig. 1a). Nascent RNA labelled with 3′-(O-propargyl)-NTPs forms a 1,4-disubstituted 1,2,3-triazole junction with azide-labelled DNA through CuAAC, as shown in Click-Code-Seq[50], whereas nascent RNA labelled with 3′-azido-3′-dNTPs forms a slightly bulkier junction with dibenzocyclooctyne labelled DNA through strain-promoted alkyne-azide cycloadditions (Extended Data Fig. 1b). Nuclear run-on with 3′-(O-propargyl)-ATP and CuAAC showed superior efficiency compared with 3′-azido-3′-dATP and strain-promoted alkyne-azide cycloadditions (Extended Data Fig. 1c).

To convert the clicked RNA–DNA conjugate to cDNA, we tested eight different reverse transcriptase enzymes, varied the temperature and duration of RT and evaluated three TSOs (Extended Data Fig. 1d–f, some results not shown). Our optimized method, which we AGTuC, was then performed in 5 million mouse ES cell nuclei. AGTuC nascent RNA profiles closely resembled PRO–seq profiles (Extended Data Fig. 2a) and exhibited strong correlations at both gene and enhancer levels (Extended Data Fig. 2b,c). Notably, the AGTuC library protocol involved significantly fewer steps than PRO–seq and could be completed in a single day (Extended Data Fig. 2d). AGTuC is a simpler, faster and cheaper alternative to GRO–seq and PRO–seq for nascent RNA sequencing from cell populations.

### Development of inAGTuC

To adapt CuAAC-mediated nascent RNA sequencing to single cells, we explored the feasibility of performing AGTuC in single cells. Implementing AGTuC at the single-cell level presented challenges as the nuclear run-on reaction with 0.5% sarkosyl disrupts the nuclear membrane before cell barcodes could be attached during the post-run-on CuAAC step, which leads to unintended mixing of nascent RNA from different cells. One potential solution was to perform AGTuC in single tubes, which would prevent nascent RNA mixing. However, this approach requires RNA purification after the run-on reaction, but purification results in further depletion of exceedingly low amounts of nascent RNA in single cells. Alternatively, omitting RNA purification would lead to an abundance of 3′-(O-propargyl)-NTPs supplied in excess during the run-on reaction, which could outcompete 5′-AzScBc DNA during CuAAC.

To address this challenge, we developed inAGTuC, a new strategy that enables labelling nascent RNA with 3′-(O-propargyl)-NTPs while preserving nuclear integrity. This approach overcomes the issues associated with nascent RNA mixing before single-cell barcoding. We proposed that performing the run-on reaction without disrupting the nuclear membrane would facilitate the easy removal of excess nucleotides through a few centrifugation and resuspension steps while retaining propargyl-labelled nascent RNA within the nuclei. This approach would produce clean nuclei with labelled nascent RNA, free from excess reactive nucleotides, which could be compartmentalized with 5′-AzScBc DNA for CuAAC. We could minimize further RNA loss by pooling and processing the single-cell-barcoded nascent RNA from multiple cells.

To achieve an efficient run-on reaction, PRO–seq and AGTuC disrupt the polymerase complex with 0.5% sarkosyl detergent, of which nuclear membrane lysis is collateral damage. We sought to identify the lowest sarkosyl concentration that maintains nuclear membrane integrity while maximizing run-on efficiency and found that a 20× reduction in sarkosyl concentration preserved nuclear intactness, with only a 20% reduction in run-on efficiency (Extended Data Fig. 3a,b). To maximize the capture efficiency of nascent RNA, we optimized the molecular crowding effect of PEG 8000 and the ratio of Cu(I) to the CuAAC accelerating ligand BTTAA (Extended Data Fig. 3c). Although a low sarkosyl concentration preserves nuclear integrity, it also retains the RNA

polymerase complex intact, thereby shielding the propargyl-labelled 3′ end of nascent RNA from reacting with 5′-AzScBc DNA. We investigated nascent RNA release from the RNA polymerase complex using common denaturants and found that 6 M urea and TRIzol was efficient (Extended Data Fig. 3d). However, the denaturant in TRIzol hindered CuAAC reaction (Extended Data Fig. 3e). Notably, urea also offered the added benefit of retaining the RNA–DNA conjugate in the aqueous phase during TRIzol clean-up to remove PEG 8000 from the CuAAC reaction (Extended Data Fig. 3f). For reaction clean-up, we assessed various methods, finding cellulose membrane to be effective in removing CuAAC reagents (Extended Data Fig. 3g), whereas silica matrix columns performed well in retaining RNA and ssDNA (Extended Data Fig. 3h). Subsequently, we evaluated DNA polymerase for library preparation and DNA size-selection methods (Extended Data Fig. 3i,j).

Considering the goal of working with single cells, we performed inAGTuC with cell numbers between 5 million used in AGTuC and 1 cell planned for scGRO–seq. Specifically, we placed 100 to 1,000 intact nuclei in each well of a 96-well plate containing urea. Nascent RNA in each well was barcoded with a unique 5′-AzScBc DNA by CuAAC and pooled from the 96 wells, and a sequencing library was prepared as in AGTuC. The inAGTuC libraries exhibited similar profiles in gene bodies compared with PRO–seq and AGTuC. However, they could not capture the paused peaks at the 5′ end of genes and enhancers (Extended Data Fig. 4a–c). This observation is consistent with the need for a higher sarkosyl concentration for efficient run-on of paused polymerase complexes. The four inAGTuC libraries correlated well with each other (Extended Data Fig. 4d), with the potential to discover more insights with deeper sequencing (Extended Data Fig. 4e,f). Despite only partially capturing nascent RNA from a paused complex, the inAGTuC libraries correlated well with those from AGTuC and PRO–seq (Extended Data Fig. 4g).

To systematically characterize the compatibility of inAGTuC with even fewer cells, we prepared four inAGTuC libraries in a 96-well plate, with 12 c.p.w., 120 c.p.w. and 1,200 c.p.w., which is roughly equivalent to 1,000, 10,000 and 100,000 nuclei, respectively. We also included a 1,200 c.p.w. plate, omitting Cu(I) as a negative control. Despite lower coverage, the inAGTuC library with 12 c.p.w. (total of about 1,000 cells) successfully captured the overall nascent RNA profile. It exhibited a good correlation with 120 c.p.w. (total of about 10,000 cells) and 1,200 c.p.w. (total of around 100,000 cells) (Extended Data Fig. 5a–c).

### 3′-(*O*-propargyl)-nucleotide synthesis
For this study, several CuAAC-compatible nucleotide analogues modified with azide or alkyne functionalities were evaluated. Ultimately, 3′-(*O*-propargyl)-NTPs were selected for three main reasons: (1) these analogues lack 3′ hydroxyl groups, making them chain-terminating and enabling single-nucleotide resolution of the 3′ end of nascent RNA; (2) the CuAAC reaction produces a compact junction due to the presence of a single carbon bond between the sugar group of the nucleotide and the propargyl group at the 3′ end position; and (3) they are relatively cost-effective compared with biotin-modified nucleotides commonly used in PRO–seq.

3′-(*O*-Propargyl)-ATP (NU-945) was offered by Jena Biosciences. To complete the set, custom synthesis requests were made for 3′-(*O*-propargyl)-CTP (NU-947), 3′-(*O*-propargyl)-GTP (NU-946) and 3′-(*O*-propargyl)-UTP (NU-948), all of which are now available for purchase from Jena Biosciences.

### Single-cell barcoded DNA adaptors
During scGRO–seq development, 3 sets of 96 5′-AzScBc DNA were synthesized by GeneLink. Each design encompassed four components: a 5′ azide positioned at the 5′ terminus, a 10–12 nucleotide sequence for the single-cell barcode, a 4–6 nucleotide sequence for the UMI and a PCR handle. The 5′ azide modification was obtained following a previously described method[51]. Specifically, an oligonucleotide containing

5′ iodo-dT was synthesized through solid-support phosphoramidite oligonucleotide synthesis, and subsequent replacement of the iodo group with an azide group was achieved through a reaction with sodium azide at 60 °C for 1 h. The sequences of three different 5′-AzScBc DNA are available in Supplementary Table 7.

The hairpin structure of the 86-nucleotide 5′-AzScBc DNA (Supplementary Fig. 3a) is formed through self-folding. The RT process is initiated using the 3′ end of the oligonucleotide, which serves as a built-in primer. This design ensures a 1:1 stoichiometry between the PCR handle and the RT primer, minimizing mispriming and nonspecific amplification during RT. The folded hairpin structure also generates a restriction site for the EagI enzyme, which is digested before PCR amplification.

Undesired extension by reverse transcriptase is effectively prevented by a three-carbon spacer at the 3′ end of the 43-nucleotide 5′-AzScBc DNA[52]. This version of the azide adaptor harbours a 5-nucleotide ACAGG sequence after the azide-dT at its 5′ end (Supplementary Fig. 3b). During RT, the extension of primers annealing to unclicked 5′-AzScBc, the addition of non-templated CCC and the incorporation of TSO results in undesired cDNA that are preferred substrates for PCR amplification. If unaddressed, these amplicons can overwhelm the sequencing library. The ACAGG sequence plays a crucial role in depleting these PCR amplicons.

A previously described method named DASH uses recombinant Cas9 protein and gRNA complex to digest and deplete undesired dsDNA[53]. The ACAGG sequence is necessary to generate a gRNA target sequence in the undesired PCR amplicons (underlined sequence). In PCR amplicons formed between nascent RNA and 5′-AzScBc DNA, the complementation of gRNA is interrupted by the presence of a nascent RNA sequence, which makes the desired products incompatible with DASH. AGG serves as the protospacer adjacent motif.

### Cell line
The V6.5 mouse ES cells used in this study were established by the Jaenisch Laboratory (Whitehead Institute, Massachusetts Institute of Technology) from the inner cell mass of a 3.5-day-old mouse embryo from a C57BL/6(F) × 129/sv(M) cross.

### Cell culture
Mouse ES cells were cultured in Dulbecco's modified Eagle medium (Gibco, 11995), plus 10% fetal bovine serum (HyClone, SH30070.03), supplemented with 1× penicillin–streptomycin (Gibco, 15140), 1× non-essential amino acids (Gibco, 1140), 1× L-glutamine (Gibco, 25030), 1× β-mercaptoethanol (Sigma, M6250) and 1,000 U ml$^{-1}$ leukaemia inhibitory factor (Sigma, ESG1107) on tissue-culture-treated 10 cm plates (Corning, CLS430167) pre-coated with 0.2% gelatin (Sigma, G1890) prepared in PBS (Fisher, MT21031CV). Cells were grown at 37 °C and 5% $CO_2$ and passed with HEPES buffered saline solution (Lonza, CC-5024) and 0.25% trypsin-EDTA (Gibco, 25200) when 70% confluency was reached (every 2 days).

### Sample preparation
Tissue culture cells were prepared for nuclear run-on reaction by either nuclei isolation or cell permeabilization as described below. All centrifugation steps were performed at 1,000$g$ for 5 min. Cells were collected by removing the tissue culture medium, rinsing with PBS and placing the plates on ice. Cells were scraped while still on ice. The cells were collected into a 15 ml conical tube and centrifuged at 1,000$g$ for 5 min.

For nuclei isolation, the pellet was resuspended in ice-cold douncing buffer (10 mM Tris-Cl pH 7.4, 300 mM sucrose, 3 mM $CaCl_2$, 2 mM $MgCl_2$, 0.1% Triton X-100, 0.5 mM DTT, 0.1× Halt protease inhibitor and 0.02 U µl$^{-1}$ RNase inhibitor) and transferred to a 7 ml dounce homogenizer (Wheaton, 357542). After incubation on ice for 5 min, the cells were dounced 25 times with a tight pestle, transferred back to the 15 ml conical tube and centrifuged to pellet the nuclei. The pellet was washed twice in a douncing buffer.

For cell permeabilization, the pellet was resuspended in ice-cold permeabilization buffer (10 mM Tris-Cl pH 7.4, 300 mM sucrose, 10 mM KCl, 5 mM $MgCl_2$, 1 mM EGTA, 0.05% Tween-20, 0.1% NP-40, 0.5 mM DTT, 0.1× Halt protease inhibitor and 0.02 U $\mu l^{-1}$ RNase inhibitor). After incubation on ice for 5 min, the cells were centrifuged to pellet the nuclei. The pellet was washed twice in the permeabilization buffer.

The washed pellet was resuspended in storage buffer (10 mM Tris-Cl pH 8.0, 5% glycerol, 5 mM $MgCl_2$, 0.1 mM EDTA, 5 mM DTT, 1× Halt protease inhibitor and 0.2 U $\mu l^{-1}$ RNase inhibitor) at a concentration of $5 \times 10^6$ nuclei per 50 $\mu l$ of storage buffer, flash-frozen in liquid nitrogen and stored at −80 °C. The nuclei and permeabilized cells in the storage buffer can be stored for up to 5 years at −80 °C, making them readily available for nuclear run-on experiments.

### Nuclear run-on with 3′-(O-propargyl)-nucleotides
A volume of 50 $\mu l$ of 2× nuclear run-on buffer (20 mM Tris-Cl pH 8.0, 10 mM $MgCl_2$, 400 mM KCl, 50 $\mu M$ 3′-(O-propargyl)-ATP, 50 $\mu M$ 3′-(O-propargyl)-CTP, 50 $\mu M$ 3′-(O-propargyl)-GTP, 50 $\mu M$ 3′-(O-propargyl)-UTP, 0.05% Sarkosyl, 1 mM DTT, 2× Halt protease inhibitor and 0.4 U $\mu l^{-1}$ RNase inhibitor) was prepared per sample and heated to 37 °C. Once thawed from −80 °C, permeabilized cells or nuclei were added to the heated tube containing nuclear run-on buffer and incubated for 5 min at 37 °C with gentle tapping at the incubation midpoint. Permeabilized cells or nuclei were centrifuged at 500g for 2 min at 4 °C, and the supernatant was aspirated off. The pellet was washed 3 times in 150 $\mu l$ resuspension buffer (5 mM Tris-Cl pH 8.0, 2.5% glycerol, 2.5 mM MgAc$_2$, 0.05 mM EDTA, 1.25 mM $MgCl_2$, 60 mM KCl, 3 mM DTT, 0.2× Halt protease inhibitor and 0.2 U $\mu l^{-1}$ RNase inhibitor). After the final wash, the permeabilized cells or nuclei were resuspended in a 2 ml resuspension buffer and passed through a 35 $\mu m$ nylon mesh (Falcon, 352235).

### Single-cell sorting and nuclei sorting
For single-cell and nuclei sorting, 96-well plates with 2.5 $\mu l$ 8 M urea were prepared using a multichannel or 96-well pipettor (Avidien MicroPro 300, 30835029). Single cell and nuclei populations characterized by forward and side scattering were sorted by FACS into the 96-well plate containing urea. The sorted plates can be used in CuAAC directly or sealed with aluminium foil or a plastic seal and stored at −80 °C.

### CuAAC
A 96-well plate containing 5′-AzScBc DNA with a unique cell barcode in each well previously synthesized and aliquoted was thawed from −80 °C. Sodium ascorbate, PEG 8000, $CuSO_4$ and accelerating ligand BTTAA were prepared and dispensed into each well of the 96-well plate containing 5′-AzScBc DNA. The CuAAC reaction mix was dispensed into individual wells containing single cells in urea using a multichannel or 96-well pipette. The final concentration of CuAAC reaction in each well was 30 nM 5′-AzScBc DNA, 800 mM sodium ascorbate, 15% PEG 8000, 1 mM $CuSO_4$, 5 mM BTTAA and 2.66 M urea in a 7.5 $\mu l$ volume. The 96-well plates were sealed, vortexed for 10 s in an orbital vortexer and centrifuged for 1 min at 500g before incubation for 2 h at 50 °C.

After incubation, the CuAAC reaction was quenched with 5 mM EDTA and pooled from 96 wells into a 1.5 ml Eppendorf tube. PEG 8000 was removed using TRIzol. The remaining CuAAC reagents (sodium ascorbate, $CuSO_4$ and BTTAA) were removed with a centrifugal filter with 3 kDa cellulose membrane (Amicon, 2020-04). The purified RNA was fragmented with 10 mM $ZnCl_2$ for 5 min at 65 °C.

### RT through the triazole link and pre-amplification
RT of the clicked RNA–DNA conjugate was performed with highly processive Moloney murine leukaemia virus (M-MuLV) reverse transcriptase lacking RNase H activity but capable of RNA-dependent and DNA-dependent polymerase activity, non-templated addition and template switching (Thermo Fisher, EP0751). RT reaction (1× RT buffer, 0.5 mM dNTPs, 0.8 U $\mu l^{-1}$ RNase inhibitor, 16% PEG 8000, 1 $\mu M$ RT primer (except for hairpin-forming 5′-AzScBc DNA), and 1 $\mu m$ TSO) was incubated with the RNA–DNA conjugate for 2 h at 50 °C. The cDNA was size-selected in 10% denaturing PAGE away from the unclicked 5′-AzScBc DNA and empty cDNA formed between the 5′-AzScBc DNA and TSO.

The purified cDNA was PCR amplified for 6 cycles to generate dsDNA with NEBNext Ultra II Q5 High-Fidelity 2× master mix (NEB, M0544) and 0.5 $\mu M$ PCR primers with unique dual index using the PCR cycles presented in Supplementary Table 8.

### Removal of empty adaptors using DASH
The dsDNA from the pre-amplification of cDNA was subjected to DASH to remove the undesired amplicons formed by RT of unclicked 5′-AzScBc DNA and TSO, as described above. Cas9–gRNA complex (6.6 $\mu M$ Streptococcus pyogenes Cas9 nuclease (NEB, M0386T), 20 $\mu M$ gRNA, 1× NEBuffer r3.1 and nuclease-free duplex buffer (IDT, 11-05-01-04)) was prepared by incubation for 15 min at 25 °C. The incubated complex was added to the cleaned PCR reaction and incubated for 1 h at 37 °C.

### PCR amplification and NGS
The DASHed library was PCR amplified with NEBNext Ultra II Q5 High-Fidelity 2× master mix (NEB, M0544) and 0.5 $\mu M$ PCR primers with a unique dual index using the two-step PCR cycles presented in Supplementary Table 9.

The NGS library was sequenced on Illumina NovaSeq SP100 flow cells with 64 nucleotides forward read, 43 nucleotides reverse read, 8 nucleotides index 1 and 8 nucleotides index 2.

### Alignment and pre-processing
Adaptor sequences were removed from paired-end fastq files using Cutadapt[54]. In brief, the read 1 sequence CCCCTGTCTCTTATACACAT and the read 2 sequence AGATCGGAAGAGCGTCGTGT were trimmed with a maximum error rate of 0.15, requiring a minimum overlap of 12 nucleotides between the read and adapter. The resulting adapter-trimmed reads were demultiplexed using Flexbar[55]. Cell barcodes and UMIs were extracted from the 5′ end of read 1, applying a barcode error rate of 0.15 and retaining reads of at least 14 nucleotides in length. The adapter-clipped and demultiplexed reads were first mapped to the mouse ribosomal genome using bowtie2 (ref. 56) in --very-sensitive mode. The reads unmapped to the ribosomal genome were mapped to the mouse genome (mm10 build) in --very-sensitive mode. After mapping, duplicate reads were identified and removed utilizing UMI and mapping coordinates with UMI-tools[57].

### Filtering experimental batches and cells
The scGRO−seq batches with $r^2$ values of at least 0.6 against at least 60% of all batches were selected for further analysis. Cells were required to contain a minimum of 1,000 UMIs and 750 features for further analysis. Our study involved 17 batches of scGRO−seq experiments across 39 96-well plates, encompassing a total of 3,744 cells. Of these, 36 plates (each containing a minimum of 24 high-quality cells) and 2,635 cells met the threshold.

### Estimation of capture efficiency
The average capture efficiency of scGRO−seq was estimated to be approximately 10%. We used data from the intron seqFISH study[17], which quantified the abundance of 34 introns by single-molecule fluorescent in-situ hybridization (smFISH). Based on the slope of the line of best fit between data from smFISH and intron seqFISH, the detection efficiency of intron seqFISH was estimated to be 44%. When scGRO−seq was compared with intron seqFISH, the detection efficiency of scGRO−seq was 26% of intron seqFISH. Based on these two detection efficiencies, the estimated capture efficiency of scGRO−seq is about

10% (26% of 44% is approximately 10%). This estimate is based on the 8 min of median time required for intron to be spliced out once it is transcribed, which ranges from 5 to 10 min according to several studies using diverse methods[58–64]. Thus, the capture efficiency of 10% is an average approximation and can vary among cells and batches.

## Enhancer annotation
Active transcription regulatory elements (TREs) in mouse ES cells were identified with PRO–seq data using dREG[65]. Further filtering of the dREG results, carried out to eliminate TREs within or proximal to 1,500 bp of the RefSeq annotated genes (n = 23,980), identified 68,299 high-confidence TREs. The remaining TREs within 500 bp of each other were combined, which resulted in the final list of 12,542 enhancers. To capture nascent RNA derived from elongating RNA polymerases at these enhancers, the TREs were extended at least 1500 bp from the TSS in both directions. The overlapping enhancers were stitched together after extension.

## Transcription unit calling
groHMM (https://www.bioconductor.org/packages/release/bioc/vignettes/groHMM/inst/doc/groHMM.pdf) was used to call de novo transcription unit on PRO–seq data. All combinations of tuning parameters (−50, −100, −200, and −400 for LP and 5, 10, and 15 for UTS) were tested. LP represents the 'log-transformed transition probability of switching from transcribed state to non-transcribed state', and UTS represents 'the variance of the emission probability for reads in the non-transcribed state'. In our test, −50 LP and 10 UTS performed best for optimal transcription unit calling.

## Evidence of bursting
Transcriptional bursting was examined de novo using scGRO–seq data by measuring two parameters: the multiplicity of RNA polymerases and the distance between the RNA polymerases. The bursting model suggests that transcription occurs in short bursts punctuated by long silent periods, which results in on and off states. The alternative model is the relatively uniform transcription initiation by primarily solitary RNA polymerase. We expected two observations under the bursting model.

First, we expected a higher incidence of more than one RNA polymerase per burst and a concurrent depletion of single RNA polymerases. To test the evidence of bursting, we selected genes longer than 11 kb (n = 13,564) and trimmed 0.5 kb regions from the 5′ and 3′ ends of the gene that are known to harbour paused polymerases. With an average transcription rate of 2.5 kb min[-1], the remaining 10 kb region resulted in an observation window of 4 min. Based on the evidence of monoallelic transcription described in the main text and a short observation window of 4 min, we assigned all signals for a gene in individual cells to one allele. We quantified the observed incidence of zero, one (singlets) and more than one RNA polymerase (multiplets) per allele. The majority of alleles had zero polymerase. To calculate the expected incidences of RNA polymerases under the non-bursting model, we permuted the cell identity of scGRO–seq reads 200 times without changing the read positions. The permutation maintains the number of UMIs per cell, breaks the bursting-mediated association between RNA polymerases, and mimics the RNA polymerases distribution under the non-bursting model. We quantified the permuted incidences of zero, singlets and multiplets.

Second, if more than one RNA polymerase is observed in the burst window, either due to transcriptional bursting or random chance, we expected the transcription bursting model would result in more closely spaced molecules than expected by the random chance. We took all multiplets in observed or permuted data and calculated the distance between RNA polymerase molecules within each pair. We binned the distances in 50 bp bins and calculated the ratio of RNA polymerase pairs between the observed and permuted data.

## Burst kinetics
Genes over 11 kb (n = 13,564) were selected for studying transcriptional bursting kinetics, and 500 nucleotide regions at both ends known to harbour paused polymerases were truncated. In cases in which genes exceeded 10 kb after trimming, they were shortened to 10 kb starting from the initiation site of the gene. With an average transcription rate of 2.5 kb min[-1], this 10 kb burst window served an average burst duration of 4 min. The calculation of burst size and burst frequency proceeded as described below.

**Burst size.** For each gene, the number of cells with at least one read within the 10 kb burst window (number of bursts) was identified, and then the average UMIs per burst was computed. If a consistent single read per burst was observed, the burst size of that gene was set to 1. However, if the average burst size was 1.2, the residual burst above 1 indicated a higher burst size. Accounting for the 10% capture efficiency, wherein the likelihood of capturing paired reads within a burst window is 1%, the residual burst was proportionally adjusted by the capture efficiency. The equation for the burst size is shown in Supplementary Fig. 4 (top).

**Burst frequency.** For each gene, the burst frequency was determined as the number of bursts per allele (two alleles in autosomal and one in sex chromosomes) per transcription time. The transcription time was calculated as the duration needed to traverse the 10 kb burst window with a uniform transcription rate of 2.5 kb min[-1], translating to 4 min. The calculated burst frequency was normalized by the capture efficiency, taking the burst size into account. Although burst events with a larger burst size, like ten, would be consistently detected even with 10% capture efficiency, normalization was applied for cases in which a burst size like four would result in a 60% false negative rate, which indicated a non-existent burst despite active bursting. Thus, burst frequency normalization was scaled by burst size to ensure accurate quantification. The equation for the burst frequency is shown in Supplementary Fig. 4 (bottom).

Genes with core promoter elements like TATA and Initiator sequences were retrieved from the Eukaryotic Promoter Database (http://epd.vital-it.ch)[66]. Genes containing a pause button, a sequence associated with promoter–proximal paused RNA polymerase, were recovered from the CoPRO dataset[67].

## Simulation of idealized burst kinetics
We simulated read counts for populations of single cells to evaluate the performance of our estimators for burst rate and size. In the first simulation, we randomly generated the true burst size ($T_{size}$) for all human genes from a normal distribution (mean = 2, standard deviation = 3). Similarly, we generated true burst rates ($T_{rate}$) for all human genes from a normal distribution (mean = 1, standard deviation = 1). $T_{size}$ less than 1 was corrected to 1, and $T_{rate}$ less than 0.1 burst per hour was corrected to 0.1. These parameters were used to simulate UMIs per gene per cell as follows:
1. For each cell and each gene, a sample from a Poisson distribution with rate parameter $\lambda = T_{rate}$.
2. Scale the sampled burst by $T_{size}$ and round to the nearest integer.
3. After generating molecule counts for all genes and all cells, randomly subsample to a specified level (for example, 10% sampling efficiency) without replacement.

In the second simulation, $T_{size}$ and $T_{rate}$ were taken from our genome-wide estimates described in Fig. 2, and UMIs per gene per cell were similarly generated. Simulations were performed ten times to ensure consistent results.

## Cell cycle analysis
Three sets of transcriptionally characterized genes were used to characterize the cell cycle phase in individual cells. Transcription of

68 replication-dependent histone genes on chromosome 3, chromosome 6, chromosome 11 and chromosome 13 were used to determine the S phase collectively. Transcription of four genes (*Orc1*, *Ccne1*, *Ccne2* and *Mcm6*) were used to assign G1/S phase, and six genes (*Wee1*, *Cdk1*, *Ccnf*, *Nusap1*, *Aurka* and *Ccna2*) were used to assign G2/M phase. Cells with more than a read in one of the genes or reads in more than one gene were hierarchically clustered, which revealed three major clusters of the cell-cycle-phase-specific transcription pattern. The other three smaller clusters without distinct transcription patterns were not considered for downstream analyses. Differentially expressed genes among G1/S, S and G2/M phases of the cell cycle were identified using the 'FindAllFeatures' function of Seurat[68] (single-cell analysis package).

## Gene–gene co-transcription

The co-transcription of genes was determined using two criteria: correlation and permutation. scGRO–seq reads were collected from up to the first 10 kb of genes after 500 bp regions at both ends were trimmed (*n* = 15,666). The genes by cells expression matrix was binarized. For the correlation approach, pairwise correlation was performed for all gene pairs, and the *P* value was calculated using the chi-square test. It was adjusted for multiple hypothesis tests using the Benjamini–Hochberg correction method.

Permutation was performed by shuffling the cell identifiers of reads while maintaining their gene assignments. The permutation method accounts for several unknown and known biases and, more importantly, maintains the number of reads in each cell. The observed and permuted co-transcription frequencies of gene pairs were calculated. The empirical *P* value for a gene pair was determined by counting the incidence of equal or higher co-transcription frequency in 1,000 permutations compared with the observed co-transcription frequency.

Gene pairs with correlation coefficients of greater than 0.1 and multiple hypothesis corrected *P* values of less than 0.05 from the correlation approach and an empirical *P* value of less than 0.05 from the permutation approach were considered co-transcribed. A network of pairwise co-transcribed genes was created using the Leiden algorithm, and the modules were selected for gene ontology analyses using the clusterProfiler R package.

## Enhancer–gene co-transcription

Enhancer–gene co-transcription was determined following the logic of gene–gene co-transcription, substituting genes on one arm with enhancers. scGRO–seq reads were collected from up to the first 10 kb of genes after 500 bp regions at both ends were trimmed, and from at least a 3 kb region around enhancers (1,500 bp sense and 1,500 bp antisense) after a 500 bp region around the TSS was removed to avoid paused polymerases. Strand-specific reads on either side of the enhancer TSS were combined to determine enhancer expression. The features (genes + enhancers) by cell expression matrix was binarized, and the co-transcribed enhancer–gene pairs were determined using the correlation and permutation tests, similar to the approach used in the gene–gene co-transcription calculation. The UMIs per cell are maintained in each permutation. Enhancer–gene pairs only from the same chromosomes were retained for downstream analyses. We also included non-overlapping SEs identified in mouse ES cells.

## Enhancers of pluripotency factors

Validated enhancers associated with pluripotency transcription factors OCT4 (also known as POU5F1), SOX2, Nanog and KLF4 were collected from studies referenced in the main text. To define time bins within genes, genes were divided into 5 kb bins (2-min bins calculated using the 2.5 kb min⁻¹ constant transcription rate of elongating RNA polymerases) in the sense and antisense direction until the end of the transcription wave called by groHMM[69], or they overlapped bins from other genes. For enhancers, the TSS was first determined based on the strongest OCT4, SOX2 and Nanog chromatin immunoprecipitation

and sequencing (ChIP–seq) peaks. The precise position was determined by evaluating the divergent transcription around them. The reads from corresponding bins in sense and antisense directions were combined.

## CRISPR-validated SEs

A set of validated SEs and their target genes were used from a previously published study referenced in the main text. SEs in gene introns or associated with miRNA were excluded due to the ambiguity in assigning reads and short gene length, respectively. For the time bin analyses, genes and SEs were divided into four 5 kb bins (2-min with the 2.5 kb min⁻¹ constant transcription rate of elongating polymerases) in the sense and antisense direction, limiting the analyses to the first 20 kb. Using a 20 kb region in this analysis yields four 5 kb bins. The TSS was first determined based on the strongest OCT4, SOX2 and Nanog ChIP–seq peaks, and precise position was determined by evaluating the divergent transcription around them. The reads from corresponding bins in sense and antisense directions were combined. The scrambled random pairs in SE–gene time bin analysis represent the co-transcribed bins between SEs and genes that are not the verified pairs.

## External data

Various data types were analysed, compared and benchmarked against this study. PRO–seq data (GSE169044), ChIP data for p300 (GSM2360934), ATAC–seq (GSE169044), CDK9 (GSM1082347), RNA PolII (GSM318444), H3K4me1 (GSM281695), H3K4me3 (GSM1082344), H3K27Ac (GSM594579), OCT4 (GSM1082340), SOX2 (GSM1082341) and Nanog (GSM1082342) were downloaded from the Gene Expression Omnibus database. PRO–seq libraries were prepared using the same cells used for scGRO–seq under identical conditions[70]. Intron seqFISH data on mouse ES cells were downloaded from table S1 of ref. 17. The genes-by-cells intron seqFISH matrix was binarized, and burst frequency was calculated assuming the signal in each gene comes from a burst equivalent to the 10 kb region used in scGRO–seq, given the probes were designed against the introns at the 5′ regions of genes. Mouse ES cell scRNA-seq was used from a previous study[7], and the burst kinetics was downloaded from 41586_2018_836_MOESM5_ESM.xlsx file associated with this study.

## Reporting summary

Further information on research design is available in the Nature Portfolio Reporting Summary linked to this article.

## Data availability

Sequencing files for scGRO–seq, inAGTuC and AGTuC experiments have been deposited into the NCBI's Gene Expression Omnibus database and are accessible through GEO series accession number GSE242176. The published datasets used in this study were obtained from the GEO repository (identifiers GSE169044, GSM2360934, GSM1082347, GSM318444, GSM281695, GSM1082344, GSM594579, GSM1082340, GSM1082341 and GSM1082342), supplementary table S1 of ref. 17, and 41586_2018_836_MOESM5_ESM.xlsx file of ref. 7.

## Code availability

The code used in this study is available from GitHub (https://github.com/jaymahat/scGROseq).

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

**Acknowledgements** We are grateful to the current and past members of the Sharp Laboratory, especially to A. Whipple, S. Garg, V. P. Chauhan, G. Shamu, J. Liberman and R. Shah for discussion and critical review of the manuscript; S. Bose for his help with FACS and A. Bhutkar for his help with computational analyses; and D. Ribeiro and J. Weber for their insight on statistical tests on co-transcriptional measurement. We thank the Koch Institute's Robert A. Swanson (1969) Biotechnology Center for technical support, specifically the Flow Cytometry Facility, for help with FACS; and S. Levine and the staff at BioMicro Center for their help with NGS. This work was supported in part by Koch Institute support (core) grant 5P30-CA014051 from the National Cancer Institute. This work was supported by Program Project grant P01-CA042063 from the NCI (P.A.S.) and by the United States Public Health Service grants R01-GM034277 from the NIH (P.A.S.). The Emerald Foundation Postdoctoral Transition Award currently supports D.B.M. The Gertrude B. Elion Research Fellowship from GSK and the Ludwig Cancer Institute at MIT previously supported him.

**Author contributions** D.B.M. and P.A.S. conceived the study. D.B.M., S.K.W. and J.F. optimized click chemistry, library preparation methods and prepared NGS libraries. D.B.M. and N.D.T. analysed the data with the help of S.E.B. on pre-processing and J.D.M.-R. on scRNA-seq analysis. D.B.M. and P.A.S. wrote the manuscript, and all co-authors provided feedback. P.A.S. supervised the project.

**Competing interests** US patent number US-11519027-B2 on 'Single-cell RNA sequencing using click-chemistry' was granted on 6 December 2022 to the Massachusetts Institute of Technology, Cambridge, MA, USA, on which P.A.S. and D.B.M. are named inventors. The other authors declare no competing interests.

**Additional information**
**Correspondence and requests for materials** should be addressed to Phillip A. Sharp.

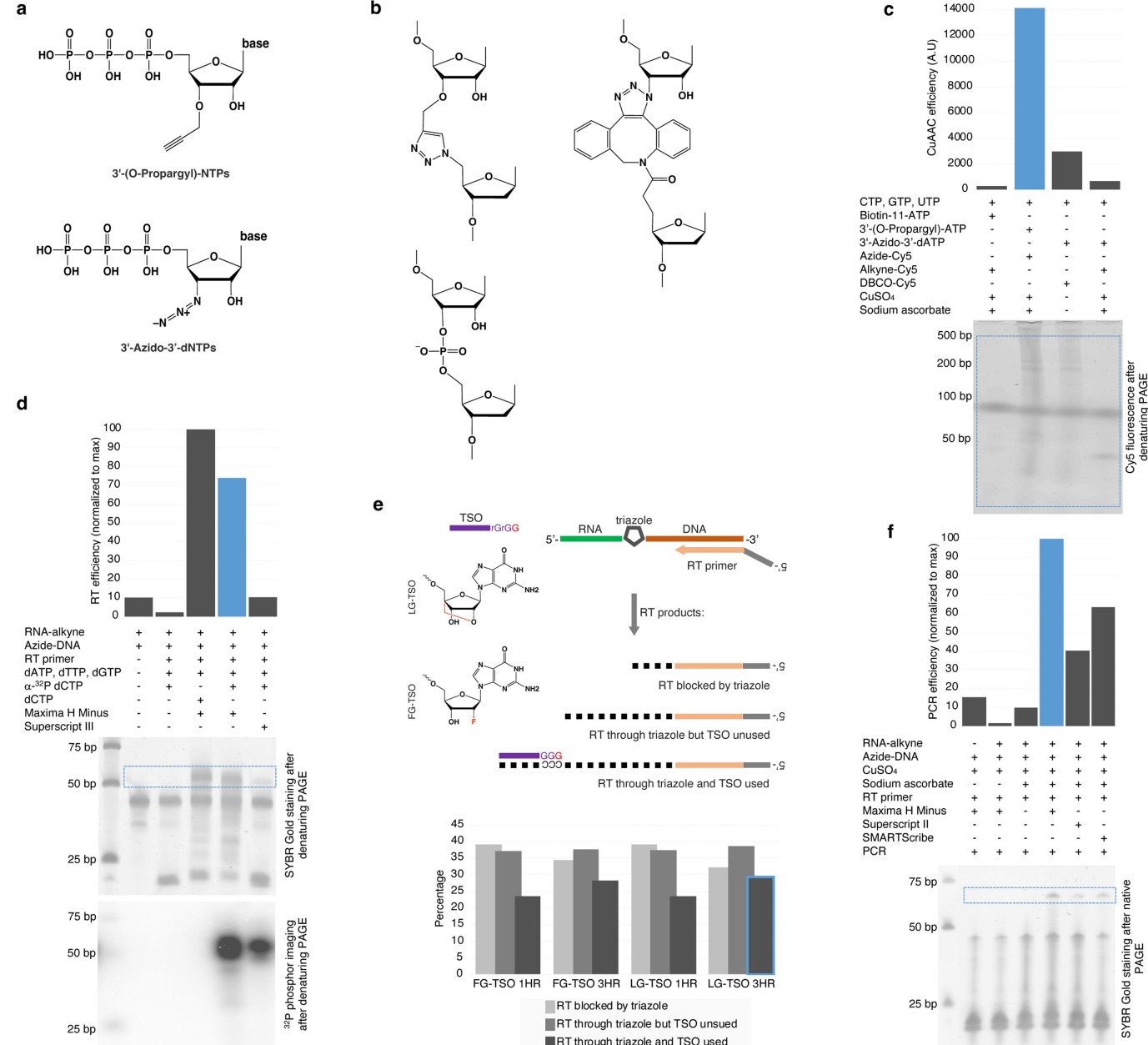

**Extended Data Fig. 1 | click-chemistry mediated nascent RNA conjugation to single-stranded DNA and optimization of reverse transcription.**
**a**, Click-chemistry compatible nucleotides tested in AGTuC development. A few nucleotide triphosphates were custom synthesized or sourced with few properties in mind - smaller size, chain termination ability, and the possibility of incorporation by native RNA polymerases. **b**, Structure of the triazole linkage formed by CuAAC between the nascent-RNA terminally labeled with 3′-(O-Propargyl)-NTPs and the azide-labeled DNA (top left), the linkage formed by SPAAC between the nascent-RNA terminally labeled with 3′-Azido-3′-dNTPs and DBCO DNA (right). The phosphodiester linkage in a native oligonucleotide is shown for comparison (bottom left). **c**, Incorporation efficiency of 3′-(O-Propargyl)-ATP or 3′-Azido-3′-dATP by native RNA polymerase in nuclear run-on reaction. The propargyl or azide labeled nascent RNA is clicked with Cy5 via CuAAC (Azide-Cy5 or Alkyne-Cy5) or SPAAC (DBCO-Cy5), resolved in a denaturing polyacrylamide gel electrophoresis (PAGE), and quantified by measuring the Cy5 fluorescent from the gel image. The blue dotted line

represents the quantified gel region. **d**, Relative quantification of reverse transcription (RT) efficiency of two commercial enzymes traversing through the triazole link formed between the alkyne-labeled RNA and azide-labeled DNA by CuAAC. RT was performed in the presence of either native dCTP or radioisotope a-$^{32}$P dCTP, and the RT reaction was resolved in denaturing PAGE and imaged sequentially for nucleic acid signal (top gel) and radioisotope signal (bottom gel). **e**, Quantification of aborted intermediate and completed desired products (RT through triazole and TSO used) formed during the one hour or three hours of RT using TSO with terminal Locked-Nucleic-Acid-Guanosine (LG) or 2′-Fluoro-Guanosine (FG). **f**, Confirmation and relative quantification of CuAAC, RT, and PCR of clicked product formed between the alkyne-labeled RNA and azide-labeled DNA by three commercial Reverse transcriptase enzymes. **Note:** The blue bar, line, or border represents the "winner" condition. Polyacrylamide gel electrophoresis for **c**, **d**, and **f** was repeated at least twice with the addition or subtraction of some conditions presented here. For gel source data, see Supplementary Fig. 1.

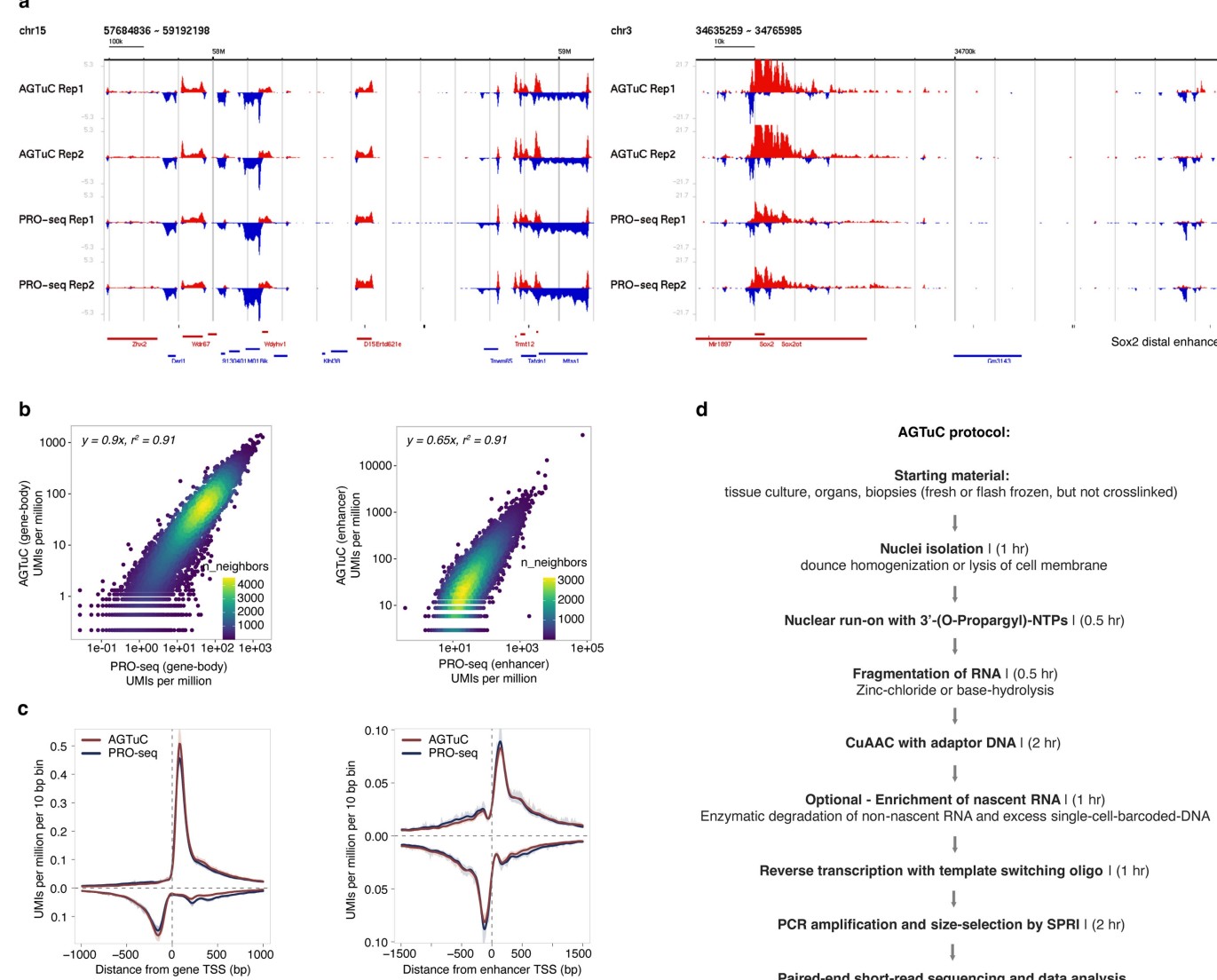

**Extended Data Fig. 2 | Comparison between AGTuC and PRO-seq.**
**a**, Representative genome-browser screenshots with two replicates of
AGTuC and PRO-seq showing a region in chromosome 15 (left) and a region
in chromosome 3 containing the Sox2 gene and its distal enhancer (right)
of the mouse genome (mm10). **b**, Correlation between AGTuC and PRO-seq
UMIs per million sequences in gene bodies (left, n = 19,961) and enhancers
(right, n = 12,542). UMIs from the 500 bp regions from each end of the genes
and 250 bp regions from each end of the enhancers were removed to only

include nascent RNA from elongating RNA polymerases, and the data was
plotted on a log-log scale to show the range of data distribution. **c**, Metagene
profiles of AGTuC and PRO-seq UMIs per million per 10 base pair bins around
the TSS of genes (left, n = 19,961) and enhancers (right, n = 12,542). The line
represents the mean, and the shaded region represents a 95% confidence
interval. **d**, Major steps with the approximate time required in AGTuC library
preparation.

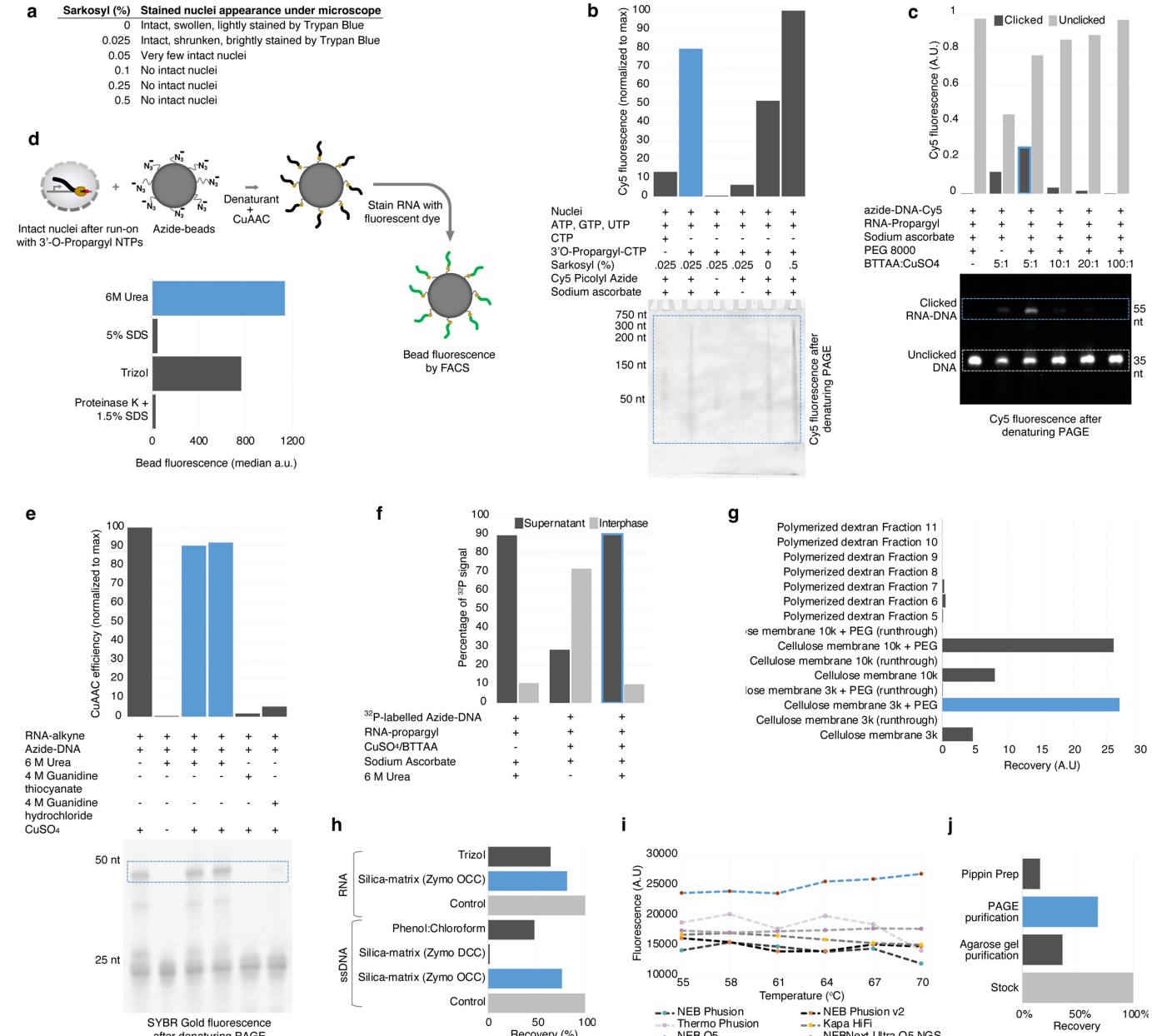

**Extended Data Fig. 3 | Optimization of intact nuclei run-on reaction and NGS library preparation steps. a**, Physical appearance of Trypan Blue stained nuclei under microscope treated with various sarkosyl concentrations. **b**, Relative quantification of nuclear run-on efficiency with various sarkosyl concentrations. Nascent RNA collected after nuclear run-on reaction with either native CTP or click-compatible 3′-(O-Propargyl)-CTP was clicked with Cy5-azide, resolved in denaturing PAGE, and imaged for Cy5 fluorescence. **c**, Effect of different ratios of CuAAC accelerating ligand BTTAA in CuAAC efficiency. RNA-propargyl was clicked with azide-DNA containing Cy5 in the presence of various ratios of BTTAA:CuSO$_4$, resolved in denaturing PAGE, and imaged for Cy5 fluorescence. **d**, Relative quantification of denaturing efficiency of commonly used denaturing agents to release the nascent RNA from RNA polymerase complex. Intact nuclei after run-on with 3′-(O-Propargyl)-NTPs were treated with denaturing agents in the presence of azide-labeled beads and CuAAC reagents, allowing nascent RNA to click with the beads. Beads were stained with RNA-binding dye and measured for fluorescence by FACS. **e**, Effect of denaturing agent's presence in CuAAC efficiency. The blue outline in the image of denaturing PAGE denotes the click product between the RNA-alkyne and azide-DNA. **f**, Role of urea in the residence of clicked RNA-DNA conjugate in either supernatant or interphase of Trizol during the clean-up of

CuAAC reaction, as quantified by the scintillation count of $^{32}$P radioisotope. **g**, Desalting (removal of CuSO4, BTTAA, and sodium ascorbate from CuAAC reaction) efficiency of polymerized dextran and cellulose membrane. Fluorescence from Cy5-labeled RNA-DNA conjugate was measured in elution fractions from columns packed with polymerized dextran and elution from different pore-size cellulose membrane centrifugation tubes with or without PEG 8000. **h**, Relative recovery of ssDNA or RNA from phenol:chloroform or silica-based matrix column purification. Clicked RNA-DNA conjugate was radioisotope labeled using Polynucleotide kinase and γ-$^{32}$P ATP, and the cleaned reaction was quantified using a scintillation counter. **i**, PCR amplification efficiency of clicked RNA-DNA conjugate using different commercial PCR amplification kits. The PCR reaction was resolved in native PAGE, stained with SYBR Gold, and quantified using ImageJ software. **j**, Relative recovery of size-selected dsDNA. A mock NGS library (purified PCR product) was selected for the desired size using various size-selection methods, and the recovered dsDNA was quantified using a dsDNA-specific fluorescence kit (Qubit). The bar represents the average of two independent replicates. **Note:** The blue bar, line, or border represents the "winner" condition. Polyacrylamide gel electrophoresis for **b**, **c**, and **e** was repeated at least twice with the addition or subtraction of some conditions presented here. For gel source data, see Supplementary Fig. 2.

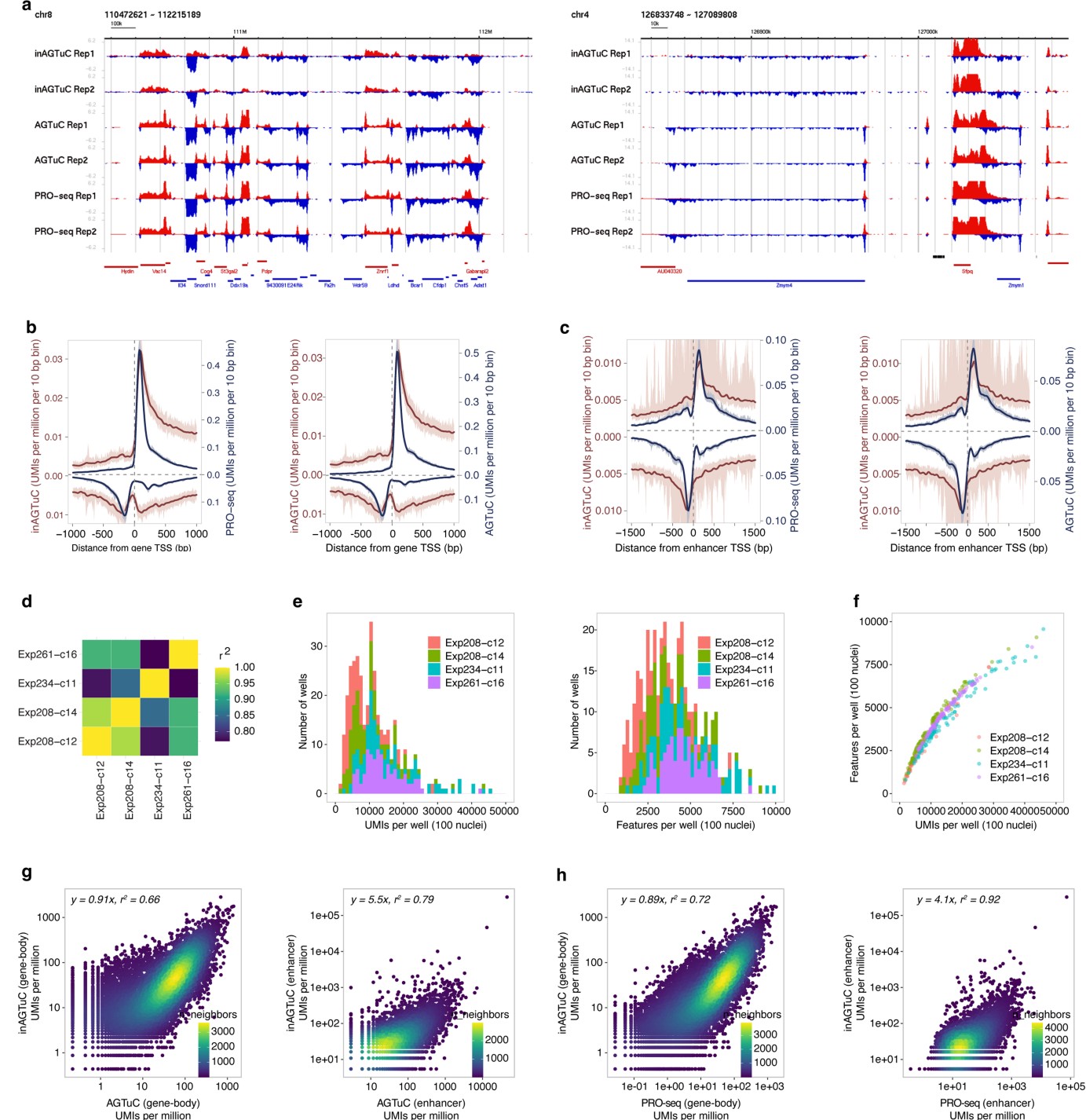

**Extended Data Fig. 4 | Benchmarking inAGTuc against AGTuC and PRO-seq.**
**a**, Representative genome-browser screenshots with two replicates of inAGTuC, AGTuC, and PRO-seq showing a region in chromosome 8 (left) and a region in chromosome 4 of the mouse genome (mm10). **b, c**, Comparison of inAGTuC metagene profiles with PRO-seq and AGTuC using UMIs per million per 10 base pair bins around **(b)** the TSS of genes (n = 19,961) and **(c)** enhancers (n = 12,542). The line represents the mean, and the shaded region represents a 95% confidence interval. **d**, Correlations of inAGTuC UMIs per million sequences in gene bodies (n = 19,961) between the four replicates. **e**, Distribution of UMIs per well (left) and features per well (right) in four replicates of 96-well plate inAGTuC

libraries. Each well contains 100 nuclei. **f**, Relationship between the UMIs per well and the number of features detected per well in four replicates of 96-well plate inAGTuC libraries. **g**, Correlation between inAGTuC and AGTuC UMIs per million sequences in the body of genes (n = 19,961) and enhancers (n = 12,542). **h**, Correlation between inAGTuC and PRO-seq UMIs per million sequences in the body of genes (n = 19,961) and enhancers (n = 12,542). For panels **g** and **h**, UMIs from the 500 bp regions from each end of the genes and 250 bp regions from each end of the enhancers were removed to only include nascent RNA from elongating RNA polymerases, and the data was plotted on a log-log scale to show the range of data distribution.

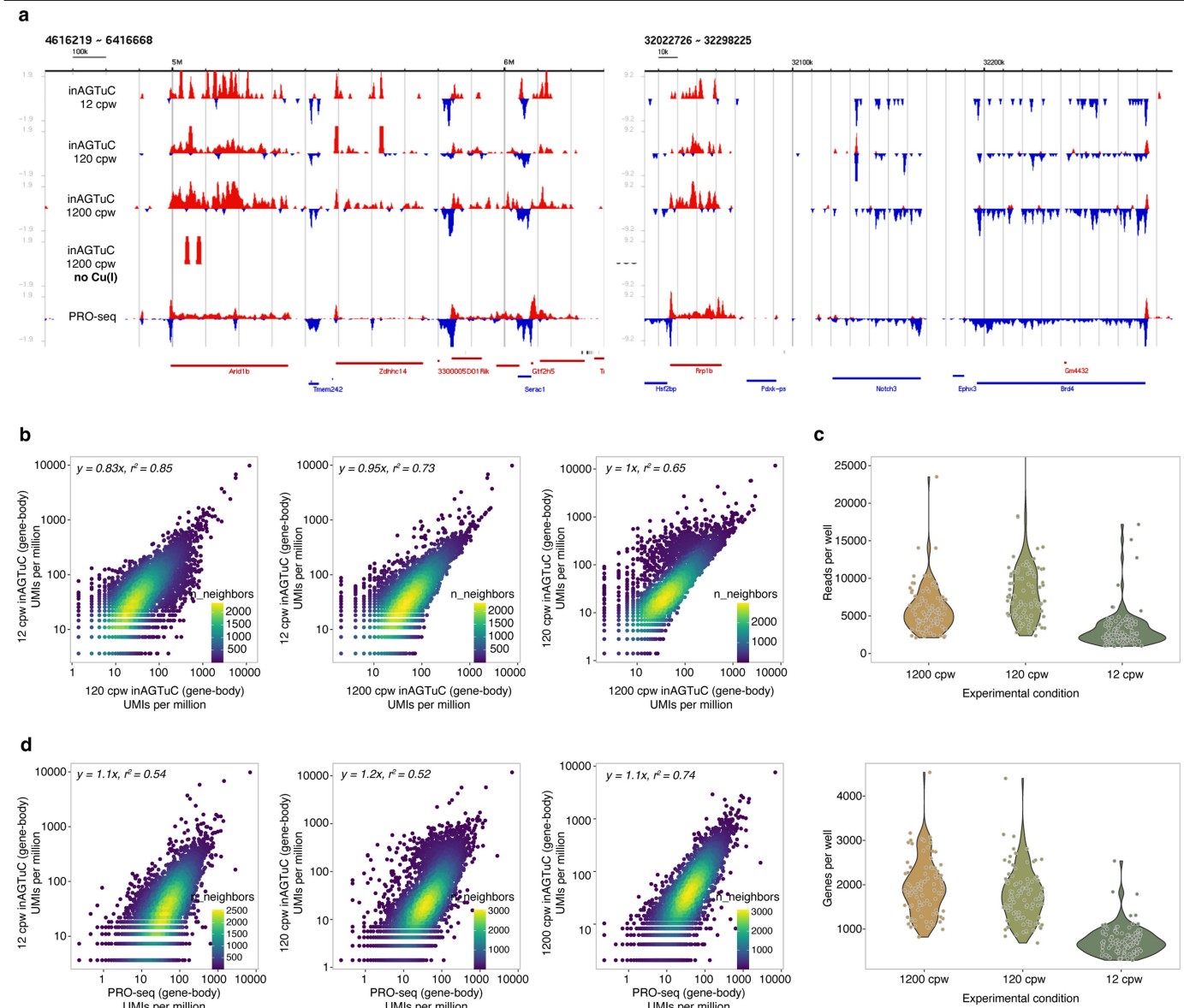

**Extended Data Fig. 5 | Feasibility demonstration of inAGTuC with fewer cells.** **a**, Representative genome-browser screenshots of inAGTuC library from one 96-well plate with each well containing either 12, 120, or 1200 cells per well (cpw) showing two regions in chromosome 17 of the mouse genome (mm10). inAGTuC library with 1200 cpw but without Cu(I) in the CuAAC reaction (fourth track) and the PRO-seq library (fifth track) serve as the negative and positive control, respectively. **b**, Correlations among 12 cpw, 120 cpw, and 1200 cpw inAGTuC libraries in the body of genes (n = 19,961). **c**, Distribution of UMIs per well (top) and genes per well (bottom) in 12 cpw, 120 cpw, and 1200 cpw inAGTuC libraries. **d**, Correlations between PRO-seq and 12 cpw, 120 cpw, and 1200 cpw inAGTuC libraries in the body of genes (n = 19,961). For panels **b** and **d**, UMIs from the 500 bp regions from each end of the genes were removed to only include nascent RNA from elongating RNA polymerases, and the data was plotted on a log-log scale to show the range of data distribution.

**scGRO-seq protocol:**

**Starting material:**
tissue culture, organs, biopsies (fresh or flash frozen, but not crosslinked)

↓

**Nuclei isolation**
dounce homogenization or lysis of cell membrane

↓

**Intact nuclear run-on with 3'-(O-Propargyl)-NTPs**
Removal of excess 3'-(O-Propargyl)-NTPs

↓

**Single nuclei deposition in multi-well plates containing Urea**
Cell dispenser, cell sorter, or limiting dilution

↓

**CuAAC with single-cell-barcoded-DNA**
Automated or manual dispense

↓

**Clean- up and Isolation of single-cell barcoded nascent RNA**
Enzymatic degradation of non-nascent RNA and excess single-cell-barcoded-DNA (optional)

↓

**Fragmentation of RNA**
Zinc-chloride or base-hydrolysis

↓

**Reverse transcription with template switching oligo**

↓

**Selection of cDNA**
Size-selection or enzymatic degradation

↓

**Incorporation of NGS adaptors by PCR pre-amplification**

↓

**Removal of empty amplicon  by CRISPR**
Formed by reverse transcription of excess single-cell-barcoded-DNA incorporating TSO

↓

**Final PCR amplification and size-selection**

↓

**Paired-end short-read sequencing and data analysis**

**Extended Data Fig. 6 | scGRO-seq library preparation.** Major steps involved in scGRO-seq library preparation.

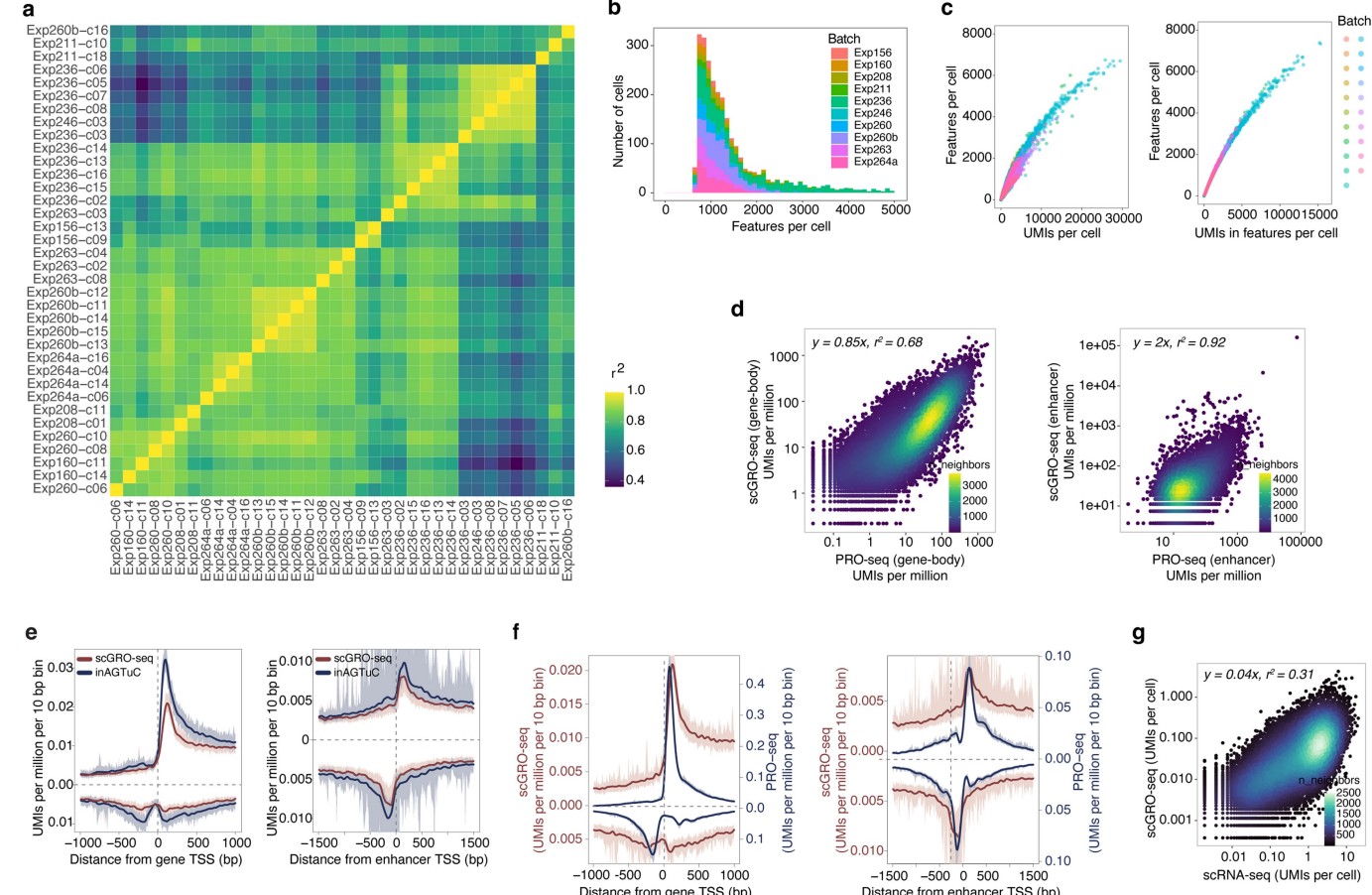

**Extended Data Fig. 7 | Additional benchmarking of scGRO-seq. a**, Coefficient of determination ($r^2$) between each 96-well plate from scGRO-seq batches that passed the quality-control threshold. $r^2$ was calculated from average UMIs per 96 cells in all genes and enhancers. **b**, Distribution of scGRO-seq features (genes + enhancers) per cell. **c**, Relationship between the number of features detected per cell and the UMIs per cell (left) or UMIs in features per cell (right) in scGRO-seq. Colors indicate different batches of scGRO-seq. **d**, Correlation between scGRO-seq and PRO-seq UMIs per million sequences in gene bodies (left, n = 19,961) and enhancers (right, n = 12,542). UMIs from the 500 bp regions from each end of the genes and 250 bp regions from each end of the enhancers were removed to only include nascent RNA from elongating RNA polymerases, and the data was plotted on a log-log scale to show the range of data distribution.

**e**, Metagene profiles of scGRO-seq compared with inAGTuC UMIs per million per 10 base pair bins around the TSS of genes (left, n = 19,961) and enhancers (right, n = 12,542). The line represents the mean, and the shaded region represents the 95% confidence interval. **f**, Comparison of metagene profiles between scGRO-seq and PRO-seq UMIs per million per 10 base pair bins around the TSS of genes (left, n = 19,961) and enhancers (right, n = 12,542). The line represents the mean, and the shaded region represents 95% confidence interval. **g**, Correlation between scGRO-seq and scRNA-seq UMIs per cell in the body of genes (left, n = 19,961). UMIs from the 500 bp regions from each end of the genes were removed to only include nascent RNA from elongating RNA polymerases, and the data was plotted on a log-log scale to show the range of data distribution.

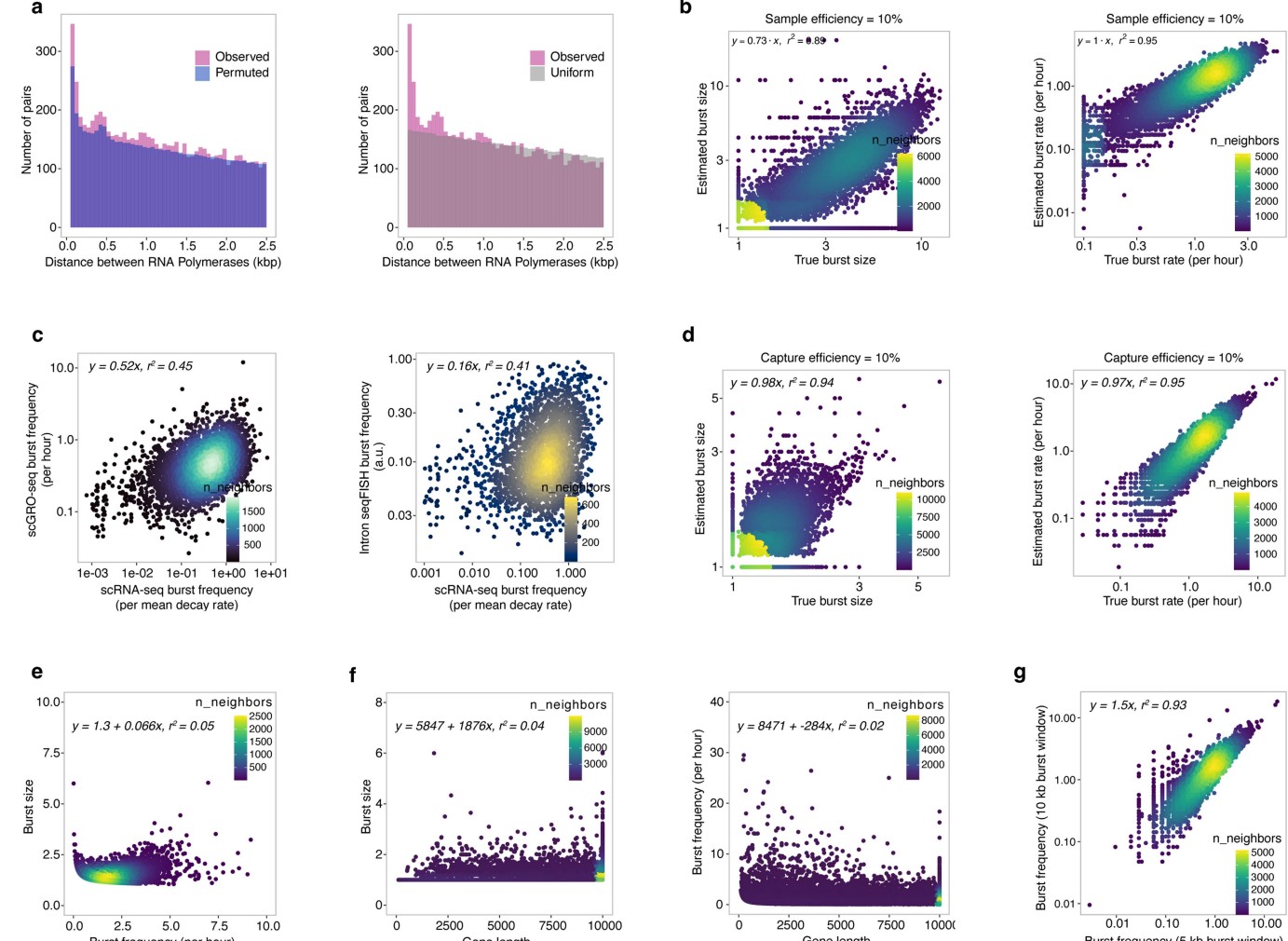

**Extended Data Fig. 8 | Effect of transcription level, gene length, and burst duration in transcription burst kinetics. a**, Distribution of distances between consecutive RNA polymerases in the first 10 kb of the gene body in single cells compared with distances from permuted data (randomized cell ID while maintaining UMIs per cell but unchanged read position, left) or uniform data (randomized read position along the gene but unchanged cell ID, right). Distances up to 2.5 kb are shown. **b**, Test of burst kinetics estimators by simulating burst size and burst frequency. **c**, Test of our burst kinetics estimators by simulating read counts using burst size and frequency inferred from observed scGRO-seq dataset. **d**, Correlation of burst frequency of genes higher than 0.1 in both datasets between scGRO-seq and intron seqFISH. **e**, Correlation between the burst frequency from scGO-seq (top) and intron seqFISH (bottom) with the burst frequency from scRNA-seq. **f**, The effect of gene length (from 100 bp to 10 kb after trimming 500 bp on either end of the genes) on burst size and frequency. **g**, Correlation between burst frequencies calculated from the burst window of either the first 5 kb or the first 10 kb gene bodies. In panels **b-g** with the log-log scale, the data was plotted on a log-log scale to show the range of data distribution. The y = mx fit was derived from linear data.

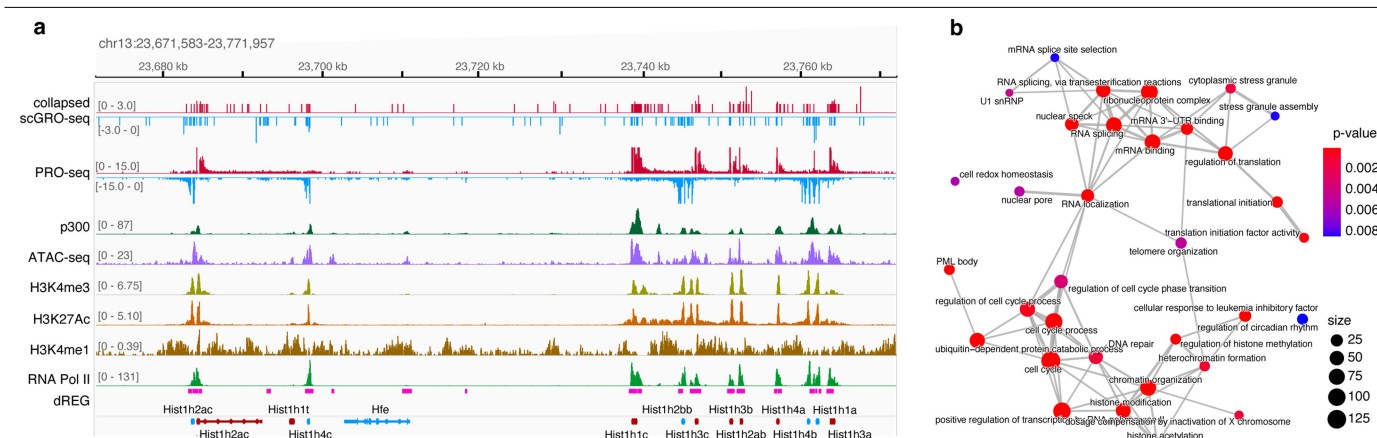

**Extended Data Fig. 9 | Co-transcription of genes with shared function. a**, Genome-browser screenshot of the histone locus body in mouse chromosome 13 showing transcription of replication-dependent histone genes. **b**, Network of enriched gene ontology terms in co-transcribed genes prepared using "enrichGO" function in clusterProfiler R package. A connecting gray line represents at least a 10% overlap of genes between the GO terms. The color of the dots represents the p-value calculated by the clusterProfiler, and the dot size represents the number of contributing genes in the GO term.

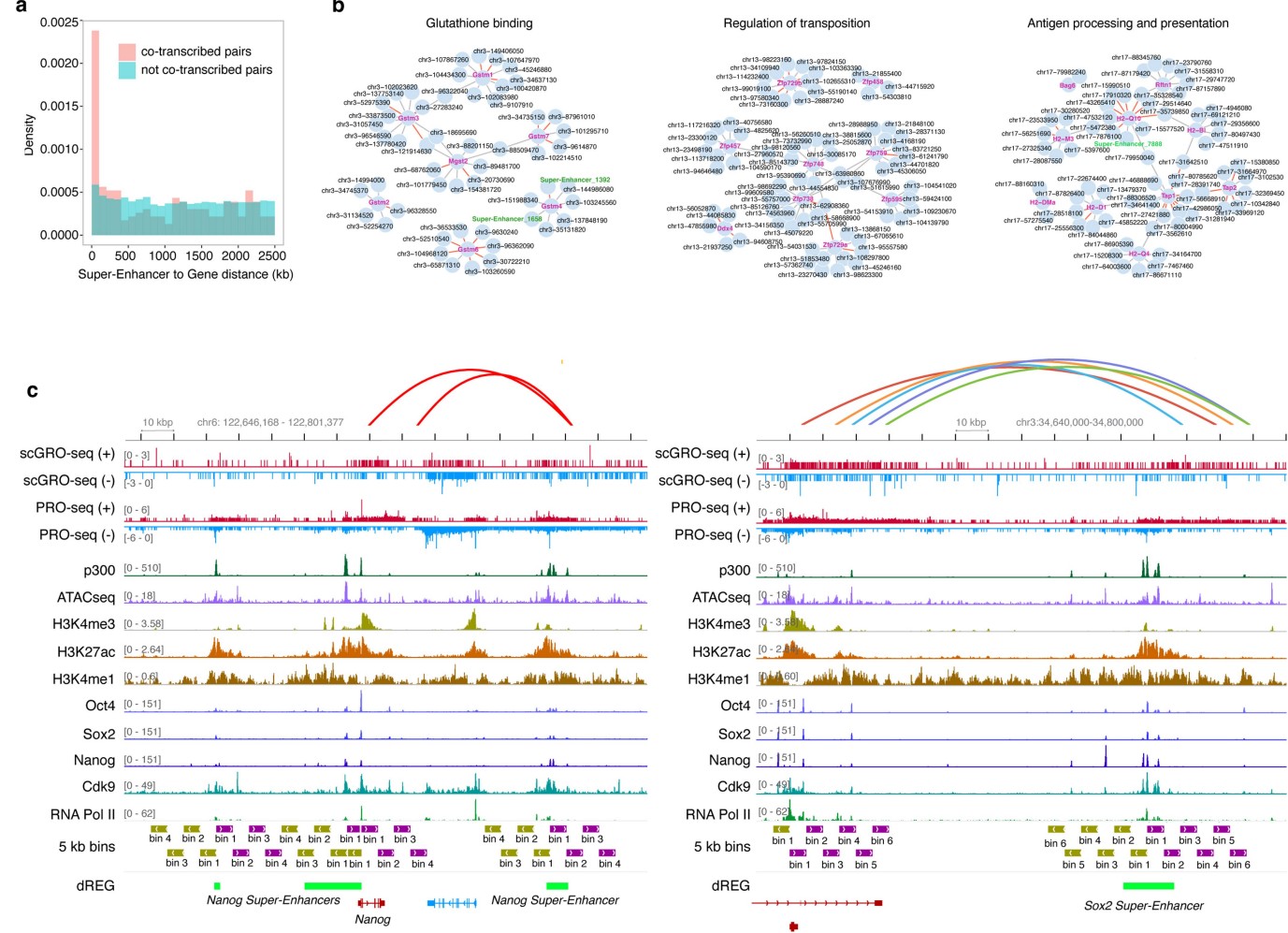

**Extended Data Fig. 10 | Organization of enhancer-gene co-transcription networks. a**, Distance between correlated and non-correlated SE-gene pairs within 2.5 Mb of each other. **b**, Co-transcription network of functionally related genes clustered together on the same chromosome shown as examples. Red edges between the enhancer-gene pairs indicate rho > 0.15, and rho > 0.1 and <0.15 are shown in gray. **c**, Co-transcription between the Sox2 gene and its distal enhancer (left), and the Nanog gene and its three enhancers (right). Green bars represent the annotated SE regions, and the 5 kb bins in sense and antisense strands are represented in magenta and yellow-green bars.

# Reporting Summary

## Statistics

For all statistical analyses, confirm that the following items are present in the figure legend, table legend, main text, or Methods section.

| n/a | Confirmed | |
|---|---|---|
| ☐ | ☒ | The exact sample size ($n$) for each experimental group/condition, given as a discrete number and unit of measurement |
| ☐ | ☒ | A statement on whether measurements were taken from distinct samples or whether the same sample was measured repeatedly |
| ☐ | ☒ | The statistical test(s) used AND whether they are one- or two-sided *Only common tests should be described solely by name; describe more complex techniques in the Methods section.* |
| ☐ | ☒ | A description of all covariates tested |
| ☐ | ☒ | A description of any assumptions or corrections, such as tests of normality and adjustment for multiple comparisons |
| ☐ | ☒ | A full description of the statistical parameters including central tendency (e.g. means) or other basic estimates (e.g. regression coefficient) AND variation (e.g. standard deviation) or associated estimates of uncertainty (e.g. confidence intervals) |
| ☐ | ☒ | For null hypothesis testing, the test statistic (e.g. $F$, $t$, $r$) with confidence intervals, effect sizes, degrees of freedom and $P$ value noted *Give P values as exact values whenever suitable.* |
| ☒ | ☐ | For Bayesian analysis, information on the choice of priors and Markov chain Monte Carlo settings |
| ☒ | ☐ | For hierarchical and complex designs, identification of the appropriate level for tests and full reporting of outcomes |
| ☐ | ☒ | Estimates of effect sizes (e.g. Cohen's $d$, Pearson's $r$), indicating how they were calculated |

*Our web collection on statistics for biologists contains articles on many of the points above.*

## Software and code

Policy information about availability of computer code

| Data collection | No software except Illumina NExtSeq 5000 and NovaSeq basecalling was used. |
|---|---|
| Data analysis | The R (v 4.2.2) scripts written in Jupyterlab (v 3.4.3) used for data analyses are publicly available on GitHub (https://github.com/jaymahat/scGROseq). We additionally used Python (v 3.6.4), cutadapt (v 1.16), bamtools (v 2.5.1), samtools (v 1.10), bedtools (v2.29.2), bowtie2 (v 2.3.5.1), flexbar (v 3.5), fastqc (v 0.11.5), DESeq2 (v 1.38.0), Seurat (v 4.3.0), clusterProfiler (v 4.6.0), IGV (v 2.13.0), and GSEA (v 4.3.3). Transcription unit calling was performed using groHMM (https://github.com/dankoc/groHMM). Enhancers were called using dREG-HD (https://github.com/Danko-Lab/dREG.HD). |

For manuscripts utilizing custom algorithms or software that are central to the research but not yet described in published literature, software must be made available to editors and reviewers. We strongly encourage code deposition in a community repository (e.g. GitHub). See the Nature Portfolio guidelines for submitting code & software for further information.

# Data

Policy information about <u>availability of data</u>

All manuscripts must include a <u>data availability statement</u>. This statement should provide the following information, where applicable:
- Accession codes, unique identifiers, or web links for publicly available datasets
- A description of any restrictions on data availability
- For clinical datasets or third party data, please ensure that the statement adheres to our <u>policy</u>

The raw and processed data generated in this study are deposited in Gene Expression Omnibus under accession number GSE242176. The published datasets analyzed for this study were obtained from the GEO repository (GSE169044, GSM2360934, GSE169044, GSM1082347, GSM318444, GSM281695, GSM1082344, GSM594579, GSM1082340, GSM1082341, GSM1082342), Supplementary Table S1 of a published manuscript (https://doi.org/10.1016/j.cell.2018.05.035) and reprocessed, and 41586_2018_836_MOESM5_ESM.xlsx file of a published manuscript (https://doi.org/10.1038/s41586-018-0836-1).

# Research involving human participants, their data, or biological material

Policy information about studies with <u>human participants or human data</u>. See also policy information about <u>sex, gender (identity/presentation), and sexual orientation</u> and <u>race, ethnicity and racism</u>.

| | |
|---|---|
| Reporting on sex and gender | N/A |
| Reporting on race, ethnicity, or other socially relevant groupings | N/A |
| Population characteristics | N/A |
| Recruitment | N/A |
| Ethics oversight | N/A |

Note that full information on the approval of the study protocol must also be provided in the manuscript.

# Field-specific reporting

Please select the one below that is the best fit for your research. If you are not sure, read the appropriate sections before making your selection.

☒ Life sciences        ☐ Behavioural & social sciences        ☐ Ecological, evolutionary & environmental sciences

For a reference copy of the document with all sections, see nature.com/documents/nr-reporting-summary-flat.pdf

# Life sciences study design

All studies must disclose on these points even when the disclosure is negative.

| | |
|---|---|
| Sample size | We performed scGRO-seq on 39 96-well plates and 3,744 cells, of which 36 plates and 2,635 cells passed the threshold.<br><br>No Sample size calculation was performed. We collapsed the scGRO-seq libraries to generate psuedo-bulk and compared against inAGTuC, AGTuC, and PRO-seq library prepared from millions of nuclei. We found robust recapitulation of nascent-RNA profiles generated from 2,635 single cells and deemed that the scGRO-seq sample size is sufficient for the analyses we performed in this manuscript. |
| Data exclusions | The scGRO-seq batches with r2 of at least 0.6 against at least 60% of all batches were selected for further analysis. Cells were required to contain a minimum of 1,000 UMIs and 750 features for further analysis. |
| Replication | We performed scGRO-seq on 39 96-well plates and 3,744 cells, of which 36 plates and 2,635 cells passed the threshold.<br><br>The 36 replicates of scGRO-seq libraries were prepared over the span of three years. The robustness of correlation among various batches as presented in Extended Data Fig. 7a demonstrates the reproducibility of scGRO-seq method.<br><br>At least two replicates were prepared for inAGTuC and AGTuC libraries. |
| Randomization | Mouse embryonic stem cells after run-on with 3'-O-Propargyl NTPs were randomly sorted into 96-well plates. 16 frozen 96-well plates with single mES cell in each well out of 40-60 plates prepared for each experiment were randomly selected for scGRO-seq library preparation. |
| Blinding | Blinding was not necessary. We performed scGRO-seq library preparation 39 times and the samples were randomly handled by three researches at various stages. The roles assigned in tissue culture of mES cells, harvesting of nuclei, run-on with 3'-O-Propargyl NTPs was random among the three researchers. |

# Reporting for specific materials, systems and methods

We require information from authors about some types of materials, experimental systems and methods used in many studies. Here, indicate whether each material, system or method listed is relevant to your study. If you are not sure if a list item applies to your research, read the appropriate section before selecting a response.

## Materials & experimental systems

| n/a | Involved in the study |
|-----|----------------------|
| ☒ | ☐ Antibodies |
| ☐ | ☒ Eukaryotic cell lines |
| ☒ | ☐ Palaeontology and archaeology |
| ☒ | ☐ Animals and other organisms |
| ☒ | ☐ Clinical data |
| ☒ | ☐ Dual use research of concern |
| ☒ | ☐ Plants |

## Methods

| n/a | Involved in the study |
|-----|----------------------|
| ☒ | ☐ ChIP-seq |
| ☒ | ☐ Flow cytometry |
| ☒ | ☐ MRI-based neuroimaging |

## Eukaryotic cell lines

Policy information about <u>cell lines and Sex and Gender in Research</u>

| | |
|---|---|
| Cell line source(s) | V6.5 mouse embryonic stem cells (mESCs) was used in this study. It was established by the Jaenisch laboratory (Whitehead Institute, Massachusetts Institute of Technology) from the inner cell mass (ICM) of a 3.5-day-old mouse embryo from a C57BL/6(F) X 129/sv(M) cross. |
| Authentication | V6.5 mouse embryonic stem cells were authenticated under microscope for tissue culture phenotype, size of nuclei during FACS, and more importantly, the nascent RNA profile compared with previously published nascent RNA profiles from mouse embryonic stem cells. |
| Mycoplasma contamination | Cell lines were tested for mycoplasma contamination on a regular basis using a PCR-based test and confirmed for mycoplasma-free. |
| Commonly misidentified lines (See <u>ICLAC</u> register) | N/A |

## Plants

| | |
|---|---|
| Seed stocks | *Report on the source of all seed stocks or other plant material used. If applicable, state the seed stock centre and catalogue number. If plant specimens were collected from the field, describe the collection location, date and sampling procedures.* |
| Novel plant genotypes | *Describe the methods by which all novel plant genotypes were produced. This includes those generated by transgenic approaches, gene editing, chemical/radiation-based mutagenesis and hybridization. For transgenic lines, describe the transformation method, the number of independent lines analyzed and the generation upon which experiments were performed. For gene-edited lines, describe the editor used, the endogenous sequence targeted for editing, the targeting guide RNA sequence (if applicable) and how the editor was applied.* |
| Authentication | *Describe any authentication procedures for each seed stock used or novel genotype generated. Describe any experiments used to assess the effect of a mutation and, where applicable, how potential secondary effects (e.g. second site T-DNA insertions, mosiacism, off-target gene editing) were examined.* |

