## [Peer Review file · Nature]

Manuscript Title: Single-cell nascent RNA sequencing unveils coordinated global transcription

Reviewer Comments & Author Rebuttals

Redactions – Third Party Material

Reviewer Reports on the Initial Version:

Referees' comments:

Referee #1 (Remarks to the Author):

In this manuscript, Mahat et al. developed two new methodologies for profiling nascent transcription: the first is an Assay of Genome-wide Transcriptome using Click chemistry (AGTuC) to join nascent RNAs labeled by a run-on reaction with 3'-(O-Propargyl)-NTPs in mESCs; the second also uses a nuclear run-on reaction with 3'-(O-Propargyl)-NTPs in absence of sarkosyl after which nuclei are sorted individually into 96-well plates and processed to produce a single cell GRO-seq (scGRO-seq). The first AGTuC assay is described as a means of doing PRO-seq on 1200 cells or possibly less (depending on coverage needs) in a protocol that takes only 8 hrs. This technical improvement of the PRO/GRO-seq protocol will make applications of GRO/PRO-seq applicable to more researchers, especially those working in systems where large cell numbers are difficult to obtain. The second provides a long sought single cell assay of transcribing RNA Polymerase II (Pol II)

The authors then use this scGRO-seq to uncover several features of transcriptional dynamics at the single-cell level. Employing the scGRO-seq assay, the authors identified and characterized the dynamics of transcription bursts, unveiled the co-transcription of functionally related genes, and they showed that the bursting of transcription at super-enhancers precedes the transcription of target genes. Identification of these transcription characteristics using scGRO-seq can greatly enhance our fundamental understanding of transcription mechanisms, including coordinated regulation between enhancers and target gene expression, which is very challenging to interrogate using existing bulk nascent transcriptomic assays. However, the current version of the method captures a very small fraction (less than 10%) of the nascent transcripts from transcriptionally highly-active, cultured mouse embryonic stem cells, and therefore, interpreting some of the presented data and applying the current version of the method to other transcriptionally less active cell types could be challenging. This and other concerns listed below need to be addressed.

Major comments

1. Since the scGRO-seq captured a very small fraction (estimated at 10%) of actively engaged Pol II, it would be valuable to provide details on the types of regions captured by scGRO-seq. Of course, the authors already acknowledge that promoter-proximal paused Pol II is not efficiently captured by scGRO-seq, and the overall correlation of inAGTuC and PRO-seq is reasonably good as shown in Extended Data Fig. 4h ($r^2=.65$). However, the absence of sarkosyl could potentially affect run-on reactions differentially across the specific regions of the genome: genes in more chromatin-open regions might be preferentially run-on transcribed; or highly expressed genes might offer more efficient run-on reactions than lowly expressed genes. Additionally, the authors could perform run-on with O-propargyl nucleotides in inAGTuC conditions, remove nucleotides and wash the nuclei,

then provide biotin-NTP under PRO-seq conditions (0.5% Sarkosyl). Sequencing the resulting biotin-nascent RNAs should reveal only paused Pol II, if indeed all the gene body RNA polymerases were already run-on and chain terminated. Such comparisons could be reassuring to users of the method or help clarify the types of specific questions that can be most rigorously addressed.

2. To demonstrate the method's potential for dissecting cellular heterogeneity using nascent transcription, the authors should use at least one heterogeneous cell population. For example, performing scGRO-seq on mouse ES cells differentiating into specific lineages (e.g., neural differentiation) could assess the method's utility in deciphering cellular and functional heterogeneity, which are fundamental applications for any single-cell methodology. Alternatively, an scGRO-seq analysis of a simple reconstructed, defined mix of different cell types, which show a range of relatedness, would help assess the effectiveness of the methodology.

3. The low capturing efficiency, combined with the technical inability of the scGRO-seq to detect promoter-proximal Pol II signal, limits the interpretation of bursting kinetics. The authors mention that highly paused genes exhibit higher burst frequency. How do the authors reconcile the concept of a burst phase with pause-release? Is bursting size and bursting frequency related to features of gene regulation that suggest these bursting properties are modulated at Pol II recruitment or pause-release? The authors do some analysis of core promoter elements, but do genes and promoters with similar burst sizes or frequencies share any functional or structural similarities?

4. The authors' explanation for the superiority of scGRO-seq over scRNA-seq for classifying cells based on their cell cycle phase is not entirely convincing. Recent versions of scRNA-seq methods are not dependent on polyadenylation of the transcript, and they can efficiently capture non-polyadenylated transcripts including histone genes (VASA-seq by Salmen et al, 2022 and STRS by McKellar et al, 2023). The authors should consider revising the text and conclusions where appropriate. Additionally, they could perform UMAP analysis to determine if cells can be clustered into distinct groups based on the cell cycle profile.

5. The manuscript's most critical findings are co-transcription of functionally related genes and temporal coordination between genes and enhancers, which can't be confidently dissected using existing bulk nascent transcriptomic methods and require nascent transcription profiling at single cell level, as perfectly employed by the authors. However, presentations of these findings need to be supported by assessment of variations between individual cells. Figures 4 and 5, along with associated supplementary figures, do not adequately highlight these variations. For example, the authors should indicate what fraction of cells exhibit co-transcription for specific processes and provide genome browser tracks illustrating co-transcribing genes in specific individual cells but not in others. The same approach should be applied to enhancer-promoter pairs in Figure 5.

Minor Comments

1. Given its single base-pair resolution nature due to the incorporation of a single chain-terminating O-Propargyl clickable nucleotide during the run-on reaction, we suggest the single-cell assay should be called single-cell click-based PRO-seq (sccPRO-seq). Also, instead of introducing too many acronyms, we suggest the following alternative names for the other two assays: for the AGTuC assay consider Click-based PRO-seq (cPRO-seq or PROclick-seq): and for inAGTuC assay, consider intact

nuclei Click-based PRO-seq (incPRO-seq).

2. While the authors effectively demonstrate the efficiency of cycloaddition using 3'-O-Propargyl-ATP and its reverse transcription, it remains unclear how efficiently Pol II incorporates O-Propargyl-NTPs compared to Biotin-NTPs. What fraction of O-Propargyl RNA is coupled with Azide-DNA via Click-chemistry?

Referee #2 (Remarks to the Author):

The manuscript from Mahat et al. describes the development of a bioorthogonal approach to bar tag immature RNA molecules for amplification and single cell sequencing. The impact of this approach is the ability to detect immature RNAs, which are typically hard to amplify given the lack of a functional handle like the poly-A tail of mature RNAs. The chemistry is an extension of "Click Code Seq," which was developed for looking at sites of base excision repair (J. Am. Chem. Soc. 140 (31): 9783–9787). Assuming that the click chemistry and RT-PCR read through of the resulting triazole are both consistent for different RNAs (see comment below), this tool is a powerful method for performing experiments that were quite difficult to do in the past, several of which they explore including co-transcription of genes and enhancers, histone transcription, etc. I believe that this method could be very useful and would support publication after the authors respond to my comments.

Comment #1 - The authors don't appear to have referenced the Click Code Seq paper mentioned above. I may have missed it, but if I didn't they should mention this technique and reference the paper.

Comment #2 - The CuAAC and RT-PCR steps do not go to 100%. Therefore, the reliability of the data should be dependent on the efficiency of these steps being the same across experiments and at each propargyl-dNTP. However, it appears that the authors only tested these step with propargyl-dATP on the RNA (Extended Data Figure 1). While I agree that the CuAAC is not likely to be dependent on the nature of the base, the same might not be true for RT-PCR. I suggest that the authors use their best conditions and confirm that the nature of the propargyl-dNTP does not alter the efficiency of either of these two steps.

Referee #3 (Remarks to the Author):

Mahat et al present scGRO-seq, an exciting new technique to study nascent transcription in single cells. Similar to GRO-seq or Pro-seq, this technique resolves the positions of individual transcriptionally engaged polymerases with base-pair resolution, but it is the first approach to provide also single cell resolution. This clearly is a methodological breakthrough and of major interest to many fields of research. However, some of the computational analyses must be developed further and better explanations are necessary. As presented, I am not yet fully convinced that the data quality is sufficiently high to provide quantitative estimates of transcriptional bursting and to "investigate the mechanisms of transcription regulation and the role of enhancers in gene

expression" as claimed in the abstract.

Major concerns:

1. The capture efficiency is not only per se an interesting parameter of scGRO-seq, the estimate (10%) obtained here is also critical to the analyses regarding "evidence of bursting de novo without prior assumptions" and the estimation of burst sizes and frequencies. The authors describe two ways to derive the capture efficiency. The first way is based on the comparison to intron seqFISH shown in Fig. 1f. Here, I have two concerns:

a. The capture efficiency of 10% is taken to be the probability of *any Pol2* that currently transcribes a gene to be detected by scGRO-seq. This would only be valid if only scGRO-seq reads were considered corresponding to Pol2 that transcribed the intron during that time the intron is not yet degraded. I give an example: For the sake of the argument let's say the capture efficiency of intron seqFISH is 100%, we have a 100kb gene, and the intron is immediately degraded after co-transcriptional splicing within 1 min after splicing. Further, let's say we have on average 1 intron per cell detected, and 0.23 scGRO-seq reads per cell (in the whole gene body) on average. In this simplified example, we thus have, on average, *one* Pol2 in the region 2.5kb after the FISH probes of the gene (this also depends on where in the intron the probes are). This means that we have 40 Pol2 currently transcribing the gene (in the whole gene body) per cell on average. The probability to detect a Pol2 is not 23%, it is 0.006%. In summary, this estimate of capture efficiency depends on the intron splicing kinetics and the length of the gene. If splicing is indeed co-transcriptional, the capture efficiency being the above mentioned probability is overestimated.

b. The 23% are apparently taken from the linear regression shown in Fig. 1f. I think I understand the basic idea ($y=0.23x$, if y is scGRO-seq reads and x is detected introns per cell). I do not understand why an intercept term was fitted and how to interpret it. Even more importantly: The fit was apparently done on the log-log plot, i.e. $\log(y)=0.23 \log(x)$, which means $y=x^{0.23}$. Why this is related to the capture efficiency is not clear to me.

2. The second way to estimate capture efficiency is based on a measurement of the number of engaged Pol2 in HeLa cells from 1996 and makes several risky assumptions (20% of Pol2 is in paused state - Reference missing; the number of active Pol2 is proportional to the genome size - why should this be the case; HeLa cells are 2.2 times more transcriptionally active than mESCs - this is based on a computational comparison of tumor vs normal RNA-seq samples with the mean factor being 2.2, but depending on the samples studied, with varying factors of 0.5 to 8; this did not include a comparison of cancer-derived cell lines such as HeLa, nor embryonic stem cells, which might indeed be more active than somatic tissue cells). Thus, how accurate the 10% capture efficiency actually is, is not clear. This is important for its usage in the analyses done in the manuscript.

3. The authors use their data to assess "evidence of bursting de novo without prior assumptions" by analysing the number of "multiplets" (genes with more than one read in a cell). The actually observed multiplets are compared to the ones observed after a simple permutation approach (for each read, maintain the gene, but assign a random cell). The permutations do not respect that cells are from different batches with quite different average read depths. After the permutation approach, all cells will have around the same number of reads. If multiplets predominantly occur in cells with many reads (which seems likely to me), doing the permutation globally would reduce the

occurrence of multiplets. The authors should adapt their permutation approach such that the total read number is maintained for each cell (and it is not sufficient to perform the same permutation within batches, since also within a batch the read numbers are quite heterogeneous).

4. From the permutation approach (see 3.), the authors conclude that the occurrence of multiplets is 2.4% higher than expected by chance. From the 10% capture efficiency estimate, the authors conclude that "the probability of detecting two consecutive RNA polymerases on a gene is 1%". This 1% is then compared to the 2.4%. I need more explanations as to why this comparison is relevant (a probability vs. a relative increase over expectation; two consecutive Pol2 vs. more than 1 observed from potentially much more Pol2).

5. scGRO-seq was used to estimate bursting parameters.

a. The simulation that is used to validate the estimates seems unrealistic: Once the number of bursts are simulated, they are "scaled" by the burst size, i.e. always the mean burst size is taken per burst, instead of drawing it randomly from an appropriate distribution. The variance of data simulated by this will therefore be much smaller than in reality.

b. Even according to these simulations, the estimators used are strongly biased (Ext Fig 8b, linear regression $y=0.96+0.64x$, y being the estimated burst size, x being the simulated burst size; for an unbiased estimator, $y=x$).

c. Why are the results in Ext Fig. 8b and d so different? I appreciate that the true parameters (x axis) in d are obtained from the estimates from the data (instead of a normal distributions as in b), but why are the corresponding estimates qualitatively that different?

6. scGRO-seq was used to assess "whether these genes are transcriptionally synchronized" (as opposed to "co-expression" of "accumulated mRNA"). I am not convinced that the data and in particular the analyses done (focus on 10kb at the start of the gene body) allow conclusions about "transcriptional coordination between any gene pair or network of genes":

a. Analyses are done on binarized matrices. While Pearson correlation can be used with binary variables, the t test computed by `cor.test` in R assumes normally distributed variables and should not be used (there are alternatives, such as chi-square statistics).

b. According to the text, an "empirical false-discovery rate" is also used to filter. The permutation approach to estimate this suffers from the same flaw as mentioned above (see concern 3). Here the authors say "The permutation method accounts for several unknown and known biases, such as read depth per cell." If just "cell IDs" are shuffled, how is this accounted for? The authors should adapt their permutation approach such that the total read number is maintained.

c. The empirical p value is derived from 1000 permutations. This is not corrected for multiple testing (and much more permutations would be necessary to apply Benjamini-Hochberg or similar).

d. In the end the argument is based on testing for association of two binary variables. Even if the authors had accounted for confounding factors, if there is a subset of the cells where the gene is expressed, and not expressed in the others (e.g. as it is expected for cell cycle genes), association would not necessarily mean "transcriptional coordination": Take any pair of genes (not correlated at all), and add more and more cells where both genes are 0. At some point there will be a highly

significant association. Clearly this does not mean that their expression is synchronized in single cells, just that they are co-expressed in the same subset of the cells. It is therefore not surprising that circadian and cell cycle related genes come out of that analysis. Thus, as presented, scGRO-seq data do not bring benefits over data of "accumulated RNA".

7. All these concerns also apply to the association of gene-enhancer pairs that were analyzed using the same methodology.

8. *Four* super enhancers have correlations in the first few 5kb bins with the first few 5kb bins of their genes that might suggest that transcription at enhancers precedes transcription of the gene. The result section rightfully is careful about this: "However, any conclusions will require a much deeper data set." However, the abstract says that this "indicates that the bursting of transcription at super-enhancers precedes the burst from associated genes", and the end of the introduction mentions "preliminary evidence for the transcription initiation at enhancers before the transcription activation". These two statements are not backed by convincing data and should be removed.

Additional concerns:

- The start of the results section is very dense. It (I think rightfully) introduces AGTuC and inAGTuC, but it refers to 5 full page Extended Figures before the first main figure is presented. There is no description of the results in these figures (except for the very short figure legends) and no discussion. I suggest to add this in a supplementary document.
- The differences of inAGTuC and scGRO-seq profiles along gene bodies in Fig 1b to PRO-seq profiles are attributed to the absence of high concentrations of a strong detergent. The authors cite the groHMM paper here, which likely is the wrong reference?
- Ext Fig. 5 says 12, 120 and 1200 cells, text says 100k, 10k and 1k nuclei.
- The text always talks about "reads", while I believe it is "deduplicated reads". I suggest to refer to them as UMIs.
- The manuscript shows a lot of log-log scatterplots. It is not clear where the zeros are (pseudocounts?), and what the colorscale is showing.
- Line 149f: Please show the correlation excluding the promoter-proximal region!
- Fig 1f: "Intron seqFISH (reads per cell)"; it is not reads!
- Fig 1g: It is not clear which scRNA-seq data set that is.
- It is not described how the fdrs in "evidence for bursting" for the data from Fig 2b were estimated.
- Fig 2c: How was a KS test computed from this? Why does the x axis stop at 2.5kb if the window is 10kb?
- Line 227f: "Genes with the TATA element exhibited a larger burst size than genes lacking it, and the presence of the Initiator sequence further increased the burst size" - p values are required to back this claim.
- Fig 5a: How were KS tests performed? Why is there a drop by 50% in the left most bin for uncorrelated pairs?

- Methods: Better descriptions of the computational approaches in general are required. One example: The provided code hints at a custom definition of transcriptional units using groHMM, this is not described. Other example: Were there cells that were filtered out? (Based on Fig. 1c it seems as if cells were filtered by a threshold on features per cell - it true, reporting 1503 features on average per cell is not reasonable).

Author Rebuttals to Initial Comments:

1 **Single-cell nascent RNA sequencing using click-chemistry unveils coordinated** 2 **transcription**

Referee expertise:

Referee #1: transcription, nascent RNA sequencing

Referee #2: click chemistry

Referee #3: single-cell analysis

9 **Note:**

Figures in the manuscript are referred to as they are in the manuscript.

Figures prepared to respond to the reviewer's comments are denoted by **R1** for **Referee**
#1, **R2** for **Referee #2**, and **R3** for **Referee #3**.

14 **Referee #1 (Remarks to the Author):**

In this manuscript, Mahat et al. developed two new methodologies for profiling nascent
transcription: the first is an Assay of Genome-wide Transcriptome using Click chemistry
(AGTuC) to join nascent RNAs labeled by a run-on reaction with 3'-(O-Propargyl)-NTPs
in mESCs; the second also uses a nuclear run-on reaction with 3'-(O-Propargyl)-NTPs in
absence of sarkosyl after which nuclei are sorted individually into 96-well plates and
processed to produce a single cell GRO-seq (scGRO-seq). The first AGTuC assay is
described as a means of doing PRO-seq on 1200 cells or possibly less (depending on
coverage needs) in a protocol that takes only 8 hrs. This technical improvement of the
PRO/GRO-seq protocol will make applications of GRO/PRO-seq applicable to more
researchers, especially those working in systems where large cell numbers are
challenging to obtain. The second provides a long-sought single-cell assay of transcribing
RNA Polymerase II (Pol II). The authors then use this scGRO-seq to uncover several
features of transcriptional dynamics at the single-cell level. Employing the scGRO-seq
assay, the authors identified and characterized the dynamics of transcription bursts,
unveiled the co-transcription of functionally related genes, and they showed that the
bursting of transcription at super-enhancers precedes the transcription of target genes.
Identification of these transcription characteristics using scGRO-seq can significantly
enhance our fundamental understanding of transcription mechanisms, including
coordinated regulation between enhancers and target gene expression, which is very
challenging to interrogate using existing bulk nascent transcriptomic assays. However,
the current version of the method captures a very small fraction (less than 10%) of the
nascent transcripts from transcriptionally highly active, cultured mouse embryonic stem
cells, and therefore, interpreting some of the presented data and applying the current
version of the method to other transcriptionally less active cell types could be challenging.
This and other concerns listed below need to be addressed.

We thank the reviewer for their insightful comments and for their time to review our
manuscript. We are grateful for their recognition of the potential our new methodologies
hold to enhance the fundamental understanding of transcription mechanisms.

We acknowledge the reviewer's concern regarding the capture efficiency of nascent
transcripts and understand the significance of these issues for the robustness and
generalizability of our techniques. While the capture efficiency achieved in this study has

room for improvement, like the first publication of every other single-cell method, it has
enabled the exploration of nascent RNA and transcriptional mechanisms at the single-
cell level for the first time. The field of nascent transcription is acutely aware of the
challenges in capturing nascent RNA molecules. This difficulty primarily stems from the
lower abundance of nascent RNA per cell, which is about one-tenth that of mRNA
molecules(Cui and Irudayaraj, 2015; Marinov et al., 2014; Shah et al., 2018). Therefore,
it is not surprising that it took us 15 years since the development of the first nascent RNA
sequencing method(Core et al., 2008) to develop the single-cell version. In contrast,
scRNA-seq(Tang et al., 2009) and scATAC-seq(Buenrostro et al., 2015) were developed
more quickly following their respective bulk versions(Bainbridge et al., 2006; Buenrostro
et al., 2013).

It is also important to note that the capture efficiency of scGRO-seq aligns with that of the
mRNA capture efficiency in scRNA-seq methods. For instance, in the landmark Drop-seq
scRNA-seq method(Macosko et al., 2015), the estimated capture efficiency of mRNA per
cell was approximately 12.8%. This efficiency decreased to 10.7% when assessed with
independent digital expression measurements using droplet digital PCR.

Even though the initial scRNA-seq and scATAC-seq studies presented findings with
limited throughput and coverage, they nevertheless set the stage for subsequent
improvements. We similarly anticipate progressive enhancements in scGRO-seq's
throughput, capture efficiency, and applicability through the contributions of the broader
scientific community in the coming years.

Major comments:

1. Since the scGRO-seq captured a very small fraction (estimated at 10%) of actively
engaged Pol II, it would be valuable to provide details on the types of regions captured
by scGRO-seq. Of course, the authors already acknowledge that promoter-proximal
paused Pol II is not efficiently captured by scGRO-seq, and the overall correlation of
inAGTuC and PRO-seq is reasonably good as shown in Extended Data Fig. 4h
($r^2=0.65$). However, the absence of sarkosyl could potentially affect run-on reactions
differentially across the specific regions of the genome: genes in more chromatin-open
regions might be preferentially run-on transcribed; or highly expressed genes might
offer more efficient run-on reactions than lowly expressed genes. Additionally, the
authors could perform run-on with O-propargyl nucleotides in inAGTuC conditions,
remove nucleotides and wash the nuclei, then provide biotin-NTP under PRO-seq
conditions (0.5% Sarkosyl). Sequencing the resulting biotin-nascent RNAs should
reveal only paused Pol II, if indeed all the gene body RNA polymerases were already
run-on and chain terminated. Such comparisons could be reassuring to users of the
method or help clarify the types of specific questions that can be most rigorously
addressed.

We appreciate the reviewer's insight regarding potential variability in run-on
transcription efficiency across genomic regions. Following this suggestion, we
assessed the run-on efficiency as a function of chromatin accessibility, categorizing
genes into four bins based on ATAC-seq-derived chromatin accessibility from mouse
embryonic stem cell data obtained from a recent study(Hu et al., 2022). We analyzed

the correlation between inAGTuC (0.025% sarkosyl) and PRO-seq (0.5% sarkosyl)
 across these gene groups, similar to the data presented in **Extended Data Figure 4h**
 of the original manuscript. We rationalized that if genes with higher chromatin
 accessibility have higher run-on efficiency at 0.025% sarkosyl, then we would observe
 a better correlation with PRO-seq. Our findings, detailed in **Figure R1.1.1**, indicate
 that genes with more open chromatin did not consistently exhibit increased run-on
 efficiency.

Figure R1.1.1. Assessment of run-on efficiency on genes with various degree of chromatin openness.

Our findings corroborate a previous study's results regarding sarkosyl's impact on
 transcription run-on efficiency. **Figure R1.1.2** (adapted from (Core et al., 2012))
 illustrates that run-on efficiency on the bodies of genes remains unaffected by the
 presence or absence of sarkosyl (**Figure R1.1.2D**). However, run-on efficiency without
 sarkosyl is reduced at the 5' ends of genes, as depicted in **Figure R1.1.2C**, likely
 attributable to the role of sarkosyl in allowing RNA polymerase to run-on by dislodging
 the transcriptional pausing factors like NELF and DSIF. Because of this reduced run-
 on efficiency at the 5' ends of genes, we excluded single-cell GRO-seq reads from the

Redactions – Third Party Material

- A. The composite profile of GRO-seq data shows the density reads in 10bp windows from 200bp to +500bp relative to TSSs for run-ons performed with or without sarkosyl. The Y-axis represents read/window/million reads sequenced. The number of genes shown is 11,800.
- B. Schematic showing how GRO-seq signal was quantified at promoters, the gene body, or at gene ends.
- C. Scatter plots showing the effects of sarkosyl on the run-on signal in promoters.
- D. Scatter plots showing the effects of sarkosyl on the run-on signal in genes.

initial 500 nucleotides of genes in our analysis to avoid misrepresentation due to
decreased labeling of nascent RNA. This exclusion is a more cautious measure than
the referenced study by an additional 200 nucleotides (**Figure R1.1.2B**). Nonetheless,
the run-on efficiency appears consistent for RNA Polymerase II molecules post-
promoter-proximal pause, as shown in **Figures R1.1.2A & R1.1.2D**.

In regard to the reviewer's suggestion of "perform run-on with O-propargyl nucleotides
in inAGTuC conditions, remove nucleotides and wash the nuclei, then provide biotin-
NTP under PRO-seq conditions," while this experiment could capture paused and
elongating Pol II, it would still be limited to bulk cell analysis, as the reviewer notes.
We and others have not established a technique to isolate biotinylated RNA from
single cells with sufficient efficiency. We acknowledge the reviewer's concerns
regarding the underrepresentation of paused RNA Polymerase II (Pol II) in our
scGRO-seq data, but the study of pause regulation is beyond the scope of this study.
Developing such methodologies would be time- and resource-intensive, making it
difficult to justify within the scope of this study.

2. To demonstrate the method's potential for dissecting cellular heterogeneity using
nascent transcription, the authors should use at least one heterogeneous cell
population. For example, performing scGRO-seq on mouse ES cells differentiating
into specific lineages (e.g., neural differentiation) could assess the method's utility in
deciphering cellular and functional heterogeneity, which are fundamental applications
for any single-cell methodology. Alternatively, a scGRO-seq analysis of a simple
reconstructed, defined mix of different cell types, which show a range of relatedness,
would help assess the effectiveness of the methodology.

Dissecting cellular heterogeneity is an important feature of single-cell RNA
sequencing experiments. To demonstrate scGRO-seq's capability, we utilized
asynchronous mouse embryonic stem cells, capturing cells in different cell cycle
stages, as shown in **Figure 3**. It's noteworthy that mouse embryonic stem cell cycle
stages are challenging to deduce using scRNA-seq, a fact underscored by a landmark
scRNA-seq study (Klein et al., 2015), which observed: "single-cell data do not reveal
broader evidence of cell-cycle-dependent transcription in ES cells." Unlike most
scRNA-seq, scGRO-seq leverages replication-dependent histone genes and
transcriptionally verified gene sets specific to cell cycle stages, thus offering insights
into cell cycle and cellular heterogeneity.

Moreover, the focus of this manuscript is on biological insights into the dynamics of
transcriptional burst kinetics, the co-transcriptional regulation of genes, and the
coordination between genes and enhancers under steady-state conditions. We,
therefore, avoided the introduction of external perturbations. Delineating cellular
heterogeneity in tissues or *in vitro* differentiated cells, such as differentiating ES cells
into specific lineages like neural differentiation, as suggested by the reviewer, would
require considerable methodological improvements in scGRO-seq's throughput in
order to encompass the full spectrum of heterogeneity. However, scGRO-seq is
currently a low-throughput method - similar to the first reports of single-cell assays,
such as scRNA-seq (Tang et al., 2009) (hundreds of individual blastomeres) and

scATAC-seq(Buenrostro et al., 2015) (few hundred homogeneous tissue culture
cells). We expect that the scGRO-seq method will see continuous improvements in its
capacity for processing large numbers of cells and its effectiveness in capturing a
higher fraction of nascent RNA through the contributions of the wider scientific
community.

- 3. The low capturing efficiency, combined with the technical inability of the scGRO-seq
to detect promoter-proximal Pol II signal, limits the interpretation of bursting kinetics.
The authors mention that highly paused genes exhibit higher burst frequency. How do
the authors reconcile the concept of a burst phase with pause-release? Is bursting
size and bursting frequency related to features of gene regulation that suggest these
bursting properties are modulated at Pol II recruitment or pause-release? The authors
do some analysis of core promoter elements, but do genes and promoters with similar
burst sizes or frequencies share any functional or structural similarities?

Our analysis indicated that genes containing motifs associated with promoter-proximal
paused Pol II, as identified by co-PRO-seq(Tome et al., 2018), exhibit increased burst
frequencies across the body of the gene. As previously discussed, the inability of
scGRO-seq to detect promoter-proximal paused Pol II constrains our ability to explore
the concept of a burst phase with pause release, as requested by the reviewer.

We concur that our current data and interpretations related to paused genes are
incomplete. Therefore, we have excluded the only instance of data and assertions
regarding paused Pol II (**Figure 2g**) as this constitutes a minor component of our study
and does not impact the findings presented in **Figure 2**.

Regarding the reviewer's additional comment on whether the modulation of bursting
parameters occurs at the stages of Pol II recruitment or pause release, we are
cautious in drawing conclusions due to the above-mentioned limitation of our data
around the promoter-proximal pause regions. In addressing whether genes and
promoters with similar burst sizes or frequencies share any functional or structural
similarities, we have presented the functional similarity using gene set enrichment
analyses in **Figure 2h**. We show that Myc target genes have increased burst size and
Aff4 target genes have higher burst frequency. A previous single-molecule imaging
study(Patange et al., 2022) illustrated that Myc increases burst duration, thus
augmenting burst size. Likewise, Aff4, integral to the super elongation complex (SEC),
is implicated in facilitating the release of paused Pol II, aligning with observations of
higher burst frequency in genes bound by SEC(Byun et al., 2012). The comprehensive
functional enrichment analysis for genes sharing burst size and frequency
characteristics is detailed in **Table 2**.

- 4. The authors' explanation for the superiority of scGRO-seq over scRNA-seq for
classifying cells based on their cell cycle phase is not entirely convincing. Recent
versions of scRNA-seq methods are not dependent on polyadenylation of the
transcript, and they can efficiently capture non-polyadenylated transcripts including
histone genes (VASA-seq by Salmen et al, 2022 and STRS by McKellar et al, 2023).
The authors should consider revising the text and conclusions where appropriate.

Additionally, they could perform UMAP analysis to determine if cells can be clustered
into distinct groups based on the cell cycle profile.

We are grateful to the reviewer for highlighting the single-cell methodologies that are
independent of polyadenylated transcripts. The cell-cycle results section of the
manuscript has been updated to state the limitation of polyadenylated RNA-based
scRNA-seq methods as opposed to scRNA-seq methods. The revised text also
includes an acknowledgment and reference of the single-cell RNA-seq methods that
are not contingent on polyadenylation (VASA-seq by Salmen et al., 2022 and STRS
by McKellar et al., 2023), as suggested by the reviewer.

However, it should be recognized that the methods cited by the reviewer generally
capture all RNA, with ribosomal RNA being depleted in VASA-seq only. The proportion
of intronic and other non-coding RNA isolated by these methods, especially the
enhancer-RNA, is considerably low. Similarly, the presence of a steady-state level of
cell-cycle-specific genes in the cytoplasm, also detected by these methods, could
hinder the precise inference of temporal resolution, which scGRO-seq overcomes by
capturing only actively transcribed nascent RNA.

Regarding the reviewer's suggestion to utilize UMAP for data representation, in light
of the limitations of dimensionality reduction (Chari and Pachter, 2023), we prefer our
current approach for determining and displaying cell-cycle stages in the manuscript
as it effectively identifies, quantifies, and communicates the essence of cell-cycle
heterogeneity in our data.

5. The manuscript's most critical findings are co-transcription of functionally related
genes and temporal coordination between genes and enhancers, which can't be
confidently dissected using existing bulk nascent transcriptomic methods and require
nascent transcription profiling at single cell level, as perfectly employed by the authors.
However, presentations of these findings need to be supported by assessment of
variations between individual cells. Figures 4 and 5, along with associated
supplementary figures, do not adequately highlight these variations. For example, the
authors should indicate what fraction of cells exhibit co-transcription for specific
processes and provide genome browser tracks illustrating co-transcribing genes in
specific individual cells but not in others. The same approach should be applied to
enhancer-promoter pairs in Figure 5.

We thank the reviewer for recognizing scGRO-seq's unique ability to analyze the co-
transcription of functionally related genes and the temporal coordination between
enhancers and genes.

We regret the oversight in not clearly stating that the supplementary tables provide
the details requested by the reviewer. The proportions of cells exhibiting significant
co-transcription of gene pairs are detailed in the **8th column of Table 4**, and genes
implicated in particular processes are listed in **Table 5**. Similarly, the proportions of
cells with enhancer-gene co-transcription are provided in the **6th column of Table 6**.
Recognizing the importance of this information as per the reviewer's advice, we have

now updated the table legends with further descriptions to clearly communicate the
data structure.

In regard to the reviewer's request for "genome browser tracks illustrating co-
transcribing genes in specific individual cells but not in others," we have tried to
interpret this request in **Figure R1.5A**. Co-transcription is difficult to visualize in
conventional genome browsers as it requires visually assigning a read to a cell (see
**Figure 1b**). Instead, we have presented an alternative visual representation for the
co-transcription of genes (**Figure R1.5B**). This graphical approach offers a more
insightful and succinct interpretation compared to traditional genome browser tracks.
Due to a rigorous definition of co-transcription—restricted to a four-minute window by
analyzing only up to 10 kb of the gene regions and excluding the initial 0.5 kb—the
fraction of cells co-transcribing a pair of genes is better suited to display in this
approach, rather than in genome browser.

Figure R1.5. Visualizing co-transcription.

A. A proposed schematics for visualization of co-transcription based on the reviewer's comments.

B. A co-transcription visualization approach currently implemented in the manuscript.

We have included a similar illustrative example in **Figure 4a** of the manuscript, which
depicts co-transcription between the genes *Smarcc1* and *Prkdc*. These plots display
both the number of cells exhibiting co-transcription and, crucially, the specific
positioning of transcribing RNA Pol II on the genes. Co-transcription events are
represented by blue circles connected with yellow lines, while RNA Pol II signals
beyond the 10 kb regions (which are not considered for co-transcriptional analyses)
are denoted by gray circles and lines (**Figure 4a**). The script for these plots is available
in our GitHub repository so that interested readers can visualize the co-transcription
of the genomic regions of their interest.

Minor Comments:

6. Given its single base-pair resolution nature due to the incorporation of a single chain-
terminating O-Propargyl clickable nucleotide during the run-on reaction, we suggest

the single-cell assay should be called single-cell click-based PRO-seq (sccPRO-seq).
Also, instead of introducing too many acronyms, we suggest the following alternative
names for the other two assays: for the AGTuC assay consider Click-based PRO-seq
(cPRO-seq or PROclick-seq): and for inAGTuC assay, consider intact nuclei Click-
based PRO-seq (incPRO-seq).

We value the reviewer's attentiveness to the precise terminology of our assay. We
acknowledge that PRO-seq was the first description of base pair resolution. However,
GRO-seq is widely understood as the nascent RNA labeling technique, which is an
important attribute of this study as the first single-cell nascent RNA sequencing
method. The name 'scGRO-seq' was chosen for its historical nod to the first nascent
RNA sequencing method and its broader recognition within the scientific community.
The reviewer suggested names could potentially be confused with co-PRO-seq (Tome
et al., 2018). Nevertheless, we are receptive to the reviewer's suggestion. However,
since our previously filed patent identifies the method as scGRO-seq, renaming our
method would necessitate a clear annotation that both scGRO-seq and scPRO-seq
denote the identical procedure.

7. While the authors effectively demonstrate the efficiency of cycloaddition using 3'-O-
Propargyl-ATP and its reverse transcription, it remains unclear how efficiently Pol II
incorporates O-Propargyl-NTPs compared to Biotin-NTPs. What fraction of O-
Propargyl RNA is coupled with Azide-DNA via Click-chemistry?

We appreciate the reviewer's focus on the comparative efficiency of O-Propargyl-
NTPs versus Biotin-NTPs as substrates for RNA Polymerase II. Although we share
this concern and have sought to address it, developing an unambiguous method to
assess the differential incorporation efficiency is challenging. For instance, performing
run-on experiments with either O-Propargyl-NTPs or Biotin-NTPs followed by
detection using click-chemistry or fluorescently-tagged streptavidin would result in a
composite measurement of run-on efficiency and the labeling method used, rather
than a direct measure of nucleotide incorporation by Pol II only.

We considered two potential solutions to this issue:
a. The synthesis of O-Propargyl-NTPs and Biotin-NTPs bearing an additional
identical label, such as a fluorophore, enables consistent detection of nascent RNA
across both modifications. Unfortunately, we could not find a vendor capable of
producing these modified nucleotides.
b. The use of O-Propargyl-NTPs and Biotin-NTPs labeled with radioactive ^{32}P at the
alpha-phosphate position. However, these nucleotides were similarly unavailable.

Given these constraints, an alternative approach would be to evaluate the end-point
nascent RNA detection efficiencies when O-Propargyl-NTPs or Biotin-NTPs are used
during run-on reaction. While this wouldn't directly measure the relative incorporation
efficiencies by Pol II, it would indicate the overall efficiency of nascent RNA detection
with O-Propargyl-NTPs or Biotin-NTPs, which is the ultimate measure of interest.

To this end, we prepared an AGTuC library with O-Propargyl-NTPs and a PRO-seq
library with Biotin-NTPs, ensuring consistent conditions, cell type (mouse pancreatic

cancer cells), cell number (700,000), processing batch, and handling protocols as shown in **Figure R1.7.1A**. To reduce variability, the PRO-seq protocol was adapted to mirror the AGTuC strategy, specifically in the attachment of a 5' adaptor through template switching oligos. Equal numbers of reads from each library were then analyzed to assess genome coverage at varying sampling rates (**Figure R1.7.1B**). This analysis revealed remarkably similar genome coverage by both methods, suggesting that the overall efficiency of nascent RNA capture and sequencing is on par whether utilizing O-Propargyl-NTPs or Biotin-NTPs.

Figure R1.7.1. Nascent RNA capture efficiency with either O-Propargyl-NTPs or Biotin-NTPs. A, A schematic of AGTuC and modified PRO-seq experiments to measure nascent RNA capture efficiency. **B**, A comparison between AGTuC and modified PRO-seq in genome coverage per sequenced reads.

In response to the reviewer's second inquiry on "What fraction of O-Propargyl RNA is coupled with Azide-DNA via Click-chemistry?", we conducted an assay to quantify the efficiency of coupling O-Propargyl RNA to Azide-DNA via Click-chemistry. The experiment utilized 28-nucleotide RNA labeled with [α -³²P]-CTP, synthesized in vitro, and incorporated either UTP-azide or UTP-alkyne. The design of the DNA template ensured that the click-compatible UTP was incorporated exclusively at the RNA's 3' end. This RNA was then subjected to either CuAAC or SPAAC using commercially synthesized 20-nucleotide azide-DNA or BCN-DNA, respectively, with click-compatible modifications present at the DNA's 5' end. The DNA was used in a 100-fold excess to replicate the scGRO-seq condition. The reactions were conducted both with and without PEG 8000, a molecular crowding agent that enhances the kinetics of the reaction. Following incubation, the products were separated by denaturing PAGE, and both the reacted (clicked) and unreacted (unclicked) products were excised and quantified through scintillation counting (**Figure R1.7.2**).

Under the optimized conditions of the CuAAC reaction—100-fold excess azide-DNA and a 2-hour incubation at 50°C in the presence of 15% PEG 8000—the click reaction

efficiency exceeded 95%. These precise conditions are used in our scGRO-seq protocol. This panel can be added to the supplementary data if the reviewer and editor find it helpful.

Figure R1.7.2. CuAAC and SPAAC quantification.

**Referee #2 (Remarks to the Author):**

The manuscript from Mahat et al. describes the development of a bioorthogonal approach
to bar tag immature RNA molecules for amplification and single cell sequencing. The
impact of this approach is the ability to detect immature RNAs, which are typically hard to
amplify given the lack of a functional handle like the poly-A tail of mature RNAs. The
chemistry is an extension of “Click Code Seq,” which was developed for looking at sites
of base excision repair (J. Am. Chem. Soc. 140 (31): 9783–9787). Assuming that the click
chemistry and RT-PCR read through of the resulting triazole are both consistent for
different RNAs (see comment below), this tool is a powerful method for performing
experiments that were quite difficult to do in the past, several of which they explore
including co-transcription of genes and enhancers, histone transcription, etc. I believe that
this method could be very useful and would support publication after the authors respond
to my comments.

We acknowledge the reviewer’s constructive comments on our manuscript and their
acknowledgment of the potential utility of our biorthogonal approach for tagging nascent
RNA molecules. We concur with the reviewer’s assessment regarding the need to
demonstrate comparable click chemistry and RT-PCR read-through efficiency across
different nucleotides. We have conducted experiments to confirm the robustness of our
method (see **Figure R1.7.2**), and also ensured that the different propargyl-NTPs do not
introduce biases in click-chemistry or RT (see **Figure R2.2**), as requested by the
reviewer.

1. The authors don’t appear to have referenced the Click Code Seq paper mentioned
above. I may have missed it, but if I didn’t they should mention this technique and
reference the paper.

We are grateful to the reviewer for bringing our attention to the omission of the Click
Code Seq paper citation. The paper is now appropriately cited in the revised version
of our manuscript.

2. The CuAAC and RT-PCR steps do not go to 100%. Therefore, the reliability of the data
should be dependent on the efficiency of these steps being the same across
experiments and at each propargyl-dNTP. However, it appears that the authors only
tested these step with propargyl-dATP on the RNA (Extended Data Figure 1). While I
agree that the CuAAC is not likely to be dependent on the nature of the base, the
same might not be true for RT-PCR. I suggest that the authors use their best
conditions and confirm that the nature of the propargyl-dNTP does not alter the
efficiency of either of these two steps.

We thank the reviewer for raising the concern of propargyl-nucleotide bias in CuAAC
and RT-PCR. The overall efficiency of CuAAC for our experimental conditions is
extremely high (**Figure R1.7.2**). Nevertheless, as suggested by the reviewer, we used
our best conditions to test the potential bias introduced by different propargyl-
nucleotides during CuAAC and RT.

Ideally, we would perform this experiment with four species of RNA, each with a
different terminal propargyl-nucleotide. However, the only vendor that synthesizes
RNA with terminal propargyl-nucleotide offers propargyl-ATP only as the terminal

nucleotide, hence the use of propargyl-ATP in **Extended Data Figure 1**. To overcome this limitation, we labeled nascent RNA in four aliquots of 2.5 million nuclei with either propargyl-ATP, propargyl-CTP, propargyl-GTP, or propargyl-UTP. By using only one propargyl-NTP at a time, with the remaining three native NTPs, we ensure that all nascent RNA is terminally labeled with the corresponding propargyl-NTP. We removed the unused NTPs, clicked Cy5-azide to the propargyl-labeled nascent RNA, and then quantified Cy5 fluorescence. We performed these experiments in replicates to assess experimental variation.

Figure R2.2. Assessment of CuAAC and Reverse Transcription bias by propargyl-NTPs. **A**, Bias in CuAAC efficiency as a function of different propargyl-NTPs measured by clicking and quantifying the individually-labelled nascent RNA with azide-Cy5. **B**, Bias in reverse transcription efficiency as a function of different propargyl-NTPs measured by scintillation counting. Propargyl-labeled nascent RNA was clicked with azide-DNA and reverse transcribed in presence of ³²P-CTP. **C**, The genomic composition of nucleotides centered at the run-on added nucleotide.

We found that the experimental variation is larger than the difference in CuAAC efficiency with different propargyl-nucleotides (**Figure R2.2A**).

Similarly, to measure the potential bias in reverse transcription due to different propargyl-nucleotide, we labeled nascent RNA with different propargyl-NTP as described above but clicked with hairpin azide-DNA instead of azide-Cy5 and reverse transcribed in the presence of ³²P-CTP. Similar to the CuAAC bias experiment, we found that the experimental variation is greater than the variation in RT as a function of the propargyl nucleotide (**Figure 2.2B**).

Because these experiments fell short of confirming or denying the potential bias
introduced by different propargyl-NTP during CuAAC and RT, we examined the
composition of the nucleotide added in the run-on reaction in scGRO-seq and its
surroundings. If CuAAC and RT have a bias towards a specific propargyl-NTP, it
would be detected in this analysis (at the position “0”). We did not observe a significant
difference in the frequency of run-on added nucleotide or its surroundings, suggesting
either the absence or undetectable levels of nucleotide bias in CuAAC and RT (**Figure**
**2.2C**).

**Referee #3 (Remarks to the Author):**

Mahat et al present scGRO-seq, an exciting new technique to study nascent transcription
in single cells. Similar to GRO-seq or Pro-seq, this technique resolves the positions of
individual transcriptionally engaged polymerases with base-pair resolution, but it is the
first approach to provide also single cell resolution. This clearly is a methodological
breakthrough and of major interest to many fields of research. However, some of the
computational analyses must be developed further and better explanations are
necessary. As presented, I am not yet fully convinced that the data quality is sufficiently
high to provide quantitative estimates of transcriptional bursting and to “investigate the
mechanisms of transcription regulation and the role of enhancers in gene expression” as
claimed in the abstract.

We express our sincere gratitude to the reviewer for their endorsement of the broad scope
and novelty of scGRO-seq. We concur with the reviewer's constructive suggestion to
enhance the computational analyses and the clarity of explanations in our manuscript. In
response to the reviewer's constructive feedback, we have re-evaluated our
computational strategies, improved data analyses, and provided more detailed
methodological expositions, as detailed below.

Major concerns:

1. The capture efficiency is not only per se an interesting parameter of scGRO-seq, the
estimate (10%) obtained here is also critical to the analyses regarding “evidence of
bursting de novo without prior assumptions” and the estimation of burst sizes and
frequencies. The authors describe two ways to derive the capture efficiency. The first
way is based on the comparison to intron seqFISH shown in Fig. 1f. Here, I have two
concerns:

a. The capture efficiency of 10% is taken to be the probability of *any Pol2* that
currently transcribes a gene to be detected by scGRO-seq. This would only be
valid if only scGRO-seq reads were considered corresponding to Pol2 that
transcribed the intron during that time the intron is not yet degraded. I give an
example: For the sake of the argument let's say the capture efficiency of intron
seqFISH is 100%, we have a 100kb gene, and the intron is immediately degraded
after co-transcriptional splicing within 1 min after splicing. Further, let's say we
have on average 1 intron per cell detected, and 0.23 scGRO-seq reads per cell (in
the whole gene body) on average. In this simplified example, we thus have, on
average, *one* Pol2 in the region 2.5kb after the FISH probes of the gene (this
also depends on where in the intron the probes are). This means that we have 40
Pol2 currently transcribing the gene (in the whole gene body) per cell on average.
The probability to detect a Pol2 is not 23%, it is 0.006%. In summary, this estimate
of capture efficiency depends on the intron splicing kinetics and the length of the
gene. If splicing is indeed co-transcriptional, the capture efficiency being the above
mentioned probability is overestimated.

The reviewer raises an important technical concern on scGROseq's estimated
capture efficiency, which is derived from a comparison with intron seqFISH data.

The reviewer is correct to point out that the estimated capture efficiency of 10%
“would only be valid if only scGRO-seq reads were considered corresponding to

Pol2 that transcribed the intron during that time the intron is not yet degraded”. We
 want to emphasize that we are only considering Pol IIs in scGRO-seq that
 correspond to a similar time window of intron detection before degradation in intron
 seqFISH data. The reviewer correctly states that the “estimate of capture efficiency
 depends on the intron splicing kinetics and the length of the gene” but did not factor
 them in their simplified example. Intron seqFISH probes targeted the first introns,
 whose median length is 7.6 kb (**Figure R3.1A**). At a transcription rate of 2.5 kb/min,
 it takes ~ 3 min to transcribe the introns used in intron seqFISH. More importantly,
 the median time required for intron to be spliced out once it is transcribed ranges
 from 5 to 10 minutes, as reported in several studies using diverse
 methods (Audibert et al., 2002; CLEMENT et al., 1999; Coulon et al., 2014;
 Neugebauer, 2019; Rabani et al., 2014, 2011; Singh and Padgett, 2009). Even if
 we conservatively assume that the fluorescent probes detect the introns only after
 the intron transcription is complete and the introns are immediately degraded after
 splicing, the introns
 are detectable during the splicing time of at
 least 5 to 10 minutes. Considering the
 average splicing time of 8 minutes (from the
 above-mentioned studies and personal
 communication with Daniel Larson from
 NCI, who studies splicing kinetics in live
 cells using advanced imaging modalities)
 corresponds to the transcription time of
 20 kb at 2.5 kb/min.

To maintain a similar
 detection time window
 between the two
 methods, we have
 used Pol II from up to
 20 kb from the TSS for
 the correlation
 analysis between
 scGRO-seq and intron
 seqFISH. We again
 observe a slope of
 0.26 (**Figure R3.1B**).
 The intron seqFISH

Figure R3.1. Correlation between scGRO-seq and intron seqFISH. **A**, Length of first introns of all genes (left panel) and genes used in intron seqFISH (right panel). **B**, Correlation between scGRO-seq UMIs per cell and intron seqFISH counts per cell shown in log scale. **C**, Correlation between scGRO-seq UMIs per cell and intron seqFISH counts per cell shown in linear scale.

estimates its capture efficiency at 44% based on a comparison of a handful of
genes using single-molecule FISH. Using these numbers, we arrive at a similar
estimate of capture efficiency of 11% ($0.23 \text{ of } 0.44 = 0.11$).

- b. The 23% are apparently taken from the linear regression shown in Fig. 1f. I think I
understand the basic idea ($y=0.23x$, if y is scGRO-seq reads and x is detected
introns per cell). I do not understand why an intercept term was fitted and how to
interpret it. Even more importantly: The fit was apparently done on the log-log plot,
i.e. $\log(y)=0.23 \log(x)$, which means $y=x^{0.23}$. Why this is related to the capture
efficiency is not clear to me.

We thank the reviewer for bringing our attention to the unwarranted use of intercept
in our equation. We concur that it is more appropriate to fit for $y = mx$. We have
replotted all of our correlation analyses for $y = mx$, dropping the intercept term. We
see improved correlation across the board. For example, r^2 between scGRO-seq
and intron seqFISH increased from 0.32 to 0.58.

However, we apologize for the reviewer's confusion about the plot. The fit is not
calculated from log-log data. The fit is calculated from the data on a linear scale,
and the data points are plotted on the log-log scale for visualization purposes only.
As shown in **Figure R3.1C**, the linear scale correlation plot fails to display the
range of data, which is clearly shown if the data points are plotted in the log scale.

- 2. The second way to estimate capture efficiency is based on a measurement of the
number of engaged Pol2 in HeLa cells from 1996 and makes several risky
assumptions (20% of Pol2 is in paused state - Reference missing; the number of active
Pol2 is proportional to the genome size - why should this be the case; HeLa cells are
2.2 times more transcriptionally active than mESCs - this is based on a computational
comparison of tumor vs normal RNA-seq samples with the mean factor being 2.2, but
depending on the samples studied, with varying factors of 0.5 to 8; this did not include
a comparison of cancer-derived cell lines such as HeLa, nor embryonic stem cells,
which might indeed be more active than somatic tissue cells). Thus, how accurate the
10% capture efficiency actually is, is not clear. This is important for its usage in the
analyses done in the manuscript.

The second way to estimate capture efficiency by using the number of Pol II molecules
in mammalian cells was intended to provide a complementary approach that is
independent of intron seqFISH. There are not many studies that measure actively
transcribing RNA polymerases in a cell, unlike mRNA measurements. We used the
biochemical studies that quantified the number of actively transcribing RNA
polymerases per cell, which is still one of the most quantitative and direct
measurements of RNA Pol II molecules. Unfortunately, this was done in Hela cells.
We made assumptions to the best of our knowledge in order to make the comparison
fair. The claim that 20% of Pol II is present in a paused state is calculated from PRO-
seq data by us in this study and, hence, no reference. We show that scGRO-seq
misses paused Pol II (**Extended Data Figure 4a & Figure R1.1.2**), and a simple
analysis of PRO-seq and AGTuC data indicates that ~20% of Pol II are in a paused
state.

However, we do agree that other assumptions in this calculation are difficult to confirm,
but neither should they be simply overlooked. We could not ignore the genome size
between mice and humans despite the similar number of genes, as transcription is
widespread beyond genes. Similarly, the transcription level in HeLa cells with abnormal
karyotypes (Landry et al., 2013; Macville et al., 1999) can be generally assumed to be
higher than in karyotypically normal cells, although the precise level may not be clear.
In the absence of a precise quantification between HeLa and mES cells, we used a
mean factor of 2.2-fold from a study comparing RNA levels between tumor and normal
cells. Nevertheless, we understand the concerns raised by the reviewer.

We, therefore, have entirely removed this second approach of estimating capture
efficiency from the manuscript.

A more important clarification, however, is that we do not think the accuracy of 10%
capture efficiency is critical for the analyses done in the manuscript. The capture
efficiency is simply a scaling factor used only to estimate the absolute burst kinetics.
Even there, we show that the absolute burst kinetics derived from scGRO-seq
correlates well with intron seqFISH (**Figure 4c**), whereas similar comparisons with
scRNA-seq-derived burst kinetics show worse correlation for both intron seqFISH and
scGRO-seq (**Extended Data Figure 8c**). The evidence of bursting (**Figures 2b & 2c**)
is unaffected by capture efficiency because the evidence is derived by comparing
against the permuted data (see the response to the reviewer's comment #4 as well).
Similarly, the role of promoter elements in burst kinetics (**Figures 2g & 2h**) is
independent of capture efficiency because the measurements are relative differences
among genes. Overall, the capture efficiency we estimated is based on the only
available single-cell intronic RNA imaging study, and, more importantly, the estimated
capture efficiency does not affect most biological interpretations in this study.

3. The authors use their data to assess "evidence of bursting de novo without prior
assumptions" by analysing the number of "multiplets" (genes with more than one read
in a cell). The actually observed multiplets are compared to the ones observed after a
simple permutation approach (for each read, maintain the gene, but assign a random
cell). The permutations do not respect that cells are from different batches with quite
different average read depths. After the permutation approach, all cells will have
around the same number of reads. If multiplets predominantly occur in cells with many
reads (which seems likely to me), doing the permutation globally would reduce the
occurrence of multiplets. The authors should adapt their permutation approach such
that the total read number is maintained for each cell (and it is not sufficient to perform
the same permutation within batches, since also within a batch the read numbers are
quite heterogeneous).

We thank the reviewer for emphasizing this important aspect of the permutation. We
confirm that the number of reads per cell is maintained by shuffling cell identifiers
across reads. We do not pool reads from all cells and equally divide among the cells
during permutation. For example, if cell A has 5 reads and cell B has 3, we might
permute AAAAABB to BABAABAA but not AABABBAB. We also explicitly ensure

that the read distribution per cell is preserved after permutation. We apologize if the
text in the methods section failed to clearly communicate this important aspect of
permutation. We have modified the text where applicable to unambiguously state that
the reads per cell are maintained in all permutations.

- 4. From the permutation approach (see 3.), the authors conclude that the occurrence of
multiplets is 2.4% higher than expected by chance. From the 10% capture efficiency
estimate, the authors conclude that "the probability of detecting two consecutive RNA
polymerases on a gene is 1%". This 1% is then compared to the 2.4%. I need more
explanations as to why this comparison is relevant (a probability vs. a relative increase
over expectation; two consecutive Pol2 vs. more than 1 observed from potentially
much more Pol2).

We apologize for the confusion about these numbers and thank the reviewer for
correctly identifying this confusion in our writing. The 1% number is an illustrative
example of a uniformly random null model assuming 10% capture efficiency, but it is
not used for comparison, as it does not consider the many complexities of real single-
cell data. We include the 1% multiplet statement to help readers understand the
impact of capture efficiency on the number of detected bursts. The only statistical
comparison on which we base our conclusions is the permuted null model, where
reads per cell are maintained. We have improved the description of **Figure 2b** in the
main manuscript to clearly state the use of the permutation null model to provide
evidence of bursting and not the hypothetical probability of 1% for multiplets detection.

- 5. scGRO-seq was used to estimate bursting parameters.

a. The simulation that is used to validate the estimates seems unrealistic: Once the
number of bursts are simulated, they are "scaled" by the burst size, i.e. always the
mean burst size is taken per burst, instead of drawing it randomly from an
appropriate distribution. The variance of data simulated by this will therefore be
much smaller than in reality.

We thank the reviewer for their suggestion to draw the burst size randomly from
an appropriate distribution. We have re-run the simulation using a normal
distribution for burst size at various capture efficiencies, as requested by the
reviewer (**Figure R3.5.1**), and updated the **Extended Data Figure 8b** with the new
plots.

b. Even according to these simulations, the estimators used are strongly biased (Ext
Fig 8b, linear regression $y=0.96+0.64x$, y being the estimated burst size, x being
the simulated burst size; for an unbiased estimator, $y=x$).

We apologize for this oversight and have updated the correlations to remove the
y -intercept term (as suggested by the reviewer in comment #1b). We have also
included the performance of these estimators at various sampling efficiencies,
including 100% (**Figure R3.5.1**). Our simple estimator explains most of the
variance across a range of sampling efficiencies. The reviewer is correct to note
that our estimators were biased by underestimating burst size, which was more
pronounced at low capture efficiencies, likely due to dropouts. However, the

updated correlation analyses without the y-intercept term significantly correct the perceived bias in the estimator ($r^2 \geq 0.9$ at capture efficiency $\geq 10\%$).

Figure R3.5.1. Performance of the burst kinetics estimators by simulating burst size and burst frequency at various capture efficiency.

c. Why are the results in Ext Fig. 8b and d so different? I appreciate that the true
parameters (x axis) in d are obtained from the estimates from the data (instead of
a normal distributions as in b), but why are the corresponding estimates qualitatively
that different?

As noted above, our simulation does not capture complexities such as differences
in cell number and batch effects. Nevertheless, the estimator's performance is
robust across both datasets (**Figures R3.5.1 & R3.5.2**) ($r^2 \geq 0.9$ at 10% capture
efficiency). The reviewer's suggestion to drop the y-intercept has improved the
estimator's performance across the board—we thank the reviewer for their insight.

Figure R3.5.2. Performance of the burst kinetics estimator by simulating read counts using burst size and frequency inferred from observed scGRO-seq dataset.

6. scGRO-seq was used to assess "whether these genes are transcriptionally
 synchronized" (as opposed to "co-expression" of "accumulated mRNA"). I am not
 convinced that the data and in particular the analyses done (focus on 10kb at the start
 of the gene body) allow conclusions about "transcriptional coordination between any
 gene pair or network of genes":

 a. Analyses are done on binarized matrices. While Pearson correlation can be used
 with binary variables, the t test computed by cor.test in R assumes normally
 distributed variables and should not be used (there are alternatives, such as chi-
 square statistics).

We thank the reviewer for bringing our
 attention to the use of t-test in the
 Pearson correlation analysis. We have
 changed the statistical test to chi-square.
 The chi-square p-value for gene-gene
 pairs is similar to the p-values previously
 calculated using a t-test (**Figure R3.6.1**).
 R scripts in GitHub for gene-gene and
 enhancer-gene are updated to reflect the
 use of the chi-square test to calculate the
 p-value, which is then corrected for
 multiple hypothesis tests.

Figure R3.6.1. Correlation between multiple hypothesis corrected p-values for gene-gene co-transcription using t-test and chi-square test. Only values less than 0.25 in either test are plotted for clarity.

 b. According to the text, an "empirical false-
 discovery rate" is also used to filter. The
 permutation approach to estimate this
 suffers from the same flaw as mentioned
 above (see concern 3). Here the authors
 say "The permutation method accounts
 for several unknown and known biases,
 such as read depth per cell." If just "cell
 IDs" are shuffled, how is this accounted
 for? The authors should adapt their
 permutation approach such that the total read number is maintained.

As explained in our response to the reviewer's comment #3, the reads per cell are
 maintained in permutations. We apologize for the confusion created by failing to
 clearly state that the reads per cell are not equally divided among the cells and
 are, in fact, maintained in each permutation. To further clarify our permutation
 approach, we have outlined a simplified example below:

Seqnames	ranges	strand	cell-ID	permuted cell-ID
chr1	4808020-4808069	+	c01	c02
chr1	4808144-4808171	+	c02	c03
chr1	4808183-4808243	+	c03	c01
chr1	4808217-4808271	+	c03	c02
chr1	4808223-4808280	+	c01	c03
chr1	4808344-4808377	+	c02	c03
chr1	4808383-4808343	+	c03	c01

chr1	4808417-4808472	+	c03	c03
chr1	4808423-4808484	+	c01	c03
chr1	4808544-4808579	+	c02	c01
chr1	4808583-4808543	+	c03	c01
chr1	4808617-4808676	+	c03	c03
chr1	4808623-4808685	+	c01	c02

Cell-ID tally: **c01=4, c02=3, c03=6**
 Permuted Cell-ID tally: **c01=4, c02=3, c03=6**

- c. The empirical p value is derived from 1000 permutations. This is not corrected for multiple testing (and much more permutations would be necessary to apply Benjamini-Hochberg or similar).

The reviewer rightly pointed out that the multiple hypothesis testing of the empirical p-value derived from 1000 permutations would require a significantly higher number of permutations than 1000. The number of hypotheses tested is 15,021 x 15021 for Gene x Gene co-transcription and would require, for example, more than 200 million permutations for Bonferroni correction. Our ability to perform more permutations was limited by computation time (1000 permutations and empirical p-value calculation takes a day using 16 core CPUs in a shared cluster). More importantly, the permuted data would begin to repeat the pattern after a certain number of permutations due to the sparsity of data. We show that the empirical p-value has a good agreement even between 200, 1000, and 2000 permutations (**Figure 3.6.2**), indicating a diminishing improvement in the accuracy of the empirical p-value with additional permutations.

Figure R3.6.2. Correlation of empirical false discovery rates at different number of permutations as indicated.

The lack of multiple hypothesis correction of the empirical p-value derived from permutation tests is precisely the reason we opted for a parallel approach of Pearson correlation, which is used by single-cell papers to measure co-expression. The p-value (now derived using the chi-square test after the reviewer's recommendation) is corrected for multiple hypothesis testing using the Benjamini-Hochberg (BH) correction method. For a Gene-Gene pair to be considered significantly co-transcribed, we require the pairs to pass a threshold in each approach: pairwise correlation ≥ 0.1 and multiple-hypothesis corrected p-value

from chi-square statistics < 0.05 from the correlation approach, and empirical FDR < 0.05 from the permutation approach. We find that the statistically significant co-transcribed gene-gene pairs from the overlap between BH corrected chi-square p-value and the empirical FDR from various numbers of permutations (**Figure 3.6.3**) remain relatively consistent, highlighting the robustness of the dual approach.

Figure R3.6.3. Overlap between co-transcribed gene-gene pairs that pass thresholds of multiple hypotheses corrected p-values from chi-square test (< 0.05) and empirical false discovery rate of 0.05 in different numbers of permutations.

- d. In the end the argument is based on testing for association of two binary variables. Even if the authors had accounted for confounding factors, if there is a subset of the cells where the gene is expressed, and not expressed in the others (e.g. as it is expected for cell cycle genes), association would not necessarily mean "transcriptional coordination": Take any pair of genes (not correlated at all), and add more and more cells where both genes are 0. At some point there will be a highly significant association. Clearly this does not mean that their expression is synchronized in single cells, just that they are co-expressed in the same subset of the cells. It is therefore not surprising that circadian and cell cycle related genes come out of that analysis. Thus, as presented, scGRO-seq data do not bring benefits over data of "accumulated RNA".

The permutation approach controls for the zero-inflated nature of the data noted by the reviewer. In any large-scale dataset such as this, it is difficult to completely

rule out false positives. Nevertheless, we attempt to minimize the false positives in
determining co-transcription by implementing two independent approaches as
described above in our response to the reviewer's concerns. To examine the
strength of scGRO-seq's co-transcriptional analysis, we compared it with gene-
gene co-transcription in intron seqFISH data and found a good agreement ($r^2 =$
0.59, **Figure 4c**).

It is important to note that co-transcription measured by nascent RNA is
substantially different than co-expression measured by steady-state mRNA levels
due to the vastly different timescales involved (4-minute detection window for a 10
791 kb transcription region in scGRO-seq vs several hours of detection window for
accumulated mRNA). We acknowledge the reviewer's concern about the use of
the term "synchronized" and apologize if the reviewer's confusion by the term
"synchrony" or "synchronized" stems from the lack of evidence of order in gene-
gene co-transcription or physical contact between the genes. We concur that we
have not inferred the order of transcription in co-transcribed gene pairs, and the
current data neither captures nor insinuates a physical contact between the pair.
We tried to explicitly state that we are measuring co-transcription, as opposed to
co-expression in scRNA-seq. We think that if two genes are transcribed within four
minutes of each other when an average gene is transcriptionally ON for 7 minutes
(median length of mouse transcription until is 17.5 kb) and remains OFF for 2 hours
(**Figure 2e**), and each phase of cell cycle lasts hours, they are coordinately
transcribed, and likely represents a shared biological function as shown in **Figure**
**4b**. We have replaced the terms "synchrony" or "synchronized" in the manuscript
to avoid the confusion that the reviewer indicated.

7. All these concerns also apply to the association of gene-enhancer pairs that were
analyzed using the same methodology.

We have addressed the reviewer's concerns about gene-gene co-transcription and
applied the relevant suggestions and modifications to the enhancer-gene pairs as well.
For example, the t-test in correlation analysis of enhancer-gene co-transcription is
replaced with a chi-square test. We also confirm that the reads per cell are maintained
during permutations in enhancer-gene analyses. We hope the novelty of the
enhancer-gene coordination analyses at the single-cell level helps readers appreciate
the utility of scGRO-seq and apply it to understand the mechanisms of transcription
regulation.

8. *Four* super enhancers have correlations in the first few 5kb bins with the first few
5kb bins of their genes that might suggest that transcription at enhancers precedes
transcription of the gene. The result section rightfully is careful about this: "However,
any conclusions will require a much deeper data set." However, the abstract says that
this "indicates that the bursting of transcription at super-enhancers precedes the burst
from associated genes", and the end of the introduction mentions "preliminary
evidence for the transcription initiation at enhancers before the transcription
activation". These two statements are not backed by convincing data and should be
removed.

We thank the reviewer for their careful assessment of the claims. We used all available
SE-gene pairs that were experimentally validated in mouse embryonic stem cells and
could not add more validated pairs.

We have made changes to both statements the reviewer indicated and softened the
claim about the order of enhancer-gene transcription to address the reviewer's
concerns.

9. Additional concerns:
a. The start of the results section is very dense. It (I think rightfully) introduces AGTuC
and inAGTuC, but it refers to 5 full page Extended Figures before the first main
figure is presented. There is no description of the results in these figures (except
for the very short figure legends) and no discussion. I suggest to add this in a
supplementary document.

We thank the reviewer for their thoughtful recommendation to enhance the
readability of our manuscript. The individual panels of these figures are explained
in the first few sections of the supplementary data. By briefly stating the presence
of the work in the main text and indicating the presence of a detailed explanation
in the supplementary file, we wanted to make readers aware of the careful
optimization of methods, which could be helpful in further enhancement of scGRO-
seq in the future. We are happy to reorganize the main and supplementary texts
in the manuscript as per the reviewers' and editors' suggestions.

b. The differences of inAGTuC and scGRO-seq profiles along gene bodies in Fig 1b
to PRO-seq profiles are attributed to the absence of high concentrations of a strong
detergent. The authors cite the groHMM paper here, which likely is the wrong
reference?

The reviewer might be referring to lines 146-148 in the manuscript, which states -
"However, scGRO-seq is less efficient in capturing nascent RNA from promoter-
proximal pause sites. We attribute this to the reduced run-on efficiency of paused
Pol II in the absence of a high concentration of strong detergent²⁷." The reference
#27 in the manuscript is the right reference (Line #453 - Core, L. J. et al. Defining
the status of RNA polymerase at promoters. Cell reports 2, 454 1025–1035). I think
the confusion arose due to the two reference lists - one for the main manuscript
and one for the supplementary file. This confusion will be resolved in the final
publication as the two reference lists will not appear in the same document as they
currently do in the submitted manuscript.

c. Ext Fig. 5 says 12, 120 and 1200 cells, text says 100k, 10k and 1k nuclei.
We thank the reviewer for highlighting the ambiguity that arose from our failure to
clearly describe how 12, 120, and 1200 cells per well correspond to 1K, 10K, and
100K total cells, respectively. The main text and the supplementary file have been
amended to resolve the confusion.

- 872 d. The text always talks about "reads", while I believe it is "deduplicated reads". I
suggest to refer to them as UMIs.
We have made changes to replace reads with UMIs. We thank the reviewer for
this suggestion.
- e. The manuscript shows a lot of log-log scatterplots. It is not clear where the zeros
are (pseudocounts?), and what the colorscale is showing.
We apologize for the confusion created by the scatterplots in the log-log scale. We
have not used pseudo counts in plotting or analyses. The linear fit is performed on
linear data, and the data points are plotted on the log-log scale for visualization
purposes only. The zeros are considered in the linear fit but are removed from the
plot due to the log transformation of zero, resulting in an infinite value. The scatter
plots are plotted using the 'geom_pointdensity()' function, where the color scale
indicates the number of neighboring points. The legend title is added to all scatter
plots. We thank the reviewer for bringing this to our attention.
- f. Line 149f: Please show the correlation excluding the promoter-proximal region!
We think the reviewer is referring to the sentences in lines 146-148: "However,
scGRO-seq is less efficient in capturing nascent RNA from promoter-proximal
pause sites. We attribute this to the reduced run-on efficiency of paused Pol II in
the absence of a high concentration of strong detergent." The plot referring to this
description is **Extended Data Figure 7c**. The correlation between scGRO-seq and
PRO-seq in this figure shows gene body regions as indicated in the x-axis and y-
axis, which precisely means the exclusion of the promoter-proximal region.
- 897 g. Fig 1f: "Intron seqFISH (reads per cell)"; it is not reads!
We have changed the reads to counts.
- 900 h. Fig 1g: It is not clear which scRNA-seq data set that is.
We have added the source of the scRNA-seq data to the figure legend. This
information is also present in the External Data section of the supplementary file.
- i. It is not described how the fdrs in "evidence for bursting" for the data from Fig 2b
were estimated.
The false discovery rates in "evidence for bursting" in **Figure 2b** were estimated
by comparing the observed data against the permuted data.
- j. Fig 2c: How was a KS test computed from this? Why does the x axis stop at 2.5kb
if the window is 10kb?
The KS test was computed between the two distributions of distances between
consecutive RNA polymerases between the observed and permuted data, as
shown in **Extended Data Figure 8a** (left panel). **Figure 2c** shows the ratio of RNA
polymerase pairs (observed data over permuted data) in 50 bp bins for ease of
visualization. We show distances up to 2.5 kb to highlight the closely spaced Pol

lts (short distance between them). The full data extending up to 10 kb is shown in **Figure R3.9**.

Figure R3.9. Ratio of the observed distance between consecutive RNA polymerases in the first 10 kb of gene-bodies in individual cells against the permuted data.

- k. Line 227f: "Genes with the TATA element exhibited a larger burst size than genes lacking it, and the presence of the Initiator sequence further increased the burst size" - p values are required to back this claim.
We thank the reviewer for bringing this oversight to our attention. P-values are provided for the promoter elements comparisons in the main text.
- l. Fig 5a: How were KS tests performed? Why is there a drop by 50% in the left most bin for uncorrelated pairs?
The asymptotic two-sample Kolmogorov-Smirnov test was performed using the `ks.test()` function in R to examine if the correlated and uncorrelated distributions came from the same distribution. The drop in the leftmost bin was a result of the `geom_histogram()` function in R. The `geom_histogram()` function with `bins = x` and by not stating `xlim` [`geom_histogram(bins = 25, mapping = aes(y = after_stat(density)))`] would result in an unintended plotting behavior as observed in Fig. 5a. We corrected the function to [`geom_histogram(binwidth = 100000, boundary = 0, closed = "right," mapping = aes(y = after_stat(density)))`], which resolves the issue. We thank the reviewer for drawing our attention to this error.
- m. Methods: Better descriptions of the computational approaches in general are required. One example: The provided code hints at a custom definition of transcriptional units using `groHMM`, this is not described. Other example: Were there cells that were filtered out? (Based on Fig. 1c it seems as if cells were filtered

by a threshold on features per cell - it true, reporting 1503 features on average per
cell is not reasonable).

We apologize for the lack of description of the custom definition of transcriptional
units and thresholds applied for filtering cells. We have added two new sub-
sections, "Filtering Experimental batches and cells" and "Transcription Unit
calling," in the Methods section and incorporated additional explanations in the
main manuscript.

Overall, we thank the reviewer for their constructive comments on the
computational analyses and for encouraging us to improve the description of
computational methods. We have made our best attempt to add explanations
where necessary and to enhance the clarity and readability of the method section.
We sincerely believe that these suggestions have improved the manuscript.

**References:**

- Audibert, A., Weil, D., Dautry, F., 2002. In Vivo Kinetics of mRNA Splicing and
Transport in Mammalian Cells. *Mol. Cell. Biol.* 22, 6706–6718.
- Bainbridge, M.N., Warren, R.L., Hirst, M., Romanuik, T., Zeng, T., Go, A., Delaney, A.,
Griffith, M., Hickenbotham, M., Magrini, V., Mardis, E.R., Sadar, M.D., Siddiqui, A.S.,
Marra, M.A., Jones, S.J., 2006. Analysis of the prostate cancer cell line LNCaP
transcriptome using a sequencing-by-synthesis approach. *BMC Genom.* 7, 246.
- Buenrostro, J.D., Giresi, P.G., Zaba, L.C., Chang, H.Y., Greenleaf, W.J., 2013.
Transposition of native chromatin for fast and sensitive epigenomic profiling of open
chromatin, DNA-binding proteins and nucleosome position. *Nat. Methods* 10, 1213–
1218.
- Buenrostro, J.D., Wu, B., Litzenburger, U.M., Ruff, D., Gonzales, M.L., Snyder, M.P.,
Chang, H.Y., Greenleaf, W.J., 2015. Single-cell chromatin accessibility reveals
principles of regulatory variation. *Nature* 523, 486–490.
- Byun, J.S., Fufa, T.D., Wakano, C., Fernandez, A., Haggerty, C.M., Sung, M.-H.,
Gardner, K., 2012. ELL facilitates RNA polymerase II pause site entry and release.
*Nature communications* 3, 633.
- Chari, T., Pachter, L., 2023. The specious art of single-cell genomics. *PLOS Comput.*
*Biol.* 19, e1011288.
- CLEMENT, J.Q., QIAN, L., KAPLINSKY, N., WILKINSON, M.F., 1999. The stability and
fate of a spliced intron from vertebrate cells. *RNA* 5, 206–220.
- Core, L.J., Waterfall, J.J., Lis, J.T., 2008. Nascent RNA sequencing reveals widespread
pausing and divergent initiation at human promoters. *Science* 322, 1845–1848.
- Coulon, A., Ferguson, M.L., Turrís, V. de, Palangat, M., Chow, C.C., Larson, D.R.,
2014. Kinetic competition during the transcription cycle results in stochastic RNA
processing. *eLife* 3, e03939.
- Cui, Y., Irudayaraj, J., 2015. Inside single cells: quantitative analysis with advanced
optics and nanomaterials. *Wiley Interdiscip. Rev.: Nanomed. Nanobiotechnology* 7,
387–407.
- Hu, S., Metcalf, E., Mahat, D.B., Chan, L., Sohal, N., Chakraborty, M., Hamilton, M.,
Singh, Arundeeep, Singh, Abhyudai, Lees, J.A., Sharp, P.A., Garg, S., 2022.
Transcription factor antagonism regulates heterogeneity in embryonic stem cell states.
*Mol. Cell* 82, 4410-4427.e12.
- Klein, A.M., Mazutis, L., Akartuna, I., Tallapragada, N., Veres, A., Li, V., Peshkin, L.,
Weitz, D.A., Kirschner, M.W., 2015. Droplet Barcoding for Single-Cell Transcriptomics
Applied to Embryonic Stem Cells. *Cell* 161, 1187–1201.
- Landry, J.J.M., Pyl, P.T., Rausch, T., Zichner, T., Tekkedil, M.M., Stütz, A.M., Jauch, A.,
Aiyar, R.S., Pau, G., Delhomme, N., Gagneur, J., Korbel, J.O., Huber, W., Steinmetz,
996 L.M., 2013. The Genomic and Transcriptomic Landscape of a HeLa Cell Line. *G3:
GenesGenomesGenet.* 3, 1213–1224.
- Macosko, E.Z., Basu, A., Satija, R., Nemesh, J., Shekhar, K., Goldman, M., Tirosh, I.,
Bialas, A.R., Kamitaki, N., Martersteck, E.M., Trombetta, J.J., Weitz, D.A., Sanes, J.R.,
Shalek, A.K., Regev, A., McCarroll, S.A., 2015. Highly Parallel Genome-wide
Expression Profiling of Individual Cells Using Nanoliter Droplets. *Cell* 161, 1202–1214.
- Macville, M., Schröck, E., Padilla-Nash, H., Keck, C., Ghadimi, B.M., Zimonjic, D.,
Popescu, N., Ried, T., 1999. Comprehensive and definitive molecular cytogenetic

characterization of HeLa cells by spectral karyotyping. *Cancer Res.* 59, 141–50.
Marinov, G.K., Williams, B.A., McCue, K., Schroth, G.P., Gertz, J., Myers, R.M., Wold,
B.J., 2014. From single-cell to cell-pool transcriptomes: Stochasticity in gene expression
and RNA splicing. *Genome Res.* 24, 496–510.
Neugebauer, K.M., 2019. Nascent RNA and the Coordination of Splicing with
Transcription. *Cold Spring Harb. Perspect. Biol.* 11, a032227.
Patange, S., Ball, D.A., Wan, Y., Karpova, T.S., Girvan, M., Levens, D., Larson, D.R.,
2022. MYC amplifies gene expression through global changes in transcription factor
dynamics. *Cell Rep.* 38, 110292.
Rabani, M., Levin, J.Z., Fan, L., Adiconis, X., Raychowdhury, R., Garber, M., Gnirke, A.,
Nusbaum, C., Hacohen, N., Friedman, N., Amit, I., Regev, A., 2011. Metabolic labeling
of RNA uncovers principles of RNA production and degradation dynamics in
mammalian cells. *Nat. Biotechnol.* 29, 436–442.
Rabani, M., Raychowdhury, R., Jovanovic, M., Rooney, M., Stumpo, D.J., Pauli, A.,
Hacohen, N., Schier, A.F., Blackshear, P.J., Friedman, N., Amit, I., Regev, A., 2014.
High-Resolution Sequencing and Modeling Identifies Distinct Dynamic RNA Regulatory
Strategies. *Cell* 159, 1698–1710.
Shah, S., Takei, Y., Zhou, W., Lubeck, E., Yun, J., Eng, C.-H.L., Koulena, N., Cronin,
C., Karp, C., Liaw, E.J., Amin, M., Cai, L., 2018. Dynamics and Spatial Genomics of the
Nascent Transcriptome by Intron seqFISH. *Cell*.
Singh, J., Padgett, R.A., 2009. Rates of in situ transcription and splicing in large human
genes. *Nature structural & molecular biology* 16, 1128–1133.
Tang, F., Barbacioru, C., Wang, Y., Nordman, E., Lee, C., Xu, N., Wang, X., Bodeau, J.,
Tuch, B.B., Siddiqui, A., Lao, K., Surani, M.A., 2009. mRNA-Seq whole-transcriptome
analysis of a single cell. *Nat Methods* 6, 377–382.
Tome, J.M., Tippens, N.D., Lis, J.T., 2018. Single-molecule nascent RNA sequencing
identifies regulatory domain architecture at promoters and enhancers. *Nature Genetics*
322, 1845.

Reviewer Reports on the First Revision:

Referees' comments:

Referee #1 (Remarks to the Author):

The authors have responded to our concerns and suggestions in a thorough manner, and they have made significant changes that have improved the manuscript. Some of the requests for additional experiments were seen by the authors as beyond the scope of the study or would significantly delay publication. I appreciate their arguments and agree that it will be useful to the scientific community to have this published as soon as possible.

I have a very minor point below that the authors may want to check.

1. Line 255. Authors please check if references 48 and 49 belong before the comma.

Referee #2 (Remarks to the Author):

The authors have performed additional experiments that have nicely addressed my concerns about controlling for the identity of the propargyl nucleotide have any outsized effect on the click chemistry or reverse transcription. I support publication.

Referee #2 (Remarks on code availability):

This is outside of my area.

Referee #3 (Remarks to the Author):

The authors have addressed most of my points in a satisfactory manner. Based on their explanation they now gave in their replies I could now much better understand the (important) details of their data analysis methods. Unfortunately, not all of these explanations have found their way into the manuscript and one of my concerns remains. However, I am confident that my remaining concerns can be addressed by revising the text.

ad 1a) In their reply the authors claimed that "the median time required for introns to be spliced ranges from 5 to 10 minutes". This reflects only part of the literature. In the review from Karla Neugebauer (referenced by the authors in their reply) it ranges from 15 sec to 14 min, with most reports being below 5 min. If 15 sec to 14 min are used for the same calculation as done by the authors in their reply, Pol II would travel less than 1 kb or up to 35 kb during the time the intron can be detected by seqFISH. Moreover, the time to transcribe the intron, the time it takes until fluorescent probes detect introns, and the intron degradation kinetics will further increase the uncertainty in this travelling distance Pol II. In addition, genes with transcription units shorter than this distance would further bias the estimate of the capture efficiency. Thus, the 10% claimed by the

authors rather might be "something in between 1% and 20%". This number of 10% might not be "critical" here (as claimed by the authors, and I agree with that), but it might be for future studies involving scGRO-seq that will just refer back to this paper and assume the 10% to be true. This limitation should be mentioned in the discussion.

I am also a bit puzzled by the fact that the slope is now higher after restricting the analysis to the first 20 kb (0.26 as opposed to 0.23). Is this an effect of now not using an intercept term? Which parts of the transcription unit is used for which analysis must be clearly described in the methods (the same is true for the detail that the regressions are done without log).

ad 3) I appreciate the clarification by the authors especially under 6b. My understanding from the previous version was that in the gene x cell count matrix, each row was permuted randomly. Now this is better described in the main text, but a detailed description in the Methods section would avoid such a confusion and should be included.

**Author Rebuttals to First Revision:**

**Single-cell nascent RNA sequencing unveils coordinated global transcription**

Manuscript #: 2023-09-16626A

First Author: Dig Bijay Mahat (mahat@mit.edu)

Last Author: Phillip A. Sharp (sharppa@mit.edu)

**Response to Referees' comments to our initial response:**

**Referee #1 (Remarks to the Author):**

The authors have responded to our concerns and suggestions in a thorough manner, and
they have made significant changes that have improved the manuscript. Some of the
requests for additional experiments were seen by the authors as beyond the scope of the
study or would significantly delay publication. I appreciate their arguments and agree that
it will be useful to the scientific community to have this published as soon as possible.

*We thank the reviewer for their understanding and support. We are pleased that the
manuscript meets their expectations and look forward to its contribution to the scientific
community.*

I have a very minor point below that the authors may want to check. Line 255. Authors
please check if references 48 and 49 belong before the comma.

*We thank the reviewer for pointing out this error. The references belong before the
comma, which has been fixed in the manuscript.*

**Referee #2 (Remarks to the Author):**

The authors have performed additional experiments that have nicely addressed my
concerns about controlling for the identity of the propargyl nucleotide have any outsized
effect on the click chemistry or reverse transcription. I support publication.

*We appreciate the reviewer's support for publication and thank them for acknowledging
the efforts made to address their concerns.*

**Referee #3 (Remarks to the Author):**

The authors have addressed most of my points in a satisfactory manner. Based on their
explanation they now gave in their replies I could now much better understand the
(important) details of their data analysis methods. Unfortunately, not all of these
explanations have found their way into the manuscript and one of my concerns remains.
However, I am confident that my remaining concerns can be addressed by revising the
text.

1) In their reply the authors claimed that "the median time required for introns to be spliced
ranges from 5 to 10 minutes". This reflects only part of the literature. In the review from
Karla Neugebauer (referenced by the authors in their reply) it ranges from 15 sec to 14
44 min, with most reports being below 5 min. If 15 sec to 14 min are used for the same
calculation as done by the authors in their reply, Pol II would travel less than 1 kb or up

to 35 kb during the time the intron can be detected by seqFISH. Moreover, the time to
transcribe the intron, the time it takes until fluorescent probes detect introns, and the intron
degradation kinetics will further increase the uncertainty in this travelling distance Pol II.
In addition, genes with transcription units shorter than this distance would further bias the
estimate of the capture efficiency. Thus, the 10% claimed by the authors rather might be
"something in between 1% and 20%". This number of 10% might not be "critical" here (as
claimed by the authors, and I agree with that), but it might be for future studies involving
scGRO-seq that will just refer back to this paper and assume the 10% to be true. This
limitation should be mentioned in the discussion.

We thank the reviewer for their careful deliberation of the capture efficiency estimated in
our study. We agree that the 10% is an average approximation. We have, therefore,
added a phrase, "This estimate is based on the 8 minutes of median time required for
intron to be spliced out once it is transcribed, which ranges from 5 to 10 minutes according
to several studies using diverse methods⁵⁸⁻⁶⁴. Thus, the capture efficiency of 10% is an
average approximation and can vary among cells and batches" in the manuscript to reflect
this approximation as per the reviewer's request.

2) I am also a bit puzzled by the fact that the slope is now higher after restricting the
analysis to the first 20 kb (0.26 as opposed to 0.23). Is this an effect of now not using an
intercept term? Which parts of the transcription unit is used for which analysis must be
clearly described in the methods (the same is true for the detail that the regressions are
done without log).

The use of 20 kb, instead of 10 kb, increases the number of scGRO-seq reads used in
correlation analysis and, therefore, slightly increases the slope, reflecting that the scGRO-
seq UMIs per gene compared to intron seqFISH UMIs per gene are somewhat improved.

We thank the reviewer for their concern about the plots' description. The figure legends
are modified to indicate that "UMIs from the 500 bp regions from each end of the genes
and 250 bp regions from each end of the enhancers were removed to only include nascent
RNA from elongating RNA polymerases, and the data was plotted on a log-log scale to
show the range of data distribution" where relevant. Similarly, we have modified the
scGRO-seq vs intron seqFISH regression with "Correlation between scGRO-seq UMIs
78 per cell from up to the first 20 kb of genes and intron seqFISH counts per cell in the body
of genes used in the intron seqFISH study (n = 9,666)".

3) I appreciate the clarification by the authors especially under 6b. My understanding from
the previous version was that in the gene x cell count matrix, each row was permuted
randomly. Now this is better described in the main text, but a detailed description in the
Methods section would avoid such a confusion and should be included.

We are delighted to know that our response has addressed the reviewer's confusion
regarding the permutation method. We have improved its clarity in the Methods section
and in the main text where applicable.

characterization of HeLa cells by spectral karyotyping. *Cancer Res.* 59, 141–50.
Marinov, G.K., Williams, B.A., McCue, K., Schroth, G.P., Gertz, J., Myers, R.M., Wold,
B.J., 2014. From single-cell to cell-pool transcriptomes: Stochasticity in gene expression
and RNA splicing. *Genome Res.* 24, 496–510.
Neugebauer, K.M., 2019. Nascent RNA and the Coordination of Splicing with
Transcription. *Cold Spring Harb. Perspect. Biol.* 11, a032227.
Patange, S., Ball, D.A., Wan, Y., Karpova, T.S., Girvan, M., Levens, D., Larson, D.R.,
2022. MYC amplifies gene expression through global changes in transcription factor
dynamics. *Cell Rep.* 38, 110292.
Rabani, M., Levin, J.Z., Fan, L., Adiconis, X., Raychowdhury, R., Garber, M., Gnirke, A.,
Nusbaum, C., Hacohen, N., Friedman, N., Amit, I., Regev, A., 2011. Metabolic labeling
of RNA uncovers principles of RNA production and degradation dynamics in
mammalian cells. *Nat. Biotechnol.* 29, 436–442.
Rabani, M., Raychowdhury, R., Jovanovic, M., Rooney, M., Stumpo, D.J., Pauli, A.,
Hacohen, N., Schier, A.F., Blackshear, P.J., Friedman, N., Amit, I., Regev, A., 2014.
High-Resolution Sequencing and Modeling Identifies Distinct Dynamic RNA Regulatory
Strategies. *Cell* 159, 1698–1710.
Shah, S., Takei, Y., Zhou, W., Lubeck, E., Yun, J., Eng, C.-H.L., Koulena, N., Cronin,
C., Karp, C., Liaw, E.J., Amin, M., Cai, L., 2018. Dynamics and Spatial Genomics of the
Nascent Transcriptome by Intron seqFISH. *Cell*.
Singh, J., Padgett, R.A., 2009. Rates of in situ transcription and splicing in large human
genes. *Nature structural & molecular biology* 16, 1128–1133.
Tang, F., Barbacioru, C., Wang, Y., Nordman, E., Lee, C., Xu, N., Wang, X., Bodeau, J.,
Tuch, B.B., Siddiqui, A., Lao, K., Surani, M.A., 2009. mRNA-Seq whole-transcriptome
analysis of a single cell. *Nat Methods* 6, 377–382.
Tome, J.M., Tippens, N.D., Lis, J.T., 2018. Single-molecule nascent RNA sequencing
identifies regulatory domain architecture at promoters and enhancers. *Nature Genetics*
322, 1845.